# Decentralized Online Convex Optimization with Efficient Communication: Improved Algorithm and Lower Bounds

**Sifan Yang** [1 2]  **Wenhao Yang** [1 2]  **Wei Jiang** [3]  **Lijun Zhang** [1 2]

## Abstract

We investigate decentralized online convex optimization with compressed communication, where $n$ learners connected by a network collaboratively minimize a sequence of global loss functions using only local information and compressed data from their neighbors. Prior work has established regret bounds of $O(\max\{\omega^{-2}\rho^{-4}n^{1/2}, \omega^{-4}\rho^{-8}\}n\sqrt{T})$ and $O(\max\{\omega^{-2}\rho^{-4}n^{1/2}, \omega^{-4}\rho^{-8}\}n\ln T)$ for convex and strongly convex functions, respectively, where $\omega \in (0, 1]$ is the compression quality factor and $\rho < 1$ is the spectral gap of the communication matrix. However, these regret bounds suffer from a prohibitively high *quadratic* or even *quartic* dependence on $\omega^{-1}$. Moreover, the *superlinear* dependence on $n$ is also undesirable. To overcome these shortcomings, we propose a novel algorithm that achieves improved regret bounds of $\tilde{O}(\omega^{-1/2}\rho^{-1}n\sqrt{T})$ and $\tilde{O}(\omega^{-1}\rho^{-2}n\ln T)$ for convex and strongly convex functions, respectively. The primary idea is to design a *two-level blocking update framework* incorporating two novel ingredients: an online gossip strategy and an error compensation scheme, which work together to *promote better consensus* among learners. Furthermore, we establish the first lower bounds for this problem, justifying the optimality of our results with respect to both $\omega$ and $T$. Additionally, we consider the bandit feedback scenario and extend our method with classical gradient estimators to enhance existing regret bounds.

[1]State Key Laboratory of Novel Software Technology, Nanjing University, Nanjing 210023, China [2]School of Artificial Intelligence, Nanjing University, Nanjing 210023, China [3]School of Computer Science and Engineering, Nanjing University of Science and Technology, Nanjing 210094, China. Correspondence to: Lijun Zhang <zhanglj@lamda.nju.edu.cn>.

*Proceedings of the $43^{rd}$ International Conference on Machine Learning*, Seoul, South Korea. PMLR 306, 2026. Copyright 2026 by the author(s).

## 1. Introduction

Decentralized online convex optimization (D-OCO) (Yan et al., 2012; Hosseini et al., 2013) has emerged as a fundamental framework for modeling decentralized real-world problems with streaming data, such as tracking in sensor networks (Li et al., 2002; Lesser et al., 2003) and dynamic packet routing (Awerbuch & Kleinberg, 2004). Specifically, it is formulated as an iterative game between an adversary and a set of local learners, indexed by $1, \ldots, n$, which are connected through a *network* defined by an undirected graph $\mathcal{G} = ([n], E)$ with $E \subseteq [n] \times [n]$. In each round $t \in [T]$, learner $i \in [n]$ plays a decision $\mathbf{x}_i(t)$ from a convex set $\mathcal{X} \subseteq \mathbb{R}^d$. Subsequently, the adversary chooses a group of convex loss functions $f_{t,i}(\cdot) \colon \mathbb{R}^d \to \mathbb{R}$ and learner $i$ receives a loss $f_{t,i}(\mathbf{x}_i(t))$. The goal of learner $i$ is to minimize the cumulative loss in terms of the global function $f_t(\mathbf{x}) = \sum_{j=1}^{n} f_{t,j}(\mathbf{x})$ over $T$ rounds, which is equivalent to minimizing the *global regret*

$$R(T, i) = \sum_{t=1}^{T}\sum_{j=1}^{n} f_{t,j}(\mathbf{x}_i(t)) - \min_{\mathbf{x}\in\mathcal{X}} \sum_{t=1}^{T}\sum_{j=1}^{n} f_{t,j}(\mathbf{x}). \quad (1)$$

Previous work (Wan et al., 2024c) has established the nearly optimal regret bounds of $O(\rho^{-1/4}n\sqrt{T\ln n})$ and $O(\rho^{-1/2}n\ln n\ln T)$ for convex and strongly convex loss functions, respectively, where $\rho < 1$ is the spectral gap of the communication matrix.

The key difficulty in D-OCO lies in the fact that each learner only has access to its local function $f_{t,i}(\mathbf{x})$. To approximate the global loss $f_t(\mathbf{x})$, prior studies (Yan et al., 2012; Hosseini et al., 2013; Zhang et al., 2017; Wan et al., 2020; Wang et al., 2023; Li et al., 2023; Wan et al., 2024a; Wang et al., 2025; Wan et al., 2025; Wan, 2025) adopt the gossip protocols to aggregate information about the global loss function, where each learner communicates with its neighbors based on a weight matrix. Nevertheless, the information (e.g., gradients) transmitted among learners incurs significant *communication overhead* when the number of learners $n$ is large, which limits the practical applicability of the existing methods in decentralized problems.

To alleviate the communication bottleneck in D-OCO, Tu et al. (2022) propose a communication-efficient algo-

*Table 1.* A comparison of our work with existing results (Tu et al., 2022) for D-OCO with compressed communication. Here, $n$ is the number of learners, $\rho < 1$ is the spectral gap of the communication matrix and $\omega \in (0, 1]$ is the compression ratio.

| Source | Loss functions | Regret bounds |
|---|---|---|
| Tu et al. (2022) | Convex | $O(\max\{\omega^{-2}\rho^{-4}n^{1/2}, \omega^{-4}\rho^{-8}\}n\sqrt{T})$ |
| | Strongly convex | $O(\max\{\omega^{-2}\rho^{-4}n^{1/2}, \omega^{-4}\rho^{-8}\}n\ln T)$ |
| **This work** | Convex | $O(\omega^{-1/2}\rho^{-1}n\sqrt{\ln n}\sqrt{T})$ |
| | Strongly convex | $O(\omega^{-1}\rho^{-2}n\ln n\ln T)$ |
| **Lower bound** | Convex | $\Omega(\omega^{-1/2}\rho^{-1/4}n\sqrt{T})$ |
| | Strongly convex | $\Omega(\omega^{-1}\rho^{-1/2}n\ln T)$ |

rithm by leveraging data compression techniques (Tang et al., 2018; Koloskova et al., 2019) to reduce the volume of the transmitted information. Their method achieves regret bounds of $O(\max\{\omega^{-2}\rho^{-4}n^{1/2}, \omega^{-4}\rho^{-8}\}n\sqrt{T})$ and $O(\max\{\omega^{-2}\rho^{-4}n^{1/2}, \omega^{-4}\rho^{-8}\}n\ln T)$ for convex and strongly convex loss functions, respectively, where $\omega \in (0, 1]$ is the compression ratio that characterizes the quality of compression ($\omega = 1$ means no compression). However, their regret bounds suffer from a prohibitively high *quadratic* or even *quartic* dependence on $\omega^{-1}$. To enhance the communication efficiency, it is common to employ a compressor with $\omega \ll 1$. In this case, the theoretical guarantees of their method degrade significantly. Moreover, the dependence on $n$ is far from that in $\Omega(\rho^{-1/4}n\sqrt{T})$ and $\Omega(\rho^{-1/2}n\ln T)$ lower bounds for convex and strongly convex functions in D-OCO (Wan et al., 2025). Thus, it is natural to ask whether *the regret bounds in D-OCO with compressed communication could be further improved*.

**Results.** In this paper, we provide an affirmative answer to this question. Specifically, we develop a novel algorithm termed T̲wo-level C̲ompressed D̲ecentralized O̲nline G̲radient D̲escent (Top-DOGD), which enjoys better regret bounds of $\tilde{O}(\omega^{-1/2}\rho^{-1}n\sqrt{T})$ and $\tilde{O}(\omega^{-1}\rho^{-2}n\ln T)$ for convex and strongly convex functions, respectively.[1] Furthermore, we establish nearly matching lower bounds of $\Omega(\omega^{-1/2}\rho^{-1/4}n\sqrt{T})$ and $\Omega(\omega^{-1}\rho^{-1/2}n\ln T)$ for convex and strongly convex functions, respectively, which are the first lower bounds for this problem. To demonstrate the significance of our work, we present a comparison of our results with those of Tu et al. (2022) in Table 1. Additionally, we consider the bandit feedback setting, and extend our method with classical gradient estimators (Flaxman et al., 2005; Agarwal et al., 2010; Shamir, 2017). Let $d$ denote the dimensionality. In the one-point bandit feedback set-

ting, we enhance the existing regret bounds for convex and strongly convex functions to $\tilde{O}(\omega^{-1/4}\rho^{-1/2}d^{1/2}nT^{3/4})$ and $\tilde{O}(\omega^{-1/3}\rho^{-2/3}d^{2/3}nT^{2/3}(\ln T)^{1/3})$, respectively. In the two-point bandit feedback setting, we improve the existing regret bounds for convex and strongly convex functions to $\tilde{O}(\omega^{-1/2}\rho^{-1}d^{1/2}n\sqrt{T})$ and $\tilde{O}(\omega^{-1}\rho^{-2}dn\ln T)$, respectively. We compare our results with previous work in the bandit feedback setting in Table 2. Experimental results on online classification, which can be found in Appendix G, demonstrate the effectiveness of our methods.

**Techniques.** The technical contributions of this paper lie in the development of two novel online strategies to mitigate the impacts of network topology, compression and projection on the regret in D-OCO with compressed communication, together with a *unified framework* that integrates them. In particular, the joint effects consist of three components: consensus error, compression error, and projection error. To control the first two, we devise an online compressed gossip strategy, which is realized through *multiple steps of gossip*. To avoid the coupling between the projection error and the compression error, we propose a projection error compensation scheme, which *recursively compresses the residual* of the projection error and transmits the data to neighbors at every step. However, both of these techniques inherently require multiple communication rounds per update, which is not allowed in D-OCO. To overcome this dilemma, we design a *two-level blocking update framework*. Specifically, we divide the total $T$ rounds into blocks of size $L = L_1 + L_2$ and only update the decision *once* at the end of each block. Within each block, we first apply the online compressed gossip strategy over $L_1$ rounds, and then perform the projection error compensation scheme over $L_2$ rounds. Since we only update the decision once per block, we can evenly distribute the communications across rounds. By selecting appropriate block sizes $L_1$ and $L_2$, we can improve the regret bound while ensuring a single communication per round.

---

[1]We use the $\tilde{O}(\cdot)$ notation to hide only constant factors and polylogarithmic factors in $n$.

*Table 2.* A comparison of our work with existing results (Tu et al., 2022) for D-OCO with compressed communication under the bandit feedback setting. Here, $d$ is the dimensionality, (1) and (2) denote one-point and two-point bandit feedback settings, respectively.

| Source | Settings | Regret bounds |
|---|---|---|
| Tu et al. (2022) | Convex (1) | $O\left(\max\{\omega^{-1}\rho^{-2}n^{1/4}, \omega^{-2}\rho^{-4}\}d^{1/2}nT^{3/4}\right)$ |
| | Strongly convex (1) | $O(\max\{\omega^{-2/3}\rho^{-4/3}n^{1/6}, \omega^{-4/3}\rho^{-8/3}\}d^{2/3}nT^{2/3}(\ln T)^{1/3})$ |
| | Convex (2) | $O(\max\{\omega^{-2}\rho^{-4}n^{1/2}, \omega^{-4}\rho^{-8}\}dn\sqrt{T})$ |
| | Strongly convex (2) | $O(\max\{\omega^{-2}\rho^{-4}n^{1/2}, \omega^{-4}\rho^{-8}\}d^2 n \ln T)$ |
| This work | Convex (1) | $O(\omega^{-1/4}\rho^{-1/2}d^{1/2}n(\ln n)^{1/4}T^{3/4})$ |
| | Strongly convex (1) | $O(\omega^{-1/3}\rho^{-2/3}d^{2/3}n(\ln n)^{1/3}T^{2/3}(\ln T)^{1/3})$ |
| | Convex (2) | $O(\omega^{-1/2}\rho^{-1}d^{1/2}n\sqrt{\ln n}\sqrt{T})$ |
| | Strongly convex (2) | $O(\omega^{-1}\rho^{-2}dn \ln n \ln T)$ |

**Notation.** We denote by $\mathbf{x}_i(t)$ the decision of learner $i$ in round $t$, and by $\Pi_{\mathcal{X}}[\mathbf{x}] = \arg\min_{\mathbf{y} \in \mathcal{X}} \|\mathbf{y} - \mathbf{x}\|$ the Euclidean projection onto domain $\mathcal{X}$.

**Organization.** The rest of the paper is organized as follows. In Section 2.1, we briefly review related work on D-OCO. In Section 3, we introduce preliminaries and present our main results for D-OCO with compressed communication. In Section 4, we extend our methods to bandit feedback settings. Finally, in Section 5, we conclude the paper and discuss future work. All proofs are provided in appendices.

## 2. Related Work

In this section, we briefly review related work on decentralized online convex optimization and compressed communication. Due to space limitations, we provide the additional discussion in Appendix A.

### 2.1. Decentralized Online Convex Optimization (D-OCO)

D-OCO is a generalization of online convex optimization (Hazan, 2016) with $n \geq 2$ local learners connected through a network defined by an undirected graph $\mathcal{G} = ([n], E)$ with $E \subseteq [n] \times [n]$. Different from single-node OCO, each learner $i$ in D-OCO aims to minimize the regret with respect to the global function $f_t(\mathbf{x}) = \sum_{j=1}^n f_{t,j}(\mathbf{x})$, while only having access to its local function $f_{t,i}(\mathbf{x})$ and the information from its neighbors. The pioneering work of Yan et al. (2012) proposes a decentralized variant of OGD (Zinkevich, 2003), named D-OGD, by directly applying the standard gossip step (Xiao & Boyd, 2004) to the local decisions, and performing a gradient descent update with the gradient of the local function. D-OGD achieves $O(\rho^{-1/2}n^{5/4}\sqrt{T})$

and $O(\rho^{-1}n^{3/2}\ln T)$ regret bounds for convex and strongly convex loss functions, respectively. Later, Hosseini et al. (2013) develop a decentralized variant of FTRL (Hazan et al., 2007), termed D-FTRL, which enjoys the same regret bounds as D-OGD. Notably, there exist large gaps between these bounds and the lower bounds established by Wan et al. (2022), i.e., $\Omega(n\sqrt{T})$ and $\Omega(n)$ lower bounds for convex and strongly convex functions. To fill these gaps, Wan et al. (2024c) design an online accelerated gossip strategy and enhance the regret bounds to $O(\rho^{-1/4}n\sqrt{\ln n}\sqrt{T})$ and $O(\rho^{-1/2}n \ln n \ln T)$. They further demonstrate the optimality of these upper bounds by deriving tighter $\Omega(\rho^{-1/4}n\sqrt{T})$ and $\Omega(\rho^{-1/2}n \ln T)$ lower bounds for convex and strongly convex functions.

In large-scale scenarios, the efficacy of D-OCO algorithms may be limited by the communication overhead associated with information exchange. To alleviate communication costs, several works (Tu et al., 2022; Yuan et al., 2022; Cao & Bacsar, 2023; Zhang et al., 2023) seek to transmit the compressed data $\mathcal{C}(\mathbf{x})$ with fewer bits instead of broadcasting the full vector $\mathbf{x}$, where $\mathbf{x} \in \mathbb{R}^d$ and $\mathcal{C}(\cdot) : \mathbb{R}^d \to \mathbb{R}^d$ is a compression operator such that $\mathcal{C}(\mathbf{x})$ can be more efficiently transmitted. In particular, Tu et al. (2022) first propose a communication-efficient method by leveraging the compressed communication (Koloskova et al., 2019), and establish $O(\max\{\omega^{-2}\rho^{-4}n^{1/2}, \omega^{-4}\rho^{-8}\}n\sqrt{T})$ and $O(\max\{\omega^{-2}\rho^{-4}n^{1/2}, \omega^{-4}\rho^{-8}\}n \ln T)$ regret bounds for convex and strongly convex loss functions, respectively. Moreover, they consider the bandit settings, where the learner only has access to the loss value. Tu et al. (2022) extend their method to bandit feedback settings by employing the gradient estimators (Flaxman et al., 2005; Agarwal et al., 2010). A contemporaneous work (Cao & Bacsar, 2023) adopts an algorithmic design similar to that of Tu

et al. (2022) and provides the same regret bound for convex loss functions in D-OCO with compressed communication.

We notice that Yang et al. (2026) explores federated online convex optimization with compressed communication, where $n$ learners connected to a central server seek to minimize a sequence of global loss functions. However, both their problem formulation and algorithmic design are fundamentally different from those in this paper due to the distinction in network topologies.

## 2.2. Compressed Communication

In order to reduce the volume of data exchanged between learners in decentralized settings, there have been several offline optimization methods that transmit the compressed information $\mathcal{C}(\mathbf{x})$ instead of the full vector $\mathbf{x}$, where $\mathbf{x} \in \mathbb{R}^d$ and $\mathcal{C}(\cdot) : \mathbb{R}^d \to \mathbb{R}^d$ is an operator chosen such that $\mathcal{C}(\mathbf{x})$ can be efficiently represented. The mainstream communication compression techniques can be summarized into two classes: unbiased compressor (Jiang & Agrawal, 2018; Tang et al., 2018; Jiang et al., 2024) and biased (contractive) compressor (Seide et al., 2014; Wangni et al., 2018; Stich et al., 2018).

An unbiased compressor produces an output $\mathcal{C}(\mathbf{x})$ such that $\mathbb{E}[\mathcal{C}(\mathbf{x})] = \mathbf{x}$ for any input $\mathbf{x} \in \mathbb{R}^d$. A classical example is stochastic quantization, which efficiently represents high-precision data using fewer bits. In contrast, a contractive compressor relaxes the unbiasedness assumption to yield a biased estimate with small variance. A prominent approach in this category is sparsification, which constructs a sparse vector by retaining only a subset of coordinates. To further reduce the compression error of the compressor, Huang et al. (2022) introduce the fast compressed communication (repeated compressor). The core idea is to recursively compress information for $L$ rounds and transmit the compressed data at each step, which reduces the compression error exponentially with respect to the number of rounds $L$. While using an unbiased compressor may achieve better theoretical guarantees, contractive compressors can offer comparable and even superior empirical performance under weaker assumptions. Therefore, following Koloskova et al. (2019), we do not distinguish these two approaches, and refer to both of them as *compressors* in this paper.

## 3. Main Results

In this section, we first introduce preliminaries for D-OCO with compressed communication, including the assumptions and techniques employed in our algorithmic design. We then present our method that achieves improved regret bounds, and establish nearly matching lower bounds.

### 3.1. Preliminaries

Similar to prior work on D-OCO (Yan et al., 2012; Hosseini et al., 2013; Wan et al., 2024a; 2025), we introduce the following standard assumptions.

**Assumption 3.1** (Communication matrix). The communication matrix $P \in \mathbb{R}^{n \times n}$ is supported on the graph $\mathcal{G} = ([n], E)$, symmetric, and doubly stochastic, which satisfies

- $0 < P_{ij} < 1$ only if $(i, j) \in E$ or $i = j$,

- $\sum_{j=1}^{n} P_{ij} = \sum_{j \in \mathcal{N}_i} P_{ij} = 1, \forall i \in [n]$,

- $\sum_{i=1}^{n} P_{ij} = \sum_{i \in \mathcal{N}_j} P_{ij} = 1, \forall j \in [n]$,

where $\mathcal{N}_i$ denotes the set including the immediate neighbors of learner $i$ and itself. Following previous studies, we assume $P$ is given beforehand, instead of being a choice of the algorithm. Moreover, $P$ is positive semi-definite, and its second largest singular value denoted by $\sigma_2(P)$ is strictly smaller than 1. We define $\rho < 1$ as the spectral gap of $P$ and $\beta = \|I_n - P\| \in [0, 2]$.

**Assumption 3.2** (Convexity). The loss function $f_{t,i}(\cdot)$ of each learner $i \in [n]$ in every round $t \in [T]$ is convex over the feasible domain $\mathcal{X}$.

**Assumption 3.3** (Strong convexity). The loss function $f_{t,i}(\cdot)$ of each learner $i \in [n]$ in every round $t \in [T]$ is $\mu$-strongly convex over the domain $\mathcal{X}$, i.e., it holds that $f_{t,i}(\mathbf{y}) \geq f_{t,i}(\mathbf{x}) + \langle \nabla f_{t,i}(\mathbf{x}), \mathbf{y} - \mathbf{x} \rangle + \frac{\mu}{2} \|\mathbf{y} - \mathbf{x}\|^2$, for $\forall \mathbf{x}, \mathbf{y} \in \mathcal{X}$.

**Assumption 3.4** (Bounded gradient norm). The gradient of function $f_{t,i}(\cdot)$ of each learner $i \in [n]$ in every round $t \in [T]$ is bounded by $G$ over the domain $\mathcal{X}$, i.e., it holds that $\|\nabla f_{t,i}(\mathbf{x})\| \leq G$, for $\forall \mathbf{x} \in \mathcal{X}$.

**Assumption 3.5** (Bounded domain). The convex set $\mathcal{X}$ contains the origin $\mathbf{0}$, i.e., $\mathbf{0} \in \mathcal{X}$, and it is bounded by $D$, i.e., it holds that $\|\mathbf{x} - \mathbf{y}\| \leq D$, for $\forall \mathbf{x}, \mathbf{y} \in \mathcal{X}$.

A *compressor* $\mathcal{C}(\cdot) : \mathbb{R}^d \to \mathbb{R}^d$ is a mapping whose output can be encoded with fewer bits than the original input. In this paper, we consider a broad class of compressors with the following general property (Koloskova et al., 2019).

**Definition 3.6** (Compressor). A compression operator $\mathcal{C}(\cdot) : \mathbb{R}^d \to \mathbb{R}^d$ is termed an $\omega$-contractive compressor, if it satisfies

$$\mathbb{E}_{\mathcal{C}} \left[ \|\mathcal{C}(\mathbf{x}) - \mathbf{x}\|^2 \right] \leq (1 - \omega) \|\mathbf{x}\|^2, \forall \mathbf{x} \in \mathbb{R}^d,$$

for a parameter $\omega \in (0, 1]$. Here, $\mathbb{E}_{\mathcal{C}}[\cdot]$ denotes the expectation over the internal randomness of $\mathcal{C}(\cdot)$.

We provide several representative examples of compressors in Appendix B. To mitigate the error of the standard

---

**Algorithm 1** Repeated compressor $\mathcal{C}_L(\cdot)$

---

1: **Input:** compression round $L$, compressor $\mathcal{C}(\cdot)$, data $\mathbf{x}$
2: Initialize $\mathbf{c}_0 = \mathbf{0}$
3: **for** $i = 1$ to $L$ **do**
4:     Compute $\Delta_i = \mathcal{C}(\mathbf{x} - \mathbf{c}_{i-1})$
5:     Send $\Delta_i$ to its neighbors
6:     Calculate $\mathbf{c}_i = \mathbf{c}_{i-1} + \Delta_i$
7: **end for**
8: **Return:** $\mathbf{c}_L$

---

compressors, Huang et al. (2022) design the *repeated compressor*, as summarized in Algorithm 1. The core idea is to repeatedly apply the standard compressor for $L$ rounds and transmit the compressed data at each round, which involves $L$ rounds of communication. When $L = 1$, the repeated compressor degenerates to the standard compressor. We state the following lemma from Huang et al. (2022) to establish the guarantee of the repeated compressor.

**Lemma 3.7** (Lemma 2 in Huang et al. (2022)). *Given an $\omega$-contractive compressor $\mathcal{C}(\cdot)$ and for any compression rounds $L \geq 1$, Algorithm 1 ensures*

$$\mathbb{E}_{\mathcal{C}} \left[ \|\mathbf{c}_L - \mathbf{x}\|^2 \right] \leq (1 - \omega)^L \|\mathbf{x}\|^2, \forall \mathbf{x} \in \mathbb{R}^d.$$

**Remark:** Lemma 3.7 shows that the compression error of the repeated compressor decays exponentially with the compression rounds $L$, albeit at the cost of requiring $L$ communication rounds.

In the context of decentralized offline optimization with compressed communication, a standard approach involves integrating a compressor directly into the gossip strategy, where each learner transmits the compressed decision to its neighbors. However, this approach fails to achieve average consensus. To address this, Koloskova et al. (2019) adopt the difference compression technique (Tang et al., 2018) to develop Choco-gossip. Specifically, each learner $i$ maintains auxiliary variables $\hat{\mathbf{x}}_j(t) \in \mathbb{R}^d$ to record the data received from neighbors $j$, and $\hat{\mathbf{x}}_i(t) \in \mathbb{R}^d$ to track the data it has transmitted to neighbors over the past rounds. At each round $t$, learner $i$ updates its decision and auxiliary variables $\hat{\mathbf{x}}_j(t)$ as follows:

$$\mathbf{x}_i(t + 1) = \mathbf{x}_i(t) + \gamma \sum_{j \in \mathcal{N}_i} P_{ij}(\hat{\mathbf{x}}_j(t) - \hat{\mathbf{x}}_i(t))$$
$$\hat{\mathbf{x}}_j(t + 1) = \hat{\mathbf{x}}_j(t) + \mathcal{C}(\mathbf{x}_j(t + 1) - \hat{\mathbf{x}}_j(t)), \forall j \in \mathcal{N}_i, \quad (2)$$

where $\gamma \leq 1$ is the consensus step size and $\mathcal{C}(\mathbf{x}_j(t + 1) - \hat{\mathbf{x}}_j(t))$ is the received data from the neighbor $j$. One might notice that Choco-gossip requires each learner to store $\deg(i) + 2$ variables, where $\deg(i)$ is the degree of learner $i$. It is not necessary and Koloskova et al. (2019) present an efficient version that only involves *three* additional variables. We provide a comprehensive description in Appendix E.4.

In D-OCO with compressed communication, Tu et al. (2022) integrate Choco-gossip with D-OGD to develop a communication-efficient method, referred to as DC-DOGD. At each round $t$, each learner $i$ plays the decision $\mathbf{x}_i(t)$ and receives the local gradient $\nabla f_{t,i}(\mathbf{x}_i(t))$. Then, learner $i$ updates its decision by leveraging both the local gradient and the information $\hat{\mathbf{x}}_j(t)$ received from its neighbors:

$$\mathbf{x}_i(t + 1) = \Pi_{\mathcal{X}} \big[ \mathbf{x}_i(t) - \eta_t \nabla f_{t,i}(\mathbf{x}_i(t))$$
$$+ \gamma \sum_{j \in \mathcal{N}_i} P_{ij}(\hat{\mathbf{x}}_j(t) - \hat{\mathbf{x}}_i(t)) \big]$$
$$\hat{\mathbf{x}}_j(t + 1) = \hat{\mathbf{x}}_j(t) + \mathcal{C}(\mathbf{x}_j(t + 1) - \hat{\mathbf{x}}_j(t)), \forall j \in \mathcal{N}_i, \quad (3)$$

where $\eta_t$ is the learning rate. In contrast to unconstrained settings (Koloskova et al., 2019), each learner is required to project its decision onto the feasible domain in D-OCO, which inevitably introduces an extra *projection error*.

### 3.2. Our Improved Algorithm

To begin with, we briefly outline the key challenges in D-OCO with compressed communication and then present the corresponding techniques we develop to address them.

The regret of existing methods for D-OCO with compressed communication can be decomposed into two parts: the regret of the averaged decision $\overline{\mathbf{x}}(t) = \frac{1}{n} \sum_{i=1}^{n} \mathbf{x}_i(t)$ and the approximation error. Specifically, the approximation error consists of three components: *(i)* consensus error, arising from the network size and topology; *(ii)* compression error, introduced by the compressed communication; and *(iii)* projection error, caused by the projection operation in (3). To achieve tighter regret bounds, we focus on controlling the approximation error.

**Online compressed gossip strategy.** The large consensus error in existing approaches stems from the fact that local decisions converge to the average decision at a slow rate. To overcome this limitation, we leverage the multi-step gossip to achieve faster consensus convergence among local learners. More precisely, we state the following lemma to establish the convergence rate.

**Lemma 3.8** (Theorem 2 in Koloskova et al. (2019)). *We define*

$$e_t = \sum_{i=1}^{n} \|\mathbf{x}_i(t) - \overline{\mathbf{x}}(t)\|^2 + \sum_{i=1}^{n} \|\mathbf{x}_i(t) - \hat{\mathbf{x}}_i(t)\|^2.$$

*The first term $\sum_{i=1}^{n} \|\mathbf{x}_i(t) - \overline{\mathbf{x}}(t)\|^2$ characterizes the consensus error, and the second term $\sum_{i=1}^{n} \|\mathbf{x}_i(t) - \hat{\mathbf{x}}_i(t)\|^2$ characterizes the compression error.*

*Given an $\omega$-contractive compressor $\mathcal{C}(\cdot)$, for any round $t$, by setting $\gamma = \frac{\rho\omega}{16\rho + \rho^2 + 4\beta^2 + 2\rho\beta^2 - 8\rho\omega}$ and performing the*

*update in (2) for $L_1$ rounds, we can ensure*

$$\mathbb{E}_{\mathcal{C}}\left[e_{t+L_1}\right] \le \left(1 - \frac{\rho^2 \omega}{82}\right)^{L_1} \mathbb{E}_{\mathcal{C}}\left[e_t\right].$$

As can be observed, the consensus and compression errors decrease *at an exponential rate as the number of gossip rounds increases*. While repeatedly executing the gossip step can mitigate errors, it results in multiple communication rounds per update, which substantially exacerbates the communication burden we aim to alleviate. Motivated by Wan et al. (2024a; 2025), we integrate the blocking update mechanism with Choco-gossip to design an *online compressed gossip strategy*. By partitioning the total rounds into blocks and performing updates only at the end of each block, we can amortize the communications across rounds. With an appropriate block size, this allows us to control the errors while keeping only one communication per round. Nevertheless, this mechanism alone is insufficient, as the projection step in D-OCO introduces an additional projection error.

**Projection error compensation scheme.** The projection operation in (3) introduces an additional error, which *couples* with the compression error and further induces an $O(n)$ dependence in the approximation error. Through careful analysis, we find that if each learner $i$ were able to add the projection error of neighbor $j \in \mathcal{N}_i$ to the auxiliary variable $\hat{\mathbf{x}}_j(t)$, the $O(n)$ dependence could be avoided. However, each learner can only broadcast the compressed information, which leads to an $O(1 - \omega)$ error. Notably, if the projection error is constrained to the order of $O(1/n)$, the upper bound becomes independent of $n$, thereby *decoupling* the projection error from the compression error. Drawing inspiration from Huang et al. (2022), we make use of the repeated compressor to transmit the compressed information with a sufficiently small error. By recursively applying a compressor over $L_2 = \lceil \ln(8n)/\omega \rceil$ rounds, we can ensure that the compression error satisfies

$$\mathbb{E}_{\mathcal{C}}\left[\|\mathcal{C}_{L_2}(\mathbf{x}) - \mathbf{x}\|^2\right] \le (1 - \omega)^{L_2} \|\mathbf{x}\|^2 \le \frac{1}{8n} \|\mathbf{x}\|^2.$$

However, a direct application of this technique incurs $L_2$ communication rounds per update. To overcome this dilemma, we again utilize the blocking update mechanism to distribute the communications across the rounds within each block.

**Overall algorithm: a two-level blocking update structure.** To unify the two strategies within a single framework, we propose a *two-level blocking update framework*. Concretely, we divide the $T$ rounds into several blocks with size $L = L_1 + L_2$ (we assume $T/L$ is an integer without loss of generality). For learner $i$, we maintain the same decision

---

**Algorithm 2** Top-DOGD

1: **Input:** consensus step size $\gamma$, learning rate $\eta_b$, block size $L = L_1 + L_2$
2: **for** all learners $i = 1, \cdots, n$ **in parallel do**
3:     Initialize $\mathbf{x}_i(1) = \mathbf{0}, \hat{\mathbf{x}}_i(1) = \mathbf{0}, \forall i \in [n]$
4:     **for** block $b = 1$ to $T/L$ **do**
5:         **if** $b = 1$ **then**
6:             **for** $t = 1$ to $L$ **do**
7:                 Play the decision $\mathbf{x}_i(1)$
8:             **end for**
9:         **else**
10:             Receive the results $\mathbf{y}_i^{(L_1+1)}(b)$ and $\hat{\mathbf{y}}_i^{(L_1+1)}(b)$ from Algorithm 3
11:             Receive the result $\mathbf{r}_j^{(L_2+1)}(b+1)$ for all $j \in \mathcal{N}_i$ from Algorithm 4
12:             Update $\hat{\mathbf{x}}_j(b+1) = \hat{\mathbf{y}}_j^{(L_1+1)}(b) + \mathbf{r}_j^{(L_2+1)}(b+1)$ for all $j \in \mathcal{N}_i$
13:             Compute $\mathbf{z}_i(b) = \sum_{t=(b-1)L+1}^{bL} \nabla f_{t,i}(\mathbf{x}_i(b))$
14:             Update $\mathbf{x}_i(b+1) = \Pi_{\mathcal{X}}\left[\mathbf{y}_i^{(L_1+1)}(b)\right]$
15:         **end if**
16:     **end for**
17: **end for**

---

$\mathbf{x}_i(b)$ in block $b$ and only update the decision at the end of the block. In each block $b$, we first apply the online compressed gossip strategy for $L_1$ rounds and then perform the projection error compensation scheme for $L_2$ rounds. Our method, named Top-DOGD, is presented in Algorithm 2.

Specifically, for each learner $i$, we first initialize the decision $\mathbf{x}_i(1) = \mathbf{0} \in \mathbb{R}^d$ and the local replica $\hat{\mathbf{x}}_j(1) = \mathbf{0} \in \mathbb{R}^d$ to store the information from its neighbors $j \in \mathcal{N}_i$. In each block $b \ge 2$, we start by updating the surrogate decision $\mathbf{y}_i^{(1)}(b) = \mathbf{x}_i(b) - \eta_b \mathbf{z}_i(b-1)$, where $\mathbf{z}_i(b-1) = \sum_{t=(b-2)L+1}^{(b-1)L} \nabla f_{t,i}(\mathbf{x}_i(b-1))$ is the sum of gradients in block $b-1$, and set the local auxiliary variable $\hat{\mathbf{y}}_i^{(1)}(b) = \hat{\mathbf{x}}_i(b)$. Next, we perform our online compressed gossip strategy for $L_1$ rounds, which is summarized in Algorithm 3. For $b_1 \in [1, L_1]$, learner $i$ transmits $\mathcal{C}(\mathbf{y}_i^{(b_1)}(b) - \hat{\mathbf{y}}_i^{(b_1)}(b))$ to neighbor $j$. After receiving the information from its neighbors, learner $i$ updates $\hat{\mathbf{y}}_j^{(b_1+1)}(b)$ and then computes the surrogate decision as

$$\mathbf{y}_i^{(b_1+1)}(b) = \mathbf{y}_i^{(b_1)}(b) + \gamma \sum_{j \in \mathcal{N}_i} P_{ij}(\hat{\mathbf{y}}_j^{(b_1+1)}(b) - \hat{\mathbf{y}}_i^{(b_1)}(b)). \tag{4}$$

Within the second sub-block, we apply our projection error compensation scheme, which is presented in Algorithm 4. Each learner $i$ recursively compresses the residual of the projection error $\mathbf{r}_i(b+1) = \mathbf{x}_i(b+1) - \mathbf{y}_i^{(L_1+1)}(b)$ over $L_2$ rounds and sends compressed data to its neighbors per round.

---

**Algorithm 3** Online compressed gossip strategy

1: **Input:** consensus step size $\gamma$, block size $L_1$
2: **for** all learners $i = 1, \cdots, n$ **in block** $b$ **in parallel do**
3:     Set $\mathbf{y}_i^{(1)}(b) = \mathbf{x}_i(b) - \eta_b \mathbf{z}_i(b-1)$, $\hat{\mathbf{y}}_i^{(1)}(b) = \hat{\mathbf{x}}_i(b), b_1 = 1$
4:     **for** $t = (b-1)L + 1$ to $(b-1)L + L_1$ **do**
5:        Play the decision $\mathbf{x}_i(b)$
6:        Transmit $\mathcal{C}(\mathbf{y}_i^{(b_1)}(b) - \hat{\mathbf{y}}_i^{(b_1)}(b))$ to $j \in \mathcal{N}_i$
7:        Compute $\hat{\mathbf{y}}_j^{(b_1+1)}(b) = \hat{\mathbf{y}}_j^{(b_1)}(b) + \mathcal{C}(\mathbf{y}_j^{(b_1)}(b) - \hat{\mathbf{y}}_j^{(b_1)}(b))$ for $j \in \mathcal{N}_i$
8:        Compute $\mathbf{y}_i^{(b_1+1)}(b)$ according to (4) and set $b_1 = b_1 + 1$
9:     **end for**
10: **end for**
11: **Return**: $\mathbf{y}_i^{(L_1+1)}(b)$ and $\hat{\mathbf{y}}_i^{(L_1+1)}(b)$

---

At the end of the block $b$, learner $i$ updates its decision $\mathbf{x}_i(b+1) = \Pi_{\mathcal{X}}[\mathbf{y}_i^{(L_1+1)}(b)]$. In the following, we establish the theoretical guarantees of Top-DOGD for convex and strongly convex loss functions, respectively.

**Theorem 3.1.** *Let* $L_1 = \lceil \frac{2\ln(14n)}{\gamma\rho} \rceil, L_2 = \lceil \frac{\ln(8n)}{\omega} \rceil, L = L_1 + L_2 = O(\omega^{-1}\rho^{-2}\ln n), \eta_b = \eta = \frac{D}{G\sqrt{LT}}, \gamma = \frac{\omega\rho}{2\rho\beta^2 + 4\beta^2 + (2-\omega)(\beta^2+2\beta)\rho+\rho^2}$. *Under Assumptions 3.1, 3.2, 3.4 and 3.5, for any* $i \in [n]$ *and any convex loss functions, Algorithm 2 ensures*

$$\mathbb{E}_{\mathcal{C}}[R(T,i)] \leq O(n\sqrt{LT}) = O(\omega^{-1/2}\rho^{-1}n\sqrt{\ln n}\sqrt{T}).$$

**Theorem 3.2.** *Let* $L_1 = \lceil \frac{2\ln(14n)}{\gamma\rho} \rceil, L_2 = \lceil \frac{\ln(8n)}{\omega} \rceil, L = L_1 + L_2 = O(\omega^{-1}\rho^{-2}\ln n), \eta_b = \frac{1}{\mu(bL+8)}, \gamma = \frac{\omega\rho}{2\rho\beta^2 + 4\beta^2 + (2-\omega)(\beta^2+2\beta)\rho+\rho^2}$. *Under Assumptions 3.1, 3.3, 3.4 and 3.5, for any* $i \in [n]$ *and any* $\mu$-*strongly convex loss functions, Algorithm 2 ensures*

$$\mathbb{E}_{\mathcal{C}}[R(T,i)] \leq O(nL\ln T) = O(\omega^{-1}\rho^{-2}n\ln n\ln T).$$

**Remark:** The regret bounds of our Top-DOGD exhibit a tighter dependence on the compression factor $\omega$, the spectral gap $\rho$ and the network size $n$ compared to the previous regret bounds of $O(\max\{\omega^{-2}\rho^{-4}n^{1/2}, \omega^{-4}\rho^{-8}\}n\sqrt{T})$ and $O(\max\{\omega^{-2}\rho^{-4}n^{1/2}, \omega^{-4}\rho^{-8}\}n\ln T)$ (Tu et al., 2022). This enhancement is particularly critical in large-scale communication environments.

**Necessity of the two-level structure.** Our refined bounds result from the *synergistic* interplay of two proposed strategies, as neither alone is sufficient to achieve the desired improvement. To highlight their significance, we conduct an ablation analysis by considering two scenarios: (i) performing the online compressed gossip strategy with

---

**Algorithm 4** Projection error compensation scheme

1: **Input:** block size $L_2$
2: **for** all learners $i = 1, \cdots, n$ **in block** $b$ **in parallel do**
3:     Set $\mathbf{r}_i^{(1)}(b+1) = \mathbf{0}, b_2 = 1, \mathbf{r}_i(b+1) = \Pi_{\mathcal{X}}[\mathbf{y}_i^{(L_1+1)}(b)] - \mathbf{y}_i^{(L_1+1)}(b)$ and $\mathbf{r}_j^{(1)}(b+1) = \mathbf{0}$ for $j \in \mathcal{N}_i$
4:     **for** $t = (b-1)L + L_1 + 1$ to $bL$ **do**
5:        Play the decision $\mathbf{x}_i(b)$
6:        Compute $\Delta_i^{(b_2)}(b) = \mathcal{C}(\mathbf{r}_i(b+1) - \mathbf{r}_i^{(b_2)}(b+1))$
7:        Send $\Delta_i^{(b_2)}(b)$ to neighbor $j$
8:        Calculate $\mathbf{r}_i^{(b_2+1)}(b+1) = \mathbf{r}_i^{(b_2)}(b+1) + \Delta_i^{(b_2)}(b)$
9:        Update $\mathbf{r}_j^{(b_2+1)}(b+1) = \mathbf{r}_j^{(b_2)}(b+1) + \Delta_j^{(b_2)}(b)$ for $j \in \mathcal{N}_i$
10:        Set $b_2 = b_2 + 1$
11:     **end for**
12: **end for**
13: **Return**: $\mathbf{r}_i^{(L_2+1)}(b+1)$

---

$L_1 = 1$, and (ii) removing the projection error compensation scheme ($L_2 = 0$). First, when $L_1 = 1$, our method reduces to a combination of DC-DOGD (Tu et al., 2022) with the projection error compensation scheme, which does not improve the regret bounds of DC-DOGD. Although we can mitigate the projection error, we cannot reduce the consensus error and compression error as we desire. If $L_2 = 0$, we suffer the coupled errors in each round and only obtain $O(\omega^{-1/2}\rho^{-1}n^{5/4}\sqrt{\ln n}\sqrt{T})$ and $O(\omega^{-1}\rho^{-2}n^{3/2}\ln n\ln T)$ regret bounds for convex and strongly convex loss functions, respectively. While the dependence on $\omega$ and $\rho$ is still tighter than that in the regret bounds of Tu et al. (2022), the dependence on $n$ is worse than that of Top-DOGD.

### 3.3. Lower Bounds

In this section, we present lower bounds for convex and strongly convex loss functions in D-OCO with compressed communication. In D-OCO, Wan et al. (2025) have derived the lower bounds of $\Omega(\rho^{-1/4}n\sqrt{T})$ and $\Omega(\rho^{-1/2}n\ln T)$ for convex and strongly convex losses. Their analysis leverages the 1-connected cycle graph (Duchi et al., 2011), where the adversary can force at least one learner to suffer $\lceil n/4 \rceil$ rounds of communication delay before receiving the information of the global function $f_t(\mathbf{x})$. By leveraging this topology, they establish the aforementioned lower bounds.

The existing literature on D-OCO lacks lower bounds that explicitly characterize the dependence on the compression ratio $\omega$. In offline decentralized optimization, Huang et al. (2022) capture the effect of compression by utilizing a specific compressor and modeling compression as probabilistic communication failures. Motivated by prior work, we model

the compression effect by adopting the randomized gossip compressor $\mathcal{C}(\cdot) : \mathbb{R}^d \to \mathbb{R}^d$, which outputs $\mathcal{C}(\mathbf{x}) = \mathbf{x}$ with probability $\omega$ and $\mathcal{C}(\mathbf{x}) = \mathbf{0}$ otherwise, thereby requiring multiple rounds in expectation for full information transmission. Under this scheme, two connected learners $i$ and $j$ can successfully exchange data only with probability $\omega$ in each round. Consequently, the expected number of rounds required for a successful transmission is $\lceil 1/\omega \rceil$. Building on this construction, we establish the following lower bounds.

**Theorem 3.3.** *Given the feasible domain $\mathcal{X} = \left[\frac{-D}{2\sqrt{d}}, \frac{D}{2\sqrt{d}}\right]^d$ and $n = 2m + 2$ for some positive integer $m$. For any D-OCO algorithm, if $n \leq 8\omega T + 8\omega$, there exists a sequence of convex loss functions satisfying Assumption 3.4, a graph $\mathcal{G} = ([n], E)$, a compressor satisfying Definition 3.6, and a matrix $P$ satisfying Assumption 3.1 such that*

$$\mathbb{E}_{\mathcal{C}}\left[R(T, 1)\right] \geq \frac{nGD(\pi T)^{1/2}}{2^5 \rho^{1/4} \omega^{1/2}}.$$

**Theorem 3.4.** *Given the feasible domain $\mathcal{X} = \left[0, \frac{D}{\sqrt{d}}\right]^d$ and $n = 2m + 2$ for some positive integer $m$. For any D-OCO algorithm, if $16n \leq \omega T - \omega$, there exists a sequence of $\mu$-strongly convex loss functions satisfying Assumption 3.4 with $G = \mu D$, a graph $\mathcal{G} = ([n], E)$, a compressor satisfying Definition 3.6, and a matrix $P$ satisfying Assumption 3.1 such that*

$$\mathbb{E}_{\mathcal{C}}\left[R(T, 1)\right] \geq \frac{(n-2)\pi\mu D^2 (\log_{16}(30\omega(T-1)/n) - 2)}{2^{22}\omega\rho^{1/2}}.$$

**Remark:** We have established the lower bounds of $\Omega(\omega^{-1/2}\rho^{-1/4}n\sqrt{T})$ and $\Omega(\omega^{-1}\rho^{-1/2}n \ln T)$ for convex and strongly convex loss functions, respectively, which match the corresponding upper bounds up to the spectral gap $\rho$ and polylogarithmic factors in $n$.

**Remark:** When there is no compression ($\omega = 1$), our results reduce to the standard D-OCO lower bounds established by Wan et al. (2025). While the upper bounds for D-OCO algorithms are established for arbitrary graphs and compressors, the lower bounds for a specific instance suffice to certify the tightness of the upper bounds in the general case. We provide further discussion on our lower bounds in Appendix D.6.

# 4. Extension to Bandit Feedback Settings

In this section, we extend our method to the bandit feedback settings by employing classical gradient estimators, with more details provided in Appendix C. Following previous work (Flaxman et al., 2005; Agarwal et al., 2010; Shamir, 2017), we introduce an assumption specific to the bandit feedback settings.

**Assumption 4.1.** The convex set $\mathcal{X}$ contains the ball with radius $r$, and is contained in the ball with radius $R$, i.e., it holds that $r\mathcal{B} \subseteq \mathcal{X} \subseteq R\mathcal{B}, \mathcal{B} = \{\mathbf{u} \in \mathbb{R}^d : \|\mathbf{u}\| \leq 1\}$.

## 4.1. One-point Bandit Feedback

We first consider the one-point bandit feedback setting, where each online learner only has access to the loss value at a single point. The key challenge in this setting is the lack of gradients. To overcome this issue, we adopt the one-point gradient estimator (Flaxman et al., 2005), which can approximate the gradient with a single loss value. In particular, given a function $f_{t,i}(\mathbf{x})$, the $\epsilon$-smoothed version of $f_{t,i}(\mathbf{x})$ is defined as

$$\hat{f}_{t,i}(\mathbf{x}) = \mathbb{E}_{\mathbf{u}\sim\mathcal{B}}[f_{t,i}(\mathbf{x} + \epsilon\mathbf{u})],$$

where $\epsilon \in (0, 1)$ is the exploration radius. The function $\hat{f}_{t,i}(\mathbf{x})$ satisfies the following property.

**Lemma 4.2** (Lemma 1 in Flaxman et al. (2005)). *Given a function $f_{t,i}(\mathbf{x})$, its $\epsilon$-smoothed version ensures*

$$\nabla \hat{f}_{t,i}(\mathbf{x}) = \mathbb{E}_{\mathbf{u}\sim\mathcal{S}}\left[\frac{d}{\epsilon}f_{t,i}(\mathbf{x} + \epsilon\mathbf{u})\mathbf{u}\right],$$

*where $\mathbf{u}$ is uniformly sampled from the unit Euclidean sphere $\mathcal{S} = \{\mathbf{u} \in \mathbb{R}^d | \|\mathbf{u}\| = 1\}$.*

Based on Lemma 4.2, we can obtain an unbiased estimator of the gradient $\nabla \hat{f}_{t,i}(\mathbf{x})$ via a single loss value $f_{t,i}(\mathbf{x} + \epsilon\mathbf{u})$.

We integrate the one-point gradient estimator with Top-DOGD to develop an algorithm for the bandit setting, termed Two-level Compressed Decentralized Online Bandit Gradient Descent with One-point Feedback (Top-DOBD-1). Due to space limitations, we do not present the full algorithm in the main text, and more details are provided in Appendix C. Specifically, in each round $t \in [(b-1)L+1, bL]$ within block $b$, each learner $i$ plays the decision $\mathbf{x}_{i,1}(t) = \mathbf{x}_i(b) + \epsilon\mathbf{u}_{t,i}$, where $0 < \epsilon \leq r$, and $\mathbf{u}_{t,i}$ is uniformly sampled from the unit sphere $\mathcal{S}$. Upon receiving the loss value $f_{t,i}(\mathbf{x}_{i,1}(t))$, learner $i$ constructs the gradient estimator according to

$$\hat{\mathbf{g}}_{t,i} = \frac{d}{\epsilon}f_{t,i}(\mathbf{x}_{i,1}(t))\mathbf{u}_{t,i}. \tag{5}$$

Moreover, to ensure the decision $\mathbf{x}_{i,1}(t) \in \mathcal{X}$, each learner needs to project onto the shrunk set

$$\mathcal{X}_\epsilon = (1 - \epsilon/r)\mathcal{X} = \{(1 - \epsilon/r)\mathbf{x} | \forall\mathbf{x} \in \mathcal{X}\}.$$

Formally, we establish the following theoretical guarantees for Top-DOBD-1.

**Theorem 4.1.** *We set $L_1 = \lceil\frac{2\ln(14n)}{\gamma\rho}\rceil, L_2 = \lceil\frac{\ln(8n)}{\omega}\rceil$, $\eta_b = \eta = \frac{R\epsilon}{d\sqrt{LT}}, \gamma = \frac{\omega\rho}{2\rho\beta^2 + 4\beta^2 + (2-\omega)(\beta^2 + 2\beta)\rho + \rho^2}$ and $\epsilon = cd^{1/2}L^{1/4}T^{-1/4}$, where $c$ is a constant such that $\epsilon \leq r$.*

*Under Assumptions 3.1, 3.2, 3.4 and 4.1, for any $i \in [n]$ and any convex loss functions, Top-DOBD-1 ensures*

$$\mathbb{E}_{\mathcal{C}}[R(T,i)] \leq O(\omega^{-1/4}\rho^{-1/2}d^{1/2}n(\ln n)^{1/4}T^{3/4}).$$

**Theorem 4.2.** *We set $L_1 = \lceil \frac{2\ln(14n)}{\gamma\rho} \rceil, L_2 = \lceil \frac{\ln(8n)}{\omega} \rceil$, $\eta_b = \frac{1}{\mu(bL+8)}, \gamma = \frac{\omega\rho}{2\rho\beta^2+4\beta^2+(2-\omega)(\beta^2+2\beta)\rho+\rho^2}$, and $\epsilon = cd^{2/3}L^{1/3}\left(\frac{\ln(T+8)}{T}\right)^{1/3}$, where $c$ is a constant such that $\epsilon \leq r$. Under Assumptions 3.1, 3.3, 3.4 and 4.1, for any $i \in [n]$ and any $\mu$-strongly convex loss functions, Top-DOBD-1 ensures*

$$\mathbb{E}_{\mathcal{C}}[R(T,i)] \leq O(\omega^{-1/3}\rho^{-2/3}d^{2/3}n(\ln n)^{1/3}T^{2/3}(\ln T)^{1/3}).$$

**Remark:** Compared to $O(\max\{\omega^{-1}\rho^{-2}n^{1/4}, \omega^{-2}\rho^{-4}\} d^{1/2}nT^{3/4})$ and $O(\max\{\omega^{-2/3}\rho^{-4/3}n^{1/6}, \omega^{-4/3}\rho^{-8/3}\} d^{2/3}nT^{2/3}(\ln T)^{1/3})$ regret bounds for convex and strongly convex loss functions (Tu et al., 2022), our method achieves tighter dependence on $\omega$, $\rho$, and $n$, which benefits from our two-level blocking update framework.

### 4.2. Two-point Bandit Feedback

In the two-point bandit feedback setting, learner $i$ has access to two loss values $f_{t,i}(\mathbf{x}_{i,1}(t))$ and $f_{t,i}(\mathbf{x}_{i,2}(t))$ in each round, and we redefine the regret as

$$R_2(T,i) = \sum_{t=1}^{T}\sum_{j=1}^{n} \frac{f_{t,j}(\mathbf{x}_{i,1}(t)) + f_{t,j}(\mathbf{x}_{i,2}(t))}{2}$$
$$- \min_{\mathbf{x}\in\mathcal{X}}\sum_{t=1}^{T}\sum_{j=1}^{n} f_{t,j}(\mathbf{x}).$$

Since each learner is able to query the loss values at two points, we employ a more accurate two-point gradient estimator (Agarwal et al., 2010; Shamir, 2017) to develop our method, termed Top-DOBD-2. Compared to Top-DOBD-1, there exist two differences. First, in each round $t$ within block $b$, learner $i$ plays two decisions $\mathbf{x}_{i,1}(t) = \mathbf{x}_i(b) + \epsilon\mathbf{u}_{t,i}$ and $\mathbf{x}_{i,2}(t) = \mathbf{x}_i(b) - \epsilon\mathbf{u}_{t,i}$, where $0 < \epsilon \leq r$, and $\mathbf{u}_{t,i}$ is sampled from $\mathcal{S}$. Second, each learner $i$ constructs the gradient estimator according to

$$\hat{\mathbf{g}}_{t,i} = \frac{d}{2\epsilon}\left(f_{t,i}(\mathbf{x}_{i,1}(t)) - f_{t,i}(\mathbf{x}_{i,2}(t))\right)\mathbf{u}_{t,i}. \quad (6)$$

In the following, we establish the regret bounds for Top-DOBD-2.

**Theorem 4.3.** *We set $L_1 = \lceil \frac{2\ln(14n)}{\gamma\rho} \rceil, L_2 = \lceil \frac{\ln(8n)}{\omega} \rceil$, $\eta_b = \eta = \frac{R}{G\sqrt{dLT}}, \gamma = \frac{\omega\rho}{2\rho\beta^2+4\beta^2+(2-\omega)(\beta^2+2\beta)\rho+\rho^2}$, and $\epsilon = cT^{-1/2}$, where $c$ is a constant such that $\epsilon \leq r$. Under Assumptions 3.1, 3.2, 3.4 and 4.1, for any $i \in [n]$ and any convex loss functions, Top-DOBD-2 ensures*

$$\mathbb{E}_{\mathcal{C}}[R_2(T,i)] \leq O(\omega^{-1/2}\rho^{-1}d^{1/2}n\sqrt{\ln n}\sqrt{T}).$$

**Theorem 4.4.** *We set $L_1 = \lceil \frac{2\ln(14n)}{\gamma\rho} \rceil, L_2 = \lceil \frac{\ln(8n)}{\omega} \rceil$, $\eta_b = \frac{1}{\mu(bL+8)}, \gamma = \frac{\omega\rho}{2\rho\beta^2+4\beta^2+(2-\omega)(\beta^2+2\beta)\rho+\rho^2}$, and $\epsilon = \frac{c\ln T}{T}$, where $c$ is a constant such that $\epsilon \leq r$. Under Assumptions 3.1, 3.3, 3.4 and 4.1, for any $i \in [n]$ and any $\mu$-strongly convex loss functions, Top-DOBD-2 ensures*

$$\mathbb{E}_{\mathcal{C}}[R_2(T,i)] \leq O(\omega^{-1}\rho^{-2}dn\ln n\ln T).$$

**Remark:** The results of our algorithm again achieve tighter dependence on $\omega$, $\rho$ and $n$ compared to existing regret bounds of $O(\max\{\omega^{-2}\rho^{-4}n^{1/2}, \omega^{-4}\rho^{-8}\}dn\sqrt{T})$ and $O(\max\{\omega^{-2}\rho^{-4}n^{1/2}, \omega^{-4}\rho^{-8}\}d^2n\ln T)$ (Tu et al., 2022), indicating that the advantage of our two-level blocking update framework carries over to the two-point bandit setting.

**Remark:** It is worth noting that our algorithm also achieves a better dependence on the dimensionality $d$. This improvement stems from a more refined analysis of the two-point gradient estimator (Shamir, 2017; Lin et al., 2022).

## 5. Conclusion and Future Work

In this paper, we investigate decentralized online convex optimization with compressed communication. First, we introduce a novel method, named Top-DOGD, achieving better regret bounds of $\tilde{O}(\omega^{-1/2}\rho^{-1}n\sqrt{T})$ and $\tilde{O}(\omega^{-1}\rho^{-2}n\ln T)$ for convex and strongly convex loss functions, respectively. In contrast to the previous theoretical guarantees, our algorithm significantly enhances the dependence on the compression ratio $\omega$, the number of learners $n$ and the spectral gap $\rho$. Furthermore, we demonstrate the near-optimality of our upper bounds by establishing the $\Omega(\omega^{-1/2}\rho^{-1/4}n\sqrt{T})$ and $\Omega(\omega^{-1}\rho^{-1/2}n\ln T)$ lower bounds for convex and strongly convex loss functions. Additionally, we extend our Top-DOGD to the one-point and two-point bandit feedback settings by utilizing the classical gradient estimators. Compared to the existing regret bounds, our proposed algorithms again improve the dependence on $\omega$, $n$ and $\rho$ under both the one-point and two-point bandit feedback settings.

Three directions are worth pursuing in future work. First, a gap remains between the regret bounds of Top-DOGD and the lower bounds established in this paper. This gap mainly lies in the dependence on the spectral gap $\rho$ and arises from our use of Choco-gossip. To close it, we would need to design a novel gossip strategy that achieves faster consensus. Second, our lower bounds are stated in terms of the time horizon $T$, and it would be interesting to further derive communication-dependent lower bounds (Wan et al., 2022). Third, in many real-world applications, information about the loss functions may arrive with unknown delays (Cao & Bacsar, 2021; Wan et al., 2024b;c; Yang et al., 2025a;b; Qiu et al., 2026). Thus, it would be worthwhile to investigate D-OCO with delayed feedback.

## Acknowledgements

This work was partially supported by NSFC (62361146852), the "111 Center" (No. B26023), and the Fundamental Research Funds for the Central Universities (2026300271). Sifan Yang was supported by the Nanjing University PhD Student Zhujian Program. The authors would also like to thank the anonymous reviewers for their helpful comments.

## Impact Statement

This paper presents work whose goal is to advance the field of machine learning. There are many potential societal consequences of our work, none of which we feel must be specifically highlighted here.

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

# A. Additional Discussion on Related Work

## A.1. Online Convex Optimization

Online convex optimization (OCO) is a special case of D-OCO with $n = 1$, in which a single learner iteratively selects a decision $\mathbf{x}(t)$ from a convex feasible set and then suffers a loss $f_t(\mathbf{x}(t))$ revealed by an adversary, aiming to minimize the regret against the best fixed decision in hindsight (Zinkevich, 2003; Hazan, 2016), i.e.,

$$R(T) = \sum_{t=1}^{T} f_t(\mathbf{x}(t)) - \min_{\mathbf{x} \in \mathcal{X}} \sum_{t=1}^{T} f_t(\mathbf{x}).$$

For general convex losses, online gradient descent attains the optimal $O(\sqrt{T})$ regret bound (Zinkevich, 2003). When the losses are strongly convex, Hazan et al. (2007) establish the optimal $O(\ln T)$ bound. Building on these results, a rich line of work has since developed algorithms for more challenging scenarios, such as dynamic and adaptive regret in non-stationary environments (Zhang et al., 2018; 2019; Zhang, 2020; Wang et al., 2024).

---

**Algorithm 5** Top-DOBD-1

---

1: **Input:** consensus step size $\gamma$, learning rate $\eta_b$, block size $L = L_1 + L_2$, shrinkage size $\xi$, exploration size $\epsilon$
2: Initialize $\mathbf{x}_i(1) = \mathbf{0}, \hat{\mathbf{x}}_i(1) = \mathbf{0}, \forall i \in [n]$
3: **for** block $b = 1$ to $T/L$ **do**
4:     **if** $b = 1$ **then**
5:         **for** $t = 1$ to $L$ **do**
6:             Play the decision $\mathbf{x}_{i,1}(t) = \mathbf{x}_i(1) + \epsilon \mathbf{u}_{t,i}$ and suffer the loss $f_{t,i}(\mathbf{x}_{i,1}(t))$
7:             Construct the gradient $\hat{\mathbf{g}}_{t,i}$ according to (5)
8:         **end for**
9:     **else**
10:         Set $\mathbf{y}_i^{(1)}(b) = \mathbf{x}_i(b) - \eta_b \mathbf{z}_i(b-1), \hat{\mathbf{y}}_i^{(1)}(b) = \hat{\mathbf{x}}_i(b), b_1 = 1$
11:         **for** $t = (b-1)L + 1$ to $(b-1)L + L_1$ **do**
12:             Play the decision $\mathbf{x}_{i,1}(t) = \mathbf{x}_i(b) + \epsilon \mathbf{u}_{t,i}$ and suffer the loss $f_{t,i}(\mathbf{x}_{i,1}(t))$
13:             Construct the gradient $\hat{\mathbf{g}}_{t,i}$ according to (5)
14:             Transmit $\mathcal{C}(\mathbf{y}_i^{(b_1)}(b) - \hat{\mathbf{y}}_i^{(b_1)}(b))$ to neighbors $j \in \mathcal{N}_i$
15:             Compute $\hat{\mathbf{y}}_j^{(b_1+1)}(b) = \hat{\mathbf{y}}_j^{(b_1)}(b) + \mathcal{C}(\mathbf{y}_j^{(b_1)}(b) - \hat{\mathbf{y}}_j^{(b_1)}(b))$ for $j \in \mathcal{N}_i$
16:             Compute $\mathbf{y}_i^{(b_1+1)}(b) = \mathbf{y}_i^{(b_1)}(b) + \gamma \sum_{j \in \mathcal{N}_i} P_{ij}(\hat{\mathbf{y}}_j^{(b_1+1)}(b) - \hat{\mathbf{y}}_i^{(b_1)}(b)), b_1 = b_1 + 1$
17:         **end for**            ▷ online compressed gossip strategy
18:         Set $\mathbf{r}_i^{(1)}(b+1) = \mathbf{0}, \mathbf{r}_i(b+1) = \Pi_{\mathcal{X}_\epsilon}\left[\mathbf{y}_i^{(L_1+1)}(b)\right] - \mathbf{y}_i^{(L_1+1)}(b), b_2 = 1$
19:         **for** $t = (b-1)L + L_1 + 1$ to $bL$ **do**
20:             Play the decision $\mathbf{x}_{i,1}(t) = \mathbf{x}_i(b) + \epsilon \mathbf{u}_{t,i}$ and suffer the loss $f_{t,i}(\mathbf{x}_{i,1}(t))$
21:             Construct the gradient $\hat{\mathbf{g}}_{t,i}$ according to (5)
22:             Transmit $\Delta_i^{(b_2)}(b) = \mathcal{C}(\mathbf{r}_i(b+1) - \mathbf{r}_i^{(b_2)}(b+1))$ and send $\Delta_i^{(b_2)}(b)$ to $j \in \mathcal{N}_i$
23:             Compute $\mathbf{r}_i^{(b_2+1)}(b+1) = \mathbf{r}_i^{(b_2)}(b+1) + \Delta_i^{(b_2)}(b)$ and set $b_2 = b_2 + 1$
24:         **end for**            ▷ projection error compensation scheme
25:         Update $\hat{\mathbf{x}}_j(b+1) = \hat{\mathbf{y}}_j^{(L_1+1)}(b) + \mathbf{r}_j^{(L_2+1)}(b+1)$ for $j \in \mathcal{N}_i$
26:         Compute $\mathbf{z}_i(b) = \sum_{t=(b-1)L+1}^{bL} \hat{\mathbf{g}}_{t,i}$ and update $\mathbf{x}_i(b+1) = \Pi_{\mathcal{X}_\epsilon}\left[\mathbf{y}_i^{(L_1+1)}(b)\right]$
27:     **end if**
28: **end for**

---

## A.2. Difference Compression and Error Feedback

Since the direct combination of a compressor and standard gossip fails to converge to the exact solution, Tang et al. (2018) propose difference compression (DC). DC adds replicas of neighboring states of each learner and transmits the compressed state-difference information. Later, Koloskova et al. (2019) propose Choco-gossip and provide the analysis for the general compressor under the offline optimization.

---

**Algorithm 6** Top-DOBD-2

---

1: **Input:** consensus step size $\gamma$, learning rate $\eta_b$, block size $L = L_1 + L_2$, shrinkage size $\xi$, exploration size $\epsilon$
2: Initialize $\mathbf{x}_i(1) = \mathbf{0}, \hat{\mathbf{x}}_i(1) = \mathbf{0}, \forall i \in [n]$
3: **for** block $b = 1$ to $T/L$ **do**
4:     **if** $b = 1$ **then**
5:         **for** $t = 1$ to $L$ **do**
6:             Play the decisions $\mathbf{x}_{i,1}(t) = \mathbf{x}_i(1) + \epsilon \mathbf{u}_{t,i}$ and $\mathbf{x}_{i,2}(t) = \mathbf{x}_i(1) - \epsilon \mathbf{u}_{t,i}$
7:             Suffer the loss $f_{t,i}(\mathbf{x}_{i,1}(t))$ and $f_{t,i}(\mathbf{x}_{i,2}(t))$
8:             Construct the gradient $\hat{\mathbf{g}}_{t,i}$ according to (6)
9:         **end for**
10:     **else**
11:         Set $\mathbf{y}_i^{(1)}(b) = \mathbf{x}_i(b) - \eta_b \mathbf{z}_i(b-1), \hat{\mathbf{y}}_i^{(1)}(b) = \hat{\mathbf{x}}_i(b), b_1 = 1$
12:         **for** $t = (b-1)L + 1$ to $(b-1)L + L_1$ **do**
13:             Play the decisions $\mathbf{x}_{i,1}(t) = \mathbf{x}_i(b) + \epsilon \mathbf{u}_{t,i}$ and $\mathbf{x}_{i,2}(t) = \mathbf{x}_i(b) - \epsilon \mathbf{u}_{t,i}$
14:             Suffer the loss $f_{t,i}(\mathbf{x}_{i,1}(t))$ and $f_{t,i}(\mathbf{x}_{i,2}(t))$
15:             Construct the gradient $\hat{\mathbf{g}}_{t,i}$ according to (6)
16:             Transmit $\mathcal{C}(\mathbf{y}_i^{(b_1)}(b) - \hat{\mathbf{y}}_i^{(b_1)}(b))$ to neighbors $j \in \mathcal{N}_i$
17:             Compute $\hat{\mathbf{y}}_j^{(b_1+1)}(b) = \hat{\mathbf{y}}_j^{(b_1)}(b) + \mathcal{C}(\mathbf{y}_j^{(b_1)}(b) - \hat{\mathbf{y}}_j^{(b_1)}(b))$ for $j \in \mathcal{N}_i$
18:             Compute $\mathbf{y}_i^{(b_1+1)}(b) = \mathbf{y}_i^{(b_1)}(b) + \gamma \sum_{j \in \mathcal{N}_i} P_{ij}(\hat{\mathbf{y}}_j^{(b_1+1)}(b) - \hat{\mathbf{y}}_i^{(b_1)}(b)), b_1 = b_1 + 1$
19:         **end for**                          ▷ online compressed gossip strategy
20:         Set $\mathbf{r}_i^{(1)}(b+1) = \mathbf{0}, \mathbf{r}_i(b+1) = \Pi_{\mathcal{X}_\epsilon}\left[\mathbf{y}_i^{(L_1+1)}(b)\right] - \mathbf{y}_i^{(L_1+1)}(b), b_2 = 1$
21:         **for** $t = (b-1)L + L_1 + 1$ to $bL$ **do**
22:             Play the decisions $\mathbf{x}_{i,1}(t) = \mathbf{x}_i(b) + \epsilon \mathbf{u}_{t,i}$ and $\mathbf{x}_{i,2}(t) = \mathbf{x}_i(b) - \epsilon \mathbf{u}_{t,i}$
23:             Suffer the loss $f_{t,i}(\mathbf{x}_{i,1}(t))$ and $f_{t,i}(\mathbf{x}_{i,2}(t))$
24:             Construct the gradient $\hat{\mathbf{g}}_{t,i}$ according to (6)
25:             Transmit $\Delta_i^{(b_2)}(b) = \mathcal{C}(\mathbf{r}_i(b+1) - \mathbf{r}_i^{(b_2)}(b+1))$ and send $\Delta_i^{(b_2)}(b)$ to $j \in \mathcal{N}_i$
26:             Compute $\mathbf{r}_i^{(b_2+1)}(b+1) = \mathbf{r}_i^{(b_2)}(b+1) + \Delta_i^{(b_2)}(b)$ and set $b_2 = b_2 + 1$
27:         **end for**                          ▷ projection error compensation scheme
28:         Update $\hat{\mathbf{x}}_j(b+1) = \hat{\mathbf{y}}_j^{(L_1+1)}(b) + \mathbf{r}_j^{(L_2+1)}(b+1)$ for $j \in \mathcal{N}_i$
29:         Compute $\mathbf{z}_i(b) = \sum_{t=(b-1)L+1}^{bL} \hat{\mathbf{g}}_{t,i}$ and update $\mathbf{x}_i(b+1) = \Pi_{\mathcal{X}_\epsilon}\left[\mathbf{y}_i^{(L_1+1)}(b)\right]$
30:     **end if**
31: **end for**

---

Error feedback (EF) (Seide et al., 2014; Karimireddy et al., 2019; Richtárik et al., 2021) is another compression scheme, aiming to correct errors introduced by the compressor. Specifically, DC focuses on the discrepancy between the current decision and its replica, which is widely used in decentralized optimization because the exchanged state variables typically converge to a nonzero limit. In contrast, EF compresses the sum of the local gradient and an accumulated residual error, popular in federated learning problems where the exchanged gradient information is expected to vanish asymptotically.

## B. Examples of Compressors

Here, we present some examples of compressors.

- *Sparsification.* Randomly selecting $k$ out of $d$ coordinates (Rand-$k$), or selecting the $k$ coordinates with the largest absolute values (Top-$k$), both yield compressors with a compression ratio of $\omega = \frac{k}{d}$.

- *Randomized gossip.* Outputting $\mathcal{C}(\mathbf{x}) = \mathbf{x}$ with probability $p \in (0, 1]$ and $\mathcal{C}(\mathbf{x}) = 0$ otherwise leads to a compression ratio of $\omega = p$.

- *Rescaled unbiased estimators.* Suppose $E_{\mathcal{C}}[\mathbf{x}] = \mathbf{x}, \mathbb{E}_{\mathcal{C}}\left[\|\mathcal{C}(\mathbf{x})\|^2\right] \leq \tau \|\mathbf{x}\|^2$, then $\mathcal{C}'(\mathbf{x}) = \frac{1}{\tau}\mathcal{C}(\mathbf{x})$ is a compressor with $\omega = \frac{1}{\tau}$.

## C. Extension to Bandit Feedback Setting

In this section, we summarize our algorithms for the bandit feedback settings. Top-DOBD-1 for the one-point bandit feedback setting is presented in Algorithm 5, and Top-DOBD-2 for the two-point bandit feedback setting is shown in Algorithm 6.

## D. Proof of Theorems

### D.1. Proof of Theorem 3.1

**Notation.** Let $n$ be the total number of learners, $d$ be the dimensionality, $L$ be the block size, $\omega$ be the compression ratio. In the proof, we use bold lower-case letters (e.g., $\mathbf{x}$) to denote vectors in $\mathbb{R}^d$, $\mathbf{1} \in \mathbb{R}^d$ to denote $[1, \ldots, 1]^\top$ and $\|\cdot\|$ to represent $\ell_2$-norm by default. We assume $T/L$ to be an integer without loss of generality.

We first give some definitions

$$\tilde{\mathbf{x}}_i(b+1) = \mathbf{y}_i^{(L_1+1)}(b),$$

$$\mathbf{r}_i(b+1) = \Pi_{\mathcal{X}}[\mathbf{y}_i^{(L_1+1)}(b)] - \mathbf{y}_i^{(L_1+1)}(b) = \mathbf{x}_i(b+1) - \tilde{\mathbf{x}}_i(b+1),$$

$$\overline{\mathbf{x}}(b) = \frac{1}{n}\sum_{i=1}^{n}\mathbf{x}_i(b),$$

$$\sum_{i=1}^{n}\sum_{j\in\mathcal{N}_i}P_{ij}(\hat{\mathbf{y}}_j^{(b_1+1)}(b) - \hat{\mathbf{y}}_i^{(b_1+1)}(b)) = 0,$$

$$\mathbf{r}_i^{\mathcal{C}}(b+1) = \mathbf{r}_i^{(L_2+1)}(b+1),$$

The fourth equality is due to the variable $\hat{\mathbf{y}}_i^{(b_1)}(b)$ being the same over all $j \in \mathcal{N}_i$. Then we define

$$X(b) = [\mathbf{x}_1(b), ..., \mathbf{x}_n(b)] \in \mathbb{R}^{d\times n}, \tilde{X}(b) = [\tilde{\mathbf{x}}_1(b), ..., \tilde{\mathbf{x}}_n(b)] \in \mathbb{R}^{d\times n},$$

$$\overline{X}(b) = [\overline{\mathbf{x}}(b), ..., \overline{\mathbf{x}}(b)] \in \mathbb{R}^{d\times n}, R(b) = [\mathbf{r}_1(b), ..., \mathbf{r}_n(b)] \in \mathbb{R}^{d\times n},$$

$$R^{\mathcal{C}}(b) = [\mathbf{r}_1^{\mathcal{C}}(b), ..., \mathbf{r}_n^{\mathcal{C}}(b)] \in \mathbb{R}^{d\times n}, Z(b) = [\mathbf{z}_1(b), ..., \mathbf{z}_n(b)] \in \mathbb{R}^{d\times n}$$

$$Y^{(b_1)}(b) = [\mathbf{y}_1^{(b_1)}(b), ..., \mathbf{y}_n^{(b_1)}(b)] \in \mathbb{R}^{d\times n}, \hat{Y}^{(b_1)}(b) = [\hat{\mathbf{y}}_1^{(b_1)}(b), ..., \hat{\mathbf{y}}_n^{(b_1)}(b)] \in \mathbb{R}^{d\times n}.$$

By using our definitions, we obtain the following equivalent update rules

$$\hat{X}(b+1) = \hat{Y}^{(L_1+1)}(b) + R^{\mathcal{C}}(b+1),$$

$$X(b+1) = \tilde{X}(b+1) + R(b+1) = Y^{(L_1+1)}(b+1) + R(b+1).$$

We next recall two basic projection inequalities:

$$\|P_{\mathcal{X}}(\mathbf{x}) - P_{\mathcal{X}}(\mathbf{y})\| \leq \|\mathbf{x} - \mathbf{y}\|, \text{ for } \forall \mathbf{x}, \mathbf{y} \in \mathbb{R}^d, \tag{7}$$

$$\langle P_{\mathcal{X}}(\mathbf{x}) - \mathbf{x}, \mathbf{x} - \mathbf{y}\rangle \leq -\|P_{\mathcal{X}}(\mathbf{x}) - \mathbf{x}\|^2 \leq 0, \text{ for } \forall \mathbf{x} \in \mathbb{R}^d, \forall \mathbf{y} \in \mathcal{X}. \tag{8}$$

We first present a lemma that characterizes the regret of learner $i$.

**Lemma D.1.** *Under Assumption 3.1, 3.2, 3.4, 3.5, the regret of learner $i$ for Algorithm 2 is*

$$
\mathbb{E}_{\mathcal{C}}\left[R(T,i)\right] = \sum_{b=1}^{T/L}\sum_{t=(b-1)L+1}^{bL}\sum_{j=1}^{n} f_{t,j}(\mathbf{x}_i(b)) - \sum_{t=1}^{T}\sum_{j=1}^{n} f_{t,j}(\mathbf{x})
$$

$$
\leq \frac{nD^2}{2\eta_{T/L}} + 3L^2G^2 n\sum_{b=1}^{T/L}\eta_b + \sum_{b=1}^{T/L}\frac{3}{2\eta_b}\underbrace{\mathbb{E}_{\mathcal{C}}\left[\|R(b+1)\|_F^2\right]}_{\text{TERM}_A} + \frac{1}{2\eta_b}\underbrace{\mathbb{E}_{\mathcal{C}}\left[\left\|X(b)-\tilde{X}(b+1)\right\|_F^2\right]}_{\text{TERM}_B}
$$

$$
+ 3nGL\sum_{b=1}^{T/L}\underbrace{\mathbb{E}_{\mathcal{C}}\left[\left\|X(b)-\overline{X}(b)\right\|_F\right]}_{\text{TERM}_C} + \sum_{b=1}^{T/L}\frac{1}{2\eta_b}\underbrace{\mathbb{E}_{\mathcal{C}}\left[\|R(b)\|_F^2\right]}_{\text{TERM}_A}.
$$

To derive the regret bound of each learner, the core challenge is to give the bound of $\text{TERM}_A$, $\text{TERM}_B$ and $\text{TERM}_C$. $\text{TERM}_A$ is the projection error, $\text{TERM}_B$ is the compression error and $\text{TERM}_C$ is the consensus error. Next, to give the bound of each term, we present the following lemma.

**Lemma D.2.** *Under Assumptions 3.1, 3.2, 3.4 and 3.5, by selecting $L_1 = \lceil\frac{2\ln(14n)}{\gamma\rho}\rceil, L_2 = \lceil\frac{\ln(8n)}{\omega}\rceil, \gamma = \frac{\omega\rho}{2\rho\beta^2+4\beta^2+(2-\omega)(\beta^2+2\beta)\rho+\rho^2}$ we have the following guarantees.*

$$
\mathbb{E}_{\mathcal{C}}\left[\|R(b+1)\|_F^2\right] \leq \frac{2}{7n}\mathbb{E}_{\mathcal{C}}\left[\left\|X(b)-\hat{X}(b)\right\|_F^2 + \left\|X(b)-\overline{X}(b)\right\|_F^2\right] + (2n+\frac{10}{7})L^2G^2\eta_b^2,
$$

$$
\mathbb{E}_{\mathcal{C}}\left[\left\|X(b+1)-\overline{X}(b+1)\right\|_F^2\right] \leq \frac{1}{7n}\mathbb{E}_{\mathcal{C}}\left[\left\|X(b)-\hat{X}(b)\right\|_F^2 + \left\|X(b)-\overline{X}(b)\right\|_F^2\right] + \frac{5}{7}L^2G^2\eta_b^2
$$

$$
\mathbb{E}_{\mathcal{C}}\left[\left\|X(b+1)-\hat{X}(b+1)\right\|_F^2\right] \leq \frac{5}{14n}\mathbb{E}_{\mathcal{C}}\left[\left\|X(b)-\hat{X}(b)\right\|_F^2 + \left\|X(b)-\overline{X}(b)\right\|_F^2\right] + 2L^2G^2\eta_b^2
$$

We define the error $e_{b+1}$ as follows

$$
e_{b+1} = \mathbb{E}_{\mathcal{C}}\left[\sum_{i=1}^{n}\|\mathbf{x}_i(b+1)-\overline{\mathbf{x}}(b+1)\|^2 + \|\mathbf{x}_i(b+1)-\hat{\mathbf{x}}_i(b+1)\|^2\right]
$$

$$
= \mathbb{E}_{\mathcal{C}}\left[\|X(b+1)-\overline{X}(b+1)\|_F^2\right] + \mathbb{E}_{\mathcal{C}}\left[\|X(b+1)-\hat{X}(b+1)\|_F^2\right].
$$

By fixing the learning rate $\eta_b = \eta$, we have the following guarantee

$$
e_{b+1} \leq \frac{1}{2n}e_b + 3\eta^2 L^2 G^2. \tag{9}
$$

By summing up, we have

$$
e_{b+1} \leq \frac{1}{1-\frac{1}{2n}}3\eta^2 L^2 G^2 \leq 6\eta^2 L^2 G^2,
$$

which is due to $\sum_{i=1}^{b}\frac{1}{(2n)^i} \leq \frac{1}{1-\frac{1}{2n}} \leq 2$ and $e_1 = 0$.

As for the term $\mathbb{E}_{\mathcal{C}}\left[\|R(b+1)\|_F^2\right]$, we have

$$
\mathbb{E}_{\mathcal{C}}\left[\|R(b+1)\|_F^2\right] \leq \frac{2}{7n}e_b + (2n+\frac{10}{7})L^2G^2\eta^2
$$

$$
\leq \frac{1}{1-\frac{2}{7n}}(2n+\frac{10}{7})L^2G^2\eta^2
$$

$$
\leq (\frac{14n}{7n-2}n + \frac{10n}{7n-2})L^2G^2\eta^2
$$

$$
\leq (3n+2)L^2G^2\eta^2,
$$

which is the same as $\mathbb{E}_{\mathcal{C}}\left[\|R(b)\|_F^2\right]$. Now, we can derive the regret bound of Top-DOGD.

First, we have

$$E_{\mathcal{C}}\left[\left\|X(b) - \overline{X}(b)\right\|_F\right] \leq \sqrt{e_b} \leq \sqrt{6}\eta LG \leq 3\eta LG.$$

As for the second term, we have

$$
\begin{aligned}
&\mathbb{E}_{\mathcal{C}}\left[\left\|X(b) - \tilde{X}(b+1)\right\|_F^2\right] \\
=&\mathbb{E}_{\mathcal{C}}\left[\sum_{i=1}^n \|\mathbf{x}_i(b) - \tilde{\mathbf{x}}_i(b+1)\|^2\right] \\
=&\mathbb{E}_{\mathcal{C}}\left[\sum_{i=1}^n \left\|\mathbf{x}_i(b) - \overline{\mathbf{y}}_i^{(L_1+1)}(b) + \overline{\mathbf{y}}_i^{(L_1+1)}(b) - \mathbf{y}_i^{(L_1+1)}(b)\right\|^2\right] \\
=&\mathbb{E}_{\mathcal{C}}\left[\sum_{i=1}^n \left\|\mathbf{x}_i(b) - \overline{\mathbf{x}}(b) + \overline{\mathbf{y}}_i^{(L_1+1)}(b) - \mathbf{y}_i^{(L_1+1)}(b)\right\|^2\right] \\
\leq&2\mathbb{E}_{\mathcal{C}}\left[\sum_{i=1}^n \|\overline{\mathbf{x}}(b) - \mathbf{x}_i(b)\|^2\right] + 2\mathbb{E}_{\mathcal{C}}\left[\sum_{i=1}^n \left\|\frac{1}{n}\sum_{i=1}^n \tilde{\mathbf{x}}_i(b+1) - \tilde{\mathbf{x}}_i(b+1)\right\|^2\right] \\
\leq&2e_b + 2e_{b+1},
\end{aligned}
\tag{10}
$$

where the third equality is due to $\overline{\mathbf{y}}^{(L_1+1)}(b) = \frac{1}{n}\sum_{i=1}^n \mathbf{y}_i^{(L_1+1)}(b) = \frac{1}{n}\sum_{i=1}^n \mathbf{y}_i^{(1)}(b) = \frac{1}{n}\sum_{i=1}^n \mathbf{x}_i(b) = \overline{\mathbf{x}}(b)$ and the last inequality is due to (30).

Therefore, we have

$$\mathbb{E}_{\mathcal{C}}\left[\left\|X(b) - \tilde{X}(b+1)\right\|_F^2\right] \leq 2e_b + 2e_{b+1} = 24\eta^2 L^2 G^2,$$

Finally, by setting $\eta_b = \eta = \frac{D}{G\sqrt{LT}}, L = L_1 + L_2 = O(\omega^{-1}\rho^{-2}\ln n)$, we have

$$
\begin{aligned}
\mathbb{E}_{\mathcal{C}}\left[R(T,i)\right] =&\mathbb{E}_{\mathcal{C}}\left[\sum_{b=1}^{T/L}\sum_{t=(b-1)L+1}^{bL}\sum_{j=1}^n f_{t,j}(\mathbf{x}_i(b)) - \sum_{t=1}^T\sum_{j=1}^n f_{t,j}(\mathbf{x})\right] \\
\leq&\frac{nD^2}{2\eta_{T/L}} + 3L^2G^2n\sum_{b=1}^{T/L}\eta_b + \sum_{b=1}^{T/L}\frac{3}{2\eta_b}\mathbb{E}_{\mathcal{C}}\left[\|R(b+1)\|_F^2\right] + \frac{1}{2\eta_b}\mathbb{E}_{\mathcal{C}}\left[\left\|X(b) - \tilde{X}(b+1)\right\|_F^2\right] \\
&+ 3nGL\sum_{b=1}^{T/L}\mathbb{E}_{\mathcal{C}}\left[\left\|X(b) - \overline{X}(b)\right\|_F\right] + \sum_{b=1}^{T/L}\frac{1}{2\eta_b}\mathbb{E}_{\mathcal{C}}\left[\|R(b)\|_F^2\right] \\
\leq&\frac{nD^2}{2\eta} + 3nLG^2\eta T + \sum_{b=1}^{T/L}\frac{3}{2\eta}\mathbb{E}_{\mathcal{C}}\left[\|R(b+1)\|_F^2\right] + \frac{1}{2\eta}\mathbb{E}_{\mathcal{C}}\left[\left\|\overline{X}(b) - \tilde{X}(b+1)\right\|_F^2\right] \\
&+ 3nGL\sum_{b=1}^{T/L}\mathbb{E}_{\mathcal{C}}\left[\left\|X(b) - \overline{X}(b)\right\|_F\right] + \sum_{b=1}^{T/L}\frac{1}{2\eta}\mathbb{E}_{\mathcal{C}}\left[\|R(b)\|_F^2\right] \\
\leq&\frac{nD^2}{2\eta} + 3nLG^2T\eta + (5n+3)LG^2T\eta + 12LG^2T\eta + 9n\eta TLG^2 + (2n+1)LG^2T\eta \\
\leq&O(n\sqrt{LT}) = O(\omega^{-1/2}\rho^{-1}n\sqrt{\ln n}\sqrt{T}).
\end{aligned}
$$

### D.2. Proof of Theorem 3.2

The proof follows a similar structure to Theorem 3.1, except that we exploit strong convexity to establish improved regret bounds.

**Lemma D.3.** *Under Assumptions 3.1, 3.3, 3.4, 3.5, the regret of learner $i$ for Algorithm 2 is*

$$
\mathbb{E}_{\mathcal{C}}\left[R(T,i)\right] = \sum_{b=1}^{T/L}\sum_{t=(b-1)L+1}^{bL}\sum_{j=1}^{n} f_{t,j}(\mathbf{x}_i(b)) - \sum_{t=1}^{T}\sum_{j=1}^{n} f_{t,j}(\mathbf{x})
$$

$$
\leq \frac{nD^2}{2}\sum_{b=1}^{T/L}(\frac{1}{\eta_b} - \frac{1}{\eta_{b-1}} - \mu L) + 3L^2 G^2 n \sum_{b=1}^{T/L}\eta_b + 3nGL\sum_{b=1}^{T/L}\mathbb{E}_{\mathcal{C}}\left[\left\|X(b) - \overline{X}(b)\right\|_F\right]
$$

$$
+ \sum_{b=1}^{T/L}\frac{3}{2\eta_b}\mathbb{E}_{\mathcal{C}}\left[\|R(b+1)\|_F^2\right] + \frac{1}{2\eta_b}\mathbb{E}_{\mathcal{C}}\left[\left\|X(b) - \tilde{X}(b+1)\right\|_F^2\right] + \frac{1}{2\eta_b}\mathbb{E}_{\mathcal{C}}\left[\|R(b)\|_F^2\right].
$$

According to Lemma D.2, we have the following

$$
e_{b+1} \leq \frac{1}{2n}e_b + 3\eta_b^2 L^2 G^2. \tag{11}
$$

To establish the bound of $e_{b+1}$, we introduce the following lemma.

**Lemma D.4.** *Let $\{e_b\}_{b\geq 1}$ denote a sequence of values satisfying $e_1 = 0$ and*

$$
e_{b+1} \leq \frac{1}{2n}e_b + q\eta_b^2 L^2,
$$

*where $q > 0$, $\eta_b = \frac{1}{\mu(bL+8)}$. We have the following guarantee*

$$
e_b \leq 4qL^2\eta_b^2.
$$

Therefore, by setting $\eta_b = \frac{1}{\mu(bL+8)}$, we have

$$
e_b \leq 12L^2 G^2 \eta_b^2.
$$

Then we give the bound of the terms in the regret individually

$$
E_{\mathcal{C}}\left[\left\|X(b) - \overline{X}(b)\right\|_F\right] \leq \sqrt{e_b} \leq 2\sqrt{3}\eta_b LG.
$$

As for the second term, we have

$$
\mathbb{E}_{\mathcal{C}}\left[\left\|X(b) - \tilde{X}(b+1)\right\|_F^2\right]
$$

$$
= \mathbb{E}_{\mathcal{C}}\left[\sum_{i=1}^{n}\|\mathbf{x}_i(b) - \tilde{\mathbf{x}}_i(b+1)\|^2\right] \tag{12}
$$

$$
\leq 2e_b + 2e_{b+1} \leq 48L^2 G^2 \eta_b^2,
$$

where the last inequality is due to $\eta_b \geq \eta_{b+1}$. For $\mathbb{E}_{\mathcal{C}}\left[\frac{1}{\eta_b}\|R(b+1)\|_F^2\right]$, we have the following.

$$
\frac{1}{\eta_b}\mathbb{E}_{\mathcal{C}}\left[\|R(b+1)\|_F^2\right] \leq \frac{1}{\eta_b}\left(\frac{2}{7n}e_b + (2n + \frac{10}{7})L^2 G^2 \eta_b^2\right)
$$

$$
\leq \frac{1}{\eta_b}\left((8n + \frac{40}{7})L^2 G^2 \eta_b^2\right) \leq (8n + 6)L^2 G^2 \eta_b.
$$

For $\mathbb{E}_{\mathcal{C}}\left[\frac{1}{\eta_b}\|R(b)\|_F^2\right]$, we have the following.

$$
\frac{1}{\eta_b}\mathbb{E}_{\mathcal{C}}\left[\|R(b)\|_F^2\right] \leq \frac{1}{\eta_b}\left(\frac{2}{7n}e_{b-1} + 4nL^2 G^2 \eta_{b-1}^2\right)
$$

$$
\leq (8n + 6)L^2 G^2 \frac{\eta_{b-1}^2}{\eta_b} \leq (16n + 12)L^2 G^2 \eta_{b-1},
$$

where the last inequality is due to $\frac{\eta_{b-1}}{\eta_b} \leq 2$.

Therefore, we can derive the regret bound of Algorithm 2 for strongly convex functions. By setting $\eta_b = \frac{1}{\mu(bL+8)}$, we have

$$
\begin{aligned}
\mathbb{E}_{\mathcal{C}}\left[R(T,i)\right] &= \sum_{b=1}^{T/L} \sum_{t=(b-1)L+1}^{bL} \sum_{j=1}^{n} f_{t,j}(\mathbf{x}_i(b)) - \sum_{t=1}^{T} \sum_{j=1}^{n} f_{t,j}(\mathbf{x}) \\
&\leq \frac{nD^2}{2} \sum_{b=1}^{T/L} \left(\frac{1}{\eta_b} - \frac{1}{\eta_{b-1}} - \mu L\right) + 3L^2 G^2 n \sum_{b=1}^{T/L} \eta_b + 3nGL \sum_{b=1}^{T/L} \mathbb{E}_{\mathcal{C}}\left[\left\|X(b) - \overline{X}(b)\right\|_F\right] \\
&\quad + \sum_{b=1}^{T/L} \frac{3}{2\eta_b} \mathbb{E}_{\mathcal{C}}\left[\left\|R(b+1)\right\|_F^2\right] + \frac{1}{2\eta_b} \mathbb{E}_{\mathcal{C}}\left[\left\|X(b) - \tilde{X}(b+1)\right\|_F^2\right] + \frac{1}{2\eta_b} \mathbb{E}_{\mathcal{C}}\left[\left\|R(b)\right\|_F^2\right] \\
&\leq \frac{nD^2}{2}\left(\frac{1}{\eta_1} - \mu L\right) + 3L^2 G^2 n \sum_{b=1}^{T/L} \eta_b + 6\sqrt{3} n L^2 G^2 \sum_{b=1}^{T/L} \eta_b \\
&\quad + (12n+9)L^2 G^2 \sum_{b=1}^{T/L} \eta_b + 24L^2 G^2 \sum_{b=1}^{T/L} \eta_b + (8n+6)L^2 G^2 \sum_{b=1}^{T/L} \eta_{b-1} \\
&\leq 4nD^2\mu + \frac{1}{\mu}\Big(3LG^2 n \ln(T+8) + 6\sqrt{3} n G^2 L \ln(T+8) \\
&\quad + (20n+15)G^2 L \ln(T+8) + 24G^2 L \ln(T+8)\Big) \\
&\leq O(Ln\ln(T)) = O(\omega^{-1}\rho^{-2} n \ln n \ln T).
\end{aligned}
$$

where the last inequality is due to $\sum_{b=1}^{T/L} \frac{L}{\mu(bL+8)} \leq \frac{1}{\mu} \sum_{b=1}^{T/L} \sum_{t=(b-1)L+1}^{bL} \frac{1}{t+8} \leq \frac{1}{\mu} \int_0^T \frac{1}{t+8} dt \leq \frac{1}{\mu} \ln(T+8) = O(\ln T)$.

### D.3. Additional Discussion

In the following, we discuss the necessity of our two proposed strategies.

(i) $L_1 = 1$. It is not hard to verify that, when $L_1 = 1$, the sum of the consensus error and the compression error is on the same order as Tu et al. (2022). Although we can reduce the projection error to $O(1)$, the consensus error is still the same as previous work, which is the leading term in the final regret. Thus, it does not help to improve the existing regret bounds.

(ii) $L_2 = 0$. When $L_2 = 0$, the upper bound of term $\|\mathbf{x}_i(b) - \hat{\mathbf{x}}_i(b)\|^2$ contains an additional projection error $\|\mathbf{r}_i(b)\|^2$ of the order $O(1)$, which consequently induces an $O(n)$ dependence on $e_{b+1}$, that is

$$
e_{b+1} \leq \frac{1}{2n} e_b + O(n\eta^2 L^2 G^2) \leq O(n\eta^2 L^2 G^2).
$$

It is not hard to verify that we can only obtain $O(\omega^{-1/2}\rho^{-1} n^{5/4} \sqrt{\ln n} \sqrt{T})$ and $O(\omega^{-1}\rho^{-2} n^{3/2} \ln n \ln T)$ regret bounds for convex and strongly convex loss functions.

### D.4. Proof of Theorem 3.3

The structure of our proof follows that of Wan et al. (2024a), with the main distinction being that we incorporate a dedicated compressor to derive the lower bound. Let $A \in \mathbb{R}^{n \times n}$ denote the adjacency matrix of $\mathcal{G}$, and let $\delta_i = |N_i| - 1$ denote the degree of node $i$. As presented in Duchi et al. (2011), for any connected undirected graph, there exists a specific gossip matrix $P$ satisfying Assumption 3.1, i.e.,

$$
P = I_n - \frac{1}{\delta_{\max} + 1}(D - A), \tag{13}
$$

where $\delta_{\max} = \max\{\delta_1, ..., \delta_n\}$ and $D = \text{diag}\{\delta_1, ..., \delta_n\}$.

To maximize the impact of communication on the regret bound, we focus on the 1-connected cycle graph, where the graph $\mathcal{G}$ is constructed by arranging $n$ nodes on a circle and connecting each node with its immediate left and right neighbors.

We adopt the randomized gossip compressor $\mathcal{C}(\cdot)$, which outputs $\mathcal{C}(\mathbf{x}) = \mathbf{x}$ with probability $\omega \in (0, 1]$ and $\mathcal{C}(\mathbf{x}) = \mathbf{0}_d$ otherwise. Under this scheme, two connected learners $i$ and $j$ can successfully exchange data only with probability $\omega$ in each round. Consequently, the expected number of rounds required for a successful exchange is $1/\omega$.

To derive the lower bound, we attempt to maximize the regret of learner 1. Specifically, we set the loss functions as

$$f_{t,\, n-\lceil m/2 \rceil +2}(\mathbf{x}) = \cdots = f_{t,n}(\mathbf{x}) = f_{t,1}(\mathbf{x}) = f_{t,2}(\mathbf{x}) = \cdots = f_{t,\lceil m/2 \rceil}(\mathbf{x}) = 0,$$

while the other loss functions are set carefully to construct the desired lower bound. It is straightforward to see that when $\omega = 1$, learner 1 must go through $\lceil m/2 \rceil$ rounds of communication to receive information from learners $\lceil m/2 \rceil + 1, \ldots, n - \lceil m/2 \rceil + 1$. Let $K_1$ denote the communication rounds. When $\omega < 1$, the expected number of communication rounds becomes

$$\mathbb{E}_{\mathcal{C}}\left[K_1\right] = \left\lceil \frac{m}{2\omega} \right\rceil. \tag{14}$$

We give the proof below. For two learners $i$ and $i+1$, the expected number of rounds $a_{i,i+1}$ for a successful transmission is

$$\mathbb{E}[a_{i,i+1}] = \sum_{k=1}^{\infty} k(1-\omega)^{k-1}\omega = \omega \sum_{k=1}^{\infty} k(1-\omega)^{k-1} = \frac{1}{\omega},$$

where the second equality is due to $\sum_{k=1}^{\infty} kq^{k-1} = \frac{1}{(1-q)^2}$ for $|q| < 1$. Therefore, we have

$$\mathbb{E}\left[K_1\right] = \mathbb{E}\left[ \sum_{1 \leq i \leq \lceil m/2 \rceil} a_{i,i+1} \right] = \sum_{1 \leq i \leq \lceil m/2 \rceil} \mathbb{E}\left[a_{i,i+1}\right] = \left\lceil \frac{m}{2\omega} \right\rceil.$$

Let $K = \lceil m/2 \rceil, Z = \lfloor (T-1)/K_1 \rfloor, c_0 = 0$ and $c_{Z+1} = T$. The total $T$ rounds can be divided into the following $Z + 1$ intervals

$$[c_0 + 1,\, c_1],\, [c_1 + 1,\, c_2],\, \ldots,\, [c_Z + 1,\, c_{Z+1}].$$

Following Wan et al. (2025), for any $i \in \{0, 1, ..., Z\}$ and $t \in [c_i + 1,\, c_{i+1}]$, we set

$$f_{t,\, \lceil m/2 \rceil + 1}(\mathbf{x}) = \cdots = f_{t,\, n-\lceil m/2 \rceil + 1}(\mathbf{x}) = h_i(\mathbf{x}) = \langle \mathbf{w}_i, \mathbf{x} \rangle,$$

where the coordinates of $\mathbf{w}_i$ are $\pm G/\sqrt{d}$ with probability $0.5$ and the feasible domain $\mathcal{X} = [-D/2\sqrt{d}, D/2\sqrt{d}]^d$. Then the global loss function is

$$f_t(\mathbf{x}) = (n - 2K + 1)h_i(\mathbf{x}).$$

Moreover, it is obvious that the decisions $\mathbf{x}_1(c_i + 1), \ldots, \mathbf{x}_1(c_{i+1})$ for any $i \in \{0, \ldots, Z\}$ are made before the learner 1 has access to $h_i(\mathbf{x})$. Then we can derive the expected lower bound for $R(T, 1)$.

$$
\begin{aligned}
\mathbb{E}_{\mathbf{w}_0,\ldots,\mathbf{w}_Z}[R(T,1)] &= \mathbb{E}_{\mathbf{w}_0,\ldots,\mathbf{w}_Z}\left[ \sum_{i=0}^{Z} \sum_{t=c_i+1}^{c_{i+1}} (n-2K+1)h_i(\mathbf{x}_1(t)) - \min_{\mathbf{x}\in\mathcal{X}} \sum_{i=0}^{Z} \sum_{t=c_i+1}^{c_{i+1}} (n-2K+1)h_i(\mathbf{x}) \right] \\
&= (n-2K+1)\mathbb{E}_{\mathbf{w}_0,\ldots,\mathbf{w}_Z}\left[ \sum_{i=0}^{Z} \sum_{t=c_i+1}^{c_{i+1}} \langle \mathbf{w}_i, \mathbf{x}_1(t) \rangle - \min_{\mathbf{x}\in\mathcal{X}} \sum_{i=0}^{Z} (c_{i+1}-c_i)\langle \mathbf{w}_i, \mathbf{x} \rangle \right] \\
&= -(n-2K+1)\mathbb{E}_{\mathbf{w}_0,\ldots,\mathbf{w}_Z}\left[ \min_{\mathbf{x}\in\mathcal{X}} \sum_{i=0}^{Z} (c_{i+1}-c_i)\langle \mathbf{w}_i, \mathbf{x} \rangle \right] \\
&= -(n-2K+1)\mathbb{E}_{\mathbf{w}_0,\ldots,\mathbf{w}_Z}\left[ \min_{\mathbf{x}\in\{-D/2\sqrt{d}, D/2\sqrt{d}\}^d} \left\langle \mathbf{x}, \sum_{i=0}^{Z} (c_{i+1}-c_i)\mathbf{w}_i \right\rangle \right],
\end{aligned} \tag{15}
$$

where the third equality is due to $\mathbb{E}_{\mathbf{w}_0,\ldots,\mathbf{w}_Z}[\langle \mathbf{w}_i, \mathbf{x}_1(t) \rangle] = 0$ for $\forall t \in [c_i + 1, c_{i+1}]$.

Then, we denote $\epsilon_{01}, ..., \epsilon_{0d}, ..., \epsilon_{Z1}, ..., \epsilon_{Zd}$ be the coordinates of $\mathbf{w}_1, ..., \mathbf{w}_Z$, which are identically distributed variables with $\mathbb{P}(\epsilon_{ij} = \pm 1) = 1/2$ for $i \in \{0, ..., Z\}$ and $j \in \{1, ..., d\}$. By using the Khintchine inequality on (15), we have

$$
\begin{aligned}
\mathbb{E}_{\mathbf{w}_0,...,\mathbf{w}_Z}[R(T,1)] &= -(n - 2K + 1)\mathbb{E}_{\epsilon_{01},...,\epsilon_{Zd}}\left[\sum_{j=1}^{d} -\frac{D}{2\sqrt{d}}\left|\sum_{i=0}^{Z}(c_{i+1} - c_i)\frac{\epsilon_{ij}G}{\sqrt{d}}\right|\right] \\
&= (n - 2K + 1)\frac{DG}{2}\mathbb{E}_{\epsilon_{01},...,\epsilon_{Zd}}\left[\left|\sum_{i=0}^{Z}(c_{i+1} - c_i)\epsilon_{i1}\right|\right] \\
&\geq \frac{(n - 2K + 1)DG}{2\sqrt{2}}\sqrt{\sum_{i=0}^{Z}(c_{i+1} - c_i)^2} \\
&\geq \frac{(n - 2K + 1)DG}{2\sqrt{2}}\sqrt{\frac{(c_{Z+1} - c_0)^2}{Z+1}} \\
&= \frac{(n - 2K + 1)DGT}{2\sqrt{2Z + 2}},
\end{aligned}
\tag{16}
$$

where the second inequality is due to the Cauchy-Schwarz inequality.

By applying $Z = \lfloor(T-1)/\lceil\frac{m}{2\omega}\rceil\rfloor \leq \frac{2\omega(T-1)}{m}$, we have

$$
\begin{aligned}
\mathbb{E}_{\mathbf{w}_0,...,\mathbf{w}_Z}[R(T,1)] &\geq \frac{(n - 2K + 1)DGT}{2\sqrt{2Z + 2}} \geq \frac{(n - m - 1)DGT}{2\sqrt{\frac{4\omega(T-1)}{m} + 2}} \\
&= \frac{(m+1)DGT}{2\sqrt{\frac{4\omega(T-1)}{m} + 2}} = \frac{(m+1)\sqrt{m+1}DGT}{2\sqrt{\frac{4\omega(T-1)}{m}(m+1) + 2m + 2}} \\
&\geq \frac{(m+1)\sqrt{m+1}DGT}{2\sqrt{8\omega(T-1) + 2m + 2}} \\
&\geq \frac{n\sqrt{n}DGT}{4\sqrt{16\omega(T-1) + 4m + 4}} \\
&\geq \frac{n\sqrt{n}DGT}{4\sqrt{16\omega T - 16\omega + 2n}},
\end{aligned}
$$

where the fourth inequality is due to $n = 2m + 2$.

Then we introduce a lemma.

**Lemma D.5** (Lemma 6 in Wan et al. (2024a))**.** *For the 1-connected cycle graph with $n = 2(m + 1)$ where $m$ denotes a positive integer, the gossip matrix defined in (13) satisfies*

$$
\frac{\pi^2}{1 - \sigma_2(P)} = \frac{\pi^2}{\rho} \leq 4n^2.
$$

If $n \leq 8\omega T + 8\omega$, by utilizing Lemma D.5, we have

$$
\mathbb{E}_{\mathbf{w}_0,...,\mathbf{w}_Z}[R(T,1)] \geq \frac{nDG\sqrt{nT}}{16\sqrt{2\omega}} \geq \frac{nGD\sqrt{\pi T}}{32\omega^{1/2}\rho^{1/4}}.
$$

### D.5. Proof of Theorem 3.4

For the proof of the lower bound for strongly convex loss functions, we follow the analysis of Wan et al. (2025) while redefining both the loss functions and the decision domain. Specifically, we choose the domain $\mathcal{X} = [0, D/\sqrt{d}]^d$ and define $\mathcal{B}_p$ as the Bernoulli distribution with probability $p$ of obtaining 1. For $t \in [c_i + 1, c_{i+1}]$ and $i \in \{0, ..., Z\}$, we set

$$
f_{t, n-\lceil m/2\rceil + 2}(\mathbf{x}) = \cdots = f_{t,n}(\mathbf{x}) = f_{t,1}(\mathbf{x}) = f_{t,2}(\mathbf{x}) = \cdots = f_{t,\lceil m/2\rceil}(\mathbf{x}) = \frac{\mu}{2}\|\mathbf{x}\|^2,
$$

$$f_{t,\,\lceil m/2 \rceil+1}(\mathbf{x}) = \cdots = f_{t,n-\lceil m/2 \rceil+1}(\mathbf{x}) = h_i(\mathbf{x}) = \frac{\mu}{2}\left\|\mathbf{x} - \frac{D}{\sqrt{d}}\mathbf{w}_i\right\|^2,$$

where $\mathbf{w}_i$ is sampled from the vectors $\mathbf{0}_d$ and $\mathbf{1}_d$ with $\mathbb{P}(\mathbf{w}_i = \mathbf{1}_d) = p$. Clearly, $h_i(\mathbf{x})$ satisfies Assumption 3.3 and Assumption 3.4 with $G = \mu D$. Then, for any $i \in \{0, \ldots, Z\}$ and $t \in [c_i + 1, c_{i+1}]$, the global loss function can be expressed as

$$f_t(\mathbf{x}) = \sum_{k=1}^{n} f_{t,k}(\mathbf{x}) = \frac{\mu}{2}(n - 2K + 1)\left\|\mathbf{x} - \frac{D}{\sqrt{d}}\mathbf{w}_i\right\|^2 + \frac{\mu}{2}(2K - 1)\|\mathbf{x}\|^2$$

$$= \frac{\mu n}{2}\|\mathbf{x}\|^2 - \frac{\mu(n - 2K + 1)D}{\sqrt{d}}\langle \mathbf{x}, \mathbf{w}_i \rangle + \frac{\mu(n - 2K + 1)^2 D^2}{2d}\|\mathbf{w}_i\|^2,$$

with expectation as

$$F(\mathbf{x}) = \mathbb{E}_{\mathbf{w}_0,\ldots,\mathbf{w}_Z}[f_t(\mathbf{x})] = \frac{\mu n}{2}\left\|\mathbf{x} - \frac{(n - 2K + 1)D\mathbf{p}}{n\sqrt{d}}\right\|^2 + \frac{\mu(n - 2K + 1)D^2}{2d}\left\langle \mathbf{1} - \frac{(n - 2K + 1)}{n}\mathbf{p}, \mathbf{p}\right\rangle,$$

where $\mathbf{p} = [p, \ldots, p] \in \mathbb{R}^d$. We denote $F(\mathbf{x}^*)$ be the minimum of $F(\mathbf{x})$. We have

$$\mathbf{x}^* = \frac{(n - 2K + 1)D\mathbf{p}}{n\sqrt{d}} \in \mathcal{X},$$

and we further have the following gap

$$F(\mathbf{x}) - F(\mathbf{x}^*) = \frac{\mu n}{2}\left\|\mathbf{x} - \frac{(n - 2K + 1)D\mathbf{p}}{n\sqrt{d}}\right\|^2. \tag{17}$$

Next, we derive the lower bound for strongly convex functions. We again choose $\mathcal{G}$ as 1-connected cycle graph, which ensures that the $\mathbf{x}_1(c_i + 1), \ldots, \mathbf{x}_1(c_{i+1})$ are independent of $\mathbf{w}_i$.

We have

$$\mathbb{E}_{\mathbf{w}_0,\ldots,\mathbf{w}_Z}[R(T, 1)] = \mathbb{E}_{\mathbf{w}_0,\ldots,\mathbf{w}_Z}\left[\sum_{i=0}^{Z}\sum_{t=c_i+1}^{c_{i+1}} f_t(\mathbf{x}_1(t)) - \min_{\mathbf{x} \in \mathcal{X}}\sum_{i=0}^{Z}\sum_{t=c_i+1}^{c_{i+1}} f_t(\mathbf{x})\right]$$

$$= \mathbb{E}_{\mathbf{w}_0,\ldots,\mathbf{w}_Z}\left[\sum_{i=0}^{Z}\sum_{t=c_i+1}^{c_{i+1}} F(\mathbf{x}_1(t))\right] - \mathbb{E}_{\mathbf{w}_0,\ldots,\mathbf{w}_Z}\left[\min_{\mathbf{x} \in \mathcal{X}}\sum_{i=0}^{Z}\sum_{t=c_i+1}^{c_{i+1}} f_t(\mathbf{x})\right] \tag{18}$$

$$\geq \mathbb{E}_{\mathbf{w}_0,\ldots,\mathbf{w}_Z}\left[\sum_{i=0}^{Z}\sum_{t=c_i+1}^{c_{i+1}} F(\mathbf{x}_1(t)) - \sum_{i=0}^{Z}\sum_{t=c_i+1}^{c_{i+1}} F(\mathbf{x}^*)\right].$$

To give the lower bound of (18), we follow the proof of Wan et al. (2025) to show that the regret of the learner 1 on a specific $p$ is large. Following Wan et al. (2025), we introduce a perturbation of the parameter from $p$ to $p'$, and the corresponding random vectors can be rewritten as $\mathbf{w}_0', \ldots, \mathbf{w}_Z'$ and $\mathbf{x}_1'(0), \ldots, \mathbf{x}_1'(T)$. We assume that the D-OCO algorithm is deterministic without loss of generality. As discussed in Wan et al. (2025), for any $t \in [c_i + 1, c_{i+1}]$, $\mathbf{x}_1(t)$ can be specified by a bit string $X \in \{0, 1\}^i$ drawn from $\mathcal{B}_p^i$. For a deterministic algorithm, the local learner 1 of the D-OCO algorithm at round $t \in [c_i + 1, c_{i+1}]$ can be denoted as a mapping function $\{0, 1\}^i \mapsto \mathcal{X}$ such that $\mathbf{x}_i(t) = \mathcal{A}_t(\mathcal{X})$.

Let $Z_1 = \lfloor \log_{16}(15Z + 16) - 1 \rfloor$ and $Z_1 \geq 1$ due to $16\omega^{-1}n + 1 \leq T$. We further divide the first $Z' = \frac{1}{15}(16^{Z_1+1} - 16)$ intervals into $Z_1$ epochs with the length $16, 16^2, \ldots, 16^{Z_1}$ and the $m$-th epoch $E_m$ consists of the intervals $\frac{1}{15}(16^m - 16), \ldots, \frac{1}{15}(16^{m+1} - 16) - 1$ with length of $16^m$. We utilize a lemma.

**Lemma D.6** (Lemma 8 in Wan et al. (2025)). *Fix a block $i$ and let $\epsilon \leq \frac{1}{32\sqrt{i+1}}$ be a parameter, $\xi = (n - 2K + 1)D/\sqrt{d}$ and $\mathbf{p} = [p, \ldots, p] \in \mathbb{R}^d$. There exists a collection of nested intervals $\left[\frac{1}{4}, \frac{3}{4}\right] \supseteq I_1 \supseteq I_2 \supseteq \cdots \supseteq I_{Z_1}$ such that interval $I_m$ corresponds to epoch $m$, with the property that $I_m$ has length $4^{-(m+3)}$, and for every $p \in I_m$, we have*

$$\mathbb{E}_X\left[\left\|\mathcal{A}_t(X) - \xi\mathbf{p}\right\|_2^2\right] \geq \frac{16^{-(m+3)}\, d\, \xi^2}{8}$$

*over at least half the rounds $t$ in intervals of epoch $m$.*

By using Lemma D.6, there exists a value of $p \in \cap_{m \in [Z_1]} I_m$ such that

$$\mathbb{E}_{\mathbf{w}_0,\ldots,\mathbf{w}_Z}[R(T,1)] \geq \mathbb{E}_{\mathbf{w}_0,\ldots,\mathbf{w}_Z}\left[\sum_{i=0}^{Z} \sum_{t=c_i+1}^{c_{i+1}} \frac{\mu n}{2} \left\| \mathbf{x}_1(t) - \frac{(n-2K+1)D\mathbf{p}}{n\sqrt{d}} \right\|^2\right]$$

$$\geq \mathbb{E}_{\mathbf{w}_0,\ldots,\mathbf{w}_Z}\left[\sum_{i=0}^{Z'} \sum_{t=c_i+1}^{c_{i+1}} \frac{\mu n}{2} \left\| \mathbf{x}_1(t) - \frac{(n-2K+1)D\mathbf{p}}{n\sqrt{d}} \right\|^2\right]$$

$$= \mathbb{E}_{\mathbf{w}_0,\ldots,\mathbf{w}_Z}\left[\sum_{m=1}^{Z_1} \sum_{i \in E_m} \sum_{t=c_i+1}^{c_{i+1}} \frac{\mu n}{2} \left\| \mathbf{x}_1(t) - \frac{(n-2K+1)D\mathbf{p}}{n\sqrt{d}} \right\|^2\right]$$

$$= \sum_{m=1}^{Z_1} \sum_{i \in E_m} \sum_{t=c_i+1}^{c_{i+1}} \mathbb{E}_X\left[ \frac{\mu n}{2} \left\| \mathcal{A}_t(X) - \frac{(n-2K+1)D\mathbf{p}}{n\sqrt{d}} \right\|^2\right]$$

$$\geq \sum_{m=1}^{Z_1} \frac{\left(c_{\frac{1}{15}(16^{m+1}-16)} - c_{\frac{1}{15}(16^m-16)}\right)16^{-(m+3)}\mu(n-2K+1)^2 D^2}{32n}$$

$$= \sum_{m=1}^{Z_1} \frac{K_1\mu(n-2K+1)^2 D^2}{16^4(2n)}$$

$$= \frac{K_1 Z_1 \mu(n-2K+1)^2 D^2}{16^4(2n)},$$

where the first inequality is due to (17) and the third equality is due to $c_i = iK_1$. Moreover, we have

$$\frac{K_1 Z_1(n-2K+1)^2}{2n} \geq \frac{m(\log_{16}(15Z+16)-2)(n-m-1)^2}{4\omega n}$$

$$\geq \frac{(\log_{16}(30\omega(T-1)/n)-2)(n-2)n}{32\omega}.$$

By using Lemma D.5, we can obtain

$$\mathbb{E}_{\mathbf{w}_0,\ldots,\mathbf{w}_Z}[R(T,1)] \geq \frac{(\log_{16}(30\omega(T-1)/n)-2)(n-2)n\mu D^2}{2^{22}\omega}$$

$$\geq \frac{(\log_{16}(30\omega(T-1)/n)-2)(n-2)\pi\mu D^2}{2^{22}\omega\rho^{1/2}}.$$

## D.6. Additional Discussion on Lower Bounds

It is worth noting that our lower bound construction relies on a specific compressor. We clarify that this design is intentional: we aim to construct a worst-case scenario that maximizes the impact of compression noise. This methodology aligns with established literature; for instance, Huang et al. (2022) derive lower bounds by employing the specific Rand$_k$ compressor. Due to the unique characteristics of our online setting and graph topology, we select the Randomized gossip compressor, which satisfies the definition of a compression operator.

## D.7. Proof of Theorem 4.1

In the one-point bandit feedback, we perform gradient descent on the function $\hat{f}_{t,i}(\mathbf{x})$ over the domain $\mathcal{X}_\epsilon$. Since Assumptions 3.4 and 4.1 hold, the value of the loss function is bounded. For convenience of the proof, we further assume that the absolute value of all loss functions $f_{t,i}(\cdot)$ over $\mathcal{X}$ is bounded by a constant $V$. According to Theorem 1 in Flaxman et al. (2005), we have the following

$$\mathbb{E}[\hat{\mathbf{g}}_{t,i}] = \nabla\hat{f}_{t,i}(\mathbf{x}_i(b)), \mathbb{E}\left[\|\hat{\mathbf{g}}_{t,i}\|^2\right] \leq \frac{d^2 V^2}{\epsilon^2} \tag{19}$$

Compared to the proof under the full information setting, the additional error in the bandit setting lies in 2 aspects: (i) the error caused by the gradient estimator; (ii) the error caused by the surrogate domain $\mathcal{X}_\epsilon$. We have the following inequality:

$$|\hat{f}_{t,i}(\mathbf{x}) - f_{t,i}(\mathbf{x})| \leq G\epsilon.$$

Then we introduce a lemma to give the error of Algorithm 5.

**Lemma D.7** (Observation 1 in Flaxman et al. (2005))**.** *The optimum in $\mathcal{X}_\epsilon$ is near the optimum in $\mathcal{X}$.*

$$\min_{\mathbf{x} \in \mathcal{X}_\epsilon} \sum_{t=1}^{T} \sum_{j=1}^{n} f_{t,j}(\mathbf{x}) \leq 2\epsilon VnT/r + \min_{\mathbf{x} \in \mathcal{X}} \sum_{t=1}^{T} \sum_{j=1}^{n} f_{t,j}(\mathbf{x}).$$

Therefore, we can derive the regret bound in the one-point bandit feedback setting.

$$
\begin{aligned}
\mathbb{E}[R(T,i)] &= \mathbb{E}\left[\sum_{t=1}^{T}\sum_{i=1}^{n} f_{t,j}(\mathbf{x}_{i,1}(t)) - \min_{\mathbf{x}\in\mathcal{X}}\sum_{t=1}^{T}\sum_{j=1}^{n} f_{t,j}(\mathbf{x})\right] \\
&\leq \mathbb{E}\left[\sum_{b=1}^{T/L}\sum_{t=bL+1}^{(b+1)L}\sum_{j=1}^{n} f_{t,j}(\mathbf{x}_i(b)+\epsilon\mathbf{u}_{t,i}) - \min_{\mathbf{x}\in\mathcal{X}_\epsilon}\sum_{t=1}^{T}\sum_{j=1}^{n} f_{t,j}(\mathbf{x})\right] + \frac{2\epsilon VnT}{r} \\
&\leq \mathbb{E}\left[\sum_{b=1}^{T/L}\sum_{t=bL+1}^{(b+1)L}\sum_{j=1}^{n} f_{t,j}(\mathbf{x}_i(b)) - \min_{\mathbf{x}\in\mathcal{X}_\epsilon}\sum_{t=1}^{T}\sum_{j=1}^{n} f_{t,j}(\mathbf{x})\right] + \frac{2\epsilon VnT}{r} + G\epsilon nT \\
&\leq \underbrace{\mathbb{E}\left[\sum_{b=1}^{T/L}\sum_{t=bL+1}^{(b+1)L}\sum_{j=1}^{n} \hat{f}_{t,j}(\mathbf{x}_i(b)) - \min_{\mathbf{x}\in\mathcal{X}_\epsilon}\sum_{t=1}^{T}\sum_{j=1}^{n} \hat{f}_{t,j}(\mathbf{x})\right]}_{\text{TERM}_R} + \frac{2\epsilon VnT}{r} + 3G\epsilon nT,
\end{aligned}
\tag{20}
$$

where the first inequality is due to Lemma D.7, the second inequality is due to $f_{t,i}(\mathbf{x}_i(b)+\epsilon\mathbf{u}_{t,i}) \leq f_{t,i}(\mathbf{x}_i(b)) + G\epsilon$ and the last inequality is due to $|\hat{f}_{t,i}(\mathbf{x}) - f_{t,i}(\mathbf{x})| \leq G\epsilon$.

$\text{TERM}_R$ is the regret of the loss function $\hat{f}_{t,i}(\cdot)$. By replacing the gradient norm with (19), we can directly use the proof of Theorem 3.1 and obtain

$$\mathbb{E}_{\mathcal{C}}\left[\text{TERM}_R\right] \leq \frac{2nR^2}{\eta} + (19n+16)LT\eta\frac{d^2V^2}{\epsilon^2}.$$

Therefore, by setting $\eta = \frac{R\epsilon}{d\sqrt{LT}}, \epsilon = cd^{1/2}L^{1/4}T^{-1/4}$, where $c$ is a constant such that $\epsilon \leq r$, we can derive the final regret

$$
\begin{aligned}
\mathbb{E}_{\mathcal{C}}[R(T,i)] &\leq \frac{2nR^2}{\eta} + (19n+16)LT\eta\frac{d^2V^2}{\epsilon^2} + \frac{2\epsilon VnT}{r} + 3G\epsilon nT \\
&\leq O(nd^{1/2}L^{1/4}T^{3/4}) = O(\omega^{-1/4}\rho^{-1/2}d^{1/2}n(\ln n)^{1/4}T^{3/4}).
\end{aligned}
$$

### D.8. Proof of Theorem 4.2

As for the strongly convex functions, we can directly apply Lemma D.3 to $\text{TERM}_R$ and set $\eta_b = \frac{1}{bL+8L}$, we have

$$
\begin{aligned}
\mathbb{E}_{\mathcal{C}}\left[\text{TERM}_R\right] &\leq 2nR^2(\frac{1}{\eta_1} - \mu L) + 3L^2n\frac{d^2V^2}{\epsilon^2}\sum_{b=1}^{T/L}\eta_b + 6\sqrt{3}nL^2\frac{d^2V^2}{\epsilon^2}\sum_{b=1}^{T/L}\eta_b \\
&\quad + (12n+9)L^2\frac{d^2V^2}{\epsilon^2}\sum_{b=1}^{T/L}\eta_b + (n+12)L^2\frac{d^2V^2}{\epsilon^2}\sum_{b=1}^{T/L}\eta_b + (8n+6)L^2\frac{d^2V^2}{\epsilon^2}\sum_{b=1}^{T/L}\eta_{b-1} \\
&\leq 16nR^2\mu + \frac{d^2V^2}{\epsilon^2\mu}(23n+6\sqrt{3}n+39)L\ln(T+8).
\end{aligned}
\tag{21}
$$

By combining (20) with (21) and setting $\epsilon = cd^{2/3}L^{1/3}(\ln(T+8))^{1/3}T^{-1/3}$, where $c$ is a constant such that $\epsilon \leq r$, we can

obtain

$$
\begin{aligned}
\mathbb{E}_{\mathcal{C}}[R(T, i)] &\leq \mathtt{TERM_R} + \frac{2\epsilon V nT}{r} + 3G\epsilon nT \\
&\leq 16nR^2\mu + \frac{d^2V^2}{\epsilon^2\mu}(23n + 6\sqrt{3}n + 39)L\ln(T + 8) + \frac{2\epsilon V nT}{r} + 3G\epsilon nT \\
&\leq O(L^{1/3}d^{2/3}nT^{2/3}\ln(T + 8)) \\
&= O(\omega^{-1/3}\rho^{-2/3}d^{2/3}n(\ln n)^{1/3}T^{2/3}(\ln T)^{1/3}).
\end{aligned}
$$

### D.9. Proof of Theorem 4.3

The proof for the two-point bandit case follows a procedure analogous to that of the one-point case. We introduce the following lemma from Shamir (2017) and Lin et al. (2022)

**Lemma D.8** (Lemma E.1 in Lin et al. (2022)). *For the two-point bandit estimator defined in* (6)*, we have*

$$
\mathbb{E}\left[\hat{\mathbf{g}}_{t,i}\right] = \nabla\hat{f}_{t,i}(\mathbf{x}_i(b)), \mathbb{E}\left[\|\hat{\mathbf{g}}_{t,i}\|^2\right] \leq 16\sqrt{2\pi}dG^2.
$$

**Remark.** The two-point estimator in (6) is originally proposed by Agarwal et al. (2010), whose analysis provides a bound of $\mathbb{E}\left[\|\hat{\mathbf{g}}_{t,i}\|^2\right] \leq O(d^2G^2)$. Shamir (2017) improves this result by proving that $\mathbb{E}\left[\|\hat{\mathbf{g}}_{t,i}\|^2\right] \leq C_2 dG^2$ for some constant $C_2$. Lin et al. (2022) explicitly identifies this constant. To simplify our proof, we denote $C_2 = 16\sqrt{2\pi}$.

Then we derive the regret bound

$$
\begin{aligned}
\mathbb{E}_{\mathcal{C}}\left[R_2(T, i)\right] =&\mathbb{E}_{\mathcal{C}}\left[\sum_{b=1}^{T/L}\sum_{t=(b-1)L+1}^{bL}\sum_{j=1}^{n}\frac{f_{t,j}(\mathbf{x}_{i,1}(t)) + f_{t,j}(\mathbf{x}_{i,2}(t))}{2} - \min_{\mathbf{x}\in\mathcal{X}}\sum_{t=1}^{T}\sum_{j=1}^{n}f_{t,j}(\mathbf{x})\right] \\
\leq&\mathbb{E}_{\mathcal{C}}\left[\sum_{b=1}^{T/L}\sum_{t=bL+1}^{bL+L}\sum_{j=1}^{n}\hat{f}_{t,j}(\mathbf{x}_i(b)) - \min_{\mathbf{x}\in\mathcal{X}_\epsilon}\sum_{t=1}^{T}\sum_{j=1}^{n}\hat{f}_{t,j}(\mathbf{x})\right] + 3G\epsilon nT + \frac{2\epsilon V nT}{r} \\
\leq&\mathbb{E}_{\mathcal{C}}\left[\mathtt{TERM_R}\right] + \frac{2\epsilon V nT}{r} + 3G\epsilon nT,
\end{aligned}
$$

where the first inequality is due to $f_{t,i}(\mathbf{x}_i(b) + \epsilon\mathbf{u}_{t,i}) \leq f_{t,i}(\mathbf{x}_i(b)) + G\epsilon$ and the last inequality is due to $|\hat{f}_{t,i}(\mathbf{x}) - f_{t,i}(\mathbf{x})| \leq G\epsilon$.

To bound the term $\mathtt{TERM_R}$, we directly follow the proof of Theorem 4.1 and replace norm of the gradient with $C_2 dG^2$. We have

$$
\mathbb{E}_{\mathcal{C}}\left[\mathtt{TERM_R}\right] \leq \frac{2nR^2}{\eta} + (19n + 16)C_2 LT\eta dG^2.
$$

Therefore, by setting $\eta = \frac{2R}{G\sqrt{dLT}}, \epsilon = cT^{-1/2}$, where $c$ is a constant such that $\epsilon \leq r$, we can derive the final regret bound

$$
\begin{aligned}
\mathbb{E}_C[R(T, i)] \leq&\frac{2nR^2}{\eta} + (19n + 16)C_2 LT\eta dG^2 + \frac{2\epsilon V nT}{r} + 3G\epsilon nT \\
\leq&O(nd^{1/2}L^{1/2}T^{1/2}) = O(\omega^{-1/2}\rho^{-1}d^{1/2}n(\ln n)^{1/2}T^{1/2}).
\end{aligned}
$$

### D.10. Proof of Theorem 4.4

The key difference of this part is to use the strong convexity to derive a tighter bound for $\mathtt{TERM_R}$. By setting $\eta_b = \frac{1}{bL+8L}$, we have

$$
\mathbb{E}_{\mathcal{C}}\left[\mathtt{TERM_R}\right] \leq 16nR^2\mu + \frac{1}{\mu}(23n + 6\sqrt{3}n + 39)C_2 dG^2 L\ln(T + 8). \tag{22}
$$

By combining (20) with (22) and setting $\epsilon = \frac{c \ln T}{T}$, where $c$ is a constant such that $\epsilon \leq r$, we can obtain

$$
\begin{aligned}
\mathbb{E}_{\mathcal{C}}[R(T,i)] &\leq \mathtt{TERM_R} + \frac{2\epsilon V n T}{r} + 3G\epsilon n T \\
&\leq 16nR^2\mu + \frac{1}{\mu}(23n + 6\sqrt{3}n + 39)C_2 dG^2 L \ln(T+8) + \frac{2\epsilon V n T}{r} + 3G\epsilon n T \\
&\leq O(dLn\ln(T+8)) \\
&= O(\omega^{-1}\rho^{-2}dn\ln n \ln T).
\end{aligned}
$$

# E. Proof for Supporting Lemmas

## E.1. Proof of Lemma D.1

Since each learner $i$ maintains the local auxiliary variable $\hat{\mathbf{x}}_j(b)$ to store the data from the neighbor $j \in \mathcal{N}_i$, the variable $\hat{\mathbf{x}}_i(b)$ is same over all learner $j \in \mathcal{N}_i$. Therefore, we have

$$
\sum_{i=1}^{n} \sum_{j \in \mathcal{N}_i} P_{ij}(\hat{\mathbf{x}}_j(b) - \hat{\mathbf{x}}_i(b)) = \mathbf{0}.
$$

Then we will demonstrate that the average decision $\overline{\mathbf{y}}^k(b)$ is same over $k \in [1, L_1 + 1]$,

$$
\frac{1}{n}\sum_{i=1}^{n} \mathbf{y}_i^{(L_1+1)}(b) = \overline{\mathbf{y}}^{k+1}(b) = \overline{\mathbf{y}}^k(b) + \gamma\frac{1}{n}\sum_{i=1}^{n}\sum_{j \in \mathcal{N}_i} P_{ij}(\hat{\mathbf{y}}_j^k(b) - \hat{\mathbf{y}}_i^b(k)) = \overline{\mathbf{y}}^k(b),
$$

which implies that

$$
\overline{\mathbf{y}}^{(L_1+1)}(b) = \frac{1}{n}\sum_{i=1}^{n}\mathbf{y}_i^{(L_1+1)}(b) = \frac{1}{n}\sum_{i=1}^{n}\mathbf{y}_i^{(1)}(b) = \frac{1}{n}\sum_{i=1}^{n}\mathbf{x}_i(b) = \overline{\mathbf{x}}(b).
$$

We can rewrite that

$$
\begin{aligned}
\overline{\mathbf{x}}(b+1) &= \frac{1}{n}\sum_{i=1}^{n}\tilde{\mathbf{x}}_i(b+1) + \mathbf{r}_i(b+1) = \frac{1}{n}\sum_{i=1}^{n}\mathbf{y}_i^{(L_1+1)}(b) + \frac{1}{n}\sum_{i=1}^{n}\mathbf{r}_i(b+1) \\
&= \frac{1}{n}\sum_{i=1}^{n}\mathbf{y}_i^{(L_1+1)}(b) + \sum_{i=1}^{n}\sum_{j \in \mathcal{N}_i}\gamma P_{ij}(\hat{\mathbf{y}}_j^{(L_1+1)}(b) - \hat{\mathbf{y}}_i^{(L_1+1)}(b)) + \frac{1}{n}\sum_{i=1}^{n}\mathbf{r}_i(b+1) \\
&= \frac{1}{n}\sum_{i=1}^{n}\mathbf{y}_i^{(L_1+1)}(b) + \frac{1}{n}\sum_{i=1}^{n}\mathbf{r}_i(b+1) \\
&= \frac{1}{n}\sum_{i=1}^{n}\mathbf{y}_i^{(1)}(b) + \frac{1}{n}\sum_{i=1}^{n}\mathbf{r}_i(b+1) \\
&= \frac{1}{n}\sum_{i=1}^{n}\mathbf{x}_i(b) - \frac{\eta_b}{n}\sum_{i=1}^{n}\mathbf{z}_i(b-1) + \frac{1}{n}\sum_{i=1}^{n}\mathbf{r}_i(b+1) \\
&= \overline{\mathbf{x}}(b) - \frac{\eta_b}{n}\sum_{i=1}^{n}\mathbf{z}_i(b-1) + \frac{1}{n}\sum_{i=1}^{n}\mathbf{r}_i(b+1).
\end{aligned}
$$

For any $\mathbf{x} \in \mathcal{X}$, we have

$$
\begin{aligned}
\|\overline{\mathbf{x}}(b+1) - \mathbf{x}\|^2 &= \|\overline{\mathbf{x}}(b) - \mathbf{x}\|^2 + \frac{1}{n^2} \left\| \sum_{i=1}^n \mathbf{r}_i(b+1) - \eta_b \sum_{j=1}^n \mathbf{z}_j(b-1) \right\|^2 \\
&\quad + 2\left\langle \frac{1}{n}\sum_{i=1}^n \mathbf{r}_i(b+1), \overline{\mathbf{x}}(b) - \mathbf{x} \right\rangle - \frac{2\eta_b}{n}\sum_{j=1}^n \langle \mathbf{z}_j(b-1), \overline{\mathbf{x}}(b) - \mathbf{x} \rangle \\
&= \|\overline{\mathbf{x}}(b) - \mathbf{x}\|^2 + \frac{1}{n^2} \left\| \sum_{i=1}^n \mathbf{r}_i(b+1) - \eta_b \sum_{j=1}^n \mathbf{z}_j(b-1) \right\|^2 \\
&\quad + 2\left\langle \frac{1}{n}\sum_{i=1}^n \mathbf{r}_i(b+1), \overline{\mathbf{x}}(b) - \mathbf{x} \right\rangle - \frac{2\eta_b}{n}\sum_{j=1}^n \left\langle \sum_{t=(b-2)L+1}^{(b-1)L} \nabla f_{t,j}(\mathbf{x}_j(b-1)), \overline{\mathbf{x}}(b) - \mathbf{x} \right\rangle.
\end{aligned}
\tag{23}
$$

For the second term, we have

$$
\frac{1}{n^2} \left\| \sum_{i=1}^n \mathbf{r}_i(b+1) - \eta_b \sum_{j=1}^n \mathbf{z}_j(b-1) \right\|^2 \leq \frac{2}{n} \|R(b+1)\|_F^2 + 2L^2 \eta_b^2 G^2.
\tag{24}
$$

For the third term, we have

$$
\begin{aligned}
&2\langle \frac{1}{n}\sum_{i=1}^n \mathbf{r}_i(b+1), \overline{\mathbf{x}}(b) - \mathbf{x}\rangle \\
&= \frac{2}{n}\sum_{i=1}^n \langle \mathbf{r}_i(b+1), \overline{\mathbf{x}}(b) - \tilde{\mathbf{x}}_i(b+1) + \tilde{\mathbf{x}}_i(b+1) - \mathbf{x}\rangle \\
&= \frac{2}{n}\sum_{i=1}^n \langle \mathbf{r}_i(b+1), \overline{\mathbf{x}}(b) - \tilde{\mathbf{x}}_i(b+1)\rangle + \frac{2}{n}\sum_{i=1}^n \langle \mathbf{r}_i(b+1), \tilde{\mathbf{x}}_i(b+1) - \mathbf{x}\rangle \\
&= \frac{2}{n}\sum_{i=1}^n \langle \mathbf{r}_i(b+1), \overline{\mathbf{x}}(b) - \tilde{\mathbf{x}}_i(b+1)\rangle + \frac{2}{n}\sum_{i=1}^n \langle P_{\mathcal{X}}(\tilde{\mathbf{x}}_i(b+1)) - \tilde{\mathbf{x}}_i(b+1), \tilde{\mathbf{x}}_i(b+1) - \mathbf{x}\rangle \\
&\leq \frac{1}{n}\sum_{i=1}^n \left( \|\mathbf{r}_i(b+1)\|^2 + \|\overline{\mathbf{x}}(b) - \tilde{\mathbf{x}}_i(b+1)\|^2 \right) \\
&\leq \frac{1}{n} \left( \|R(b+1)\|_F^2 + \left\| \overline{X}(b) - \tilde{X}(b+1) \right\|_F^2 \right),
\end{aligned}
\tag{25}
$$

where the first inequality is due to $2\langle a, b\rangle \leq \|a\|^2 + \|b\|^2$ and inequality (8).

By using the convexity, we have

$$
f_{t,j}(\mathbf{x}_j(b)) \geq f_{t,j}(\mathbf{x}_i(b)) - G \|\mathbf{x}_i(b) - \mathbf{x}_j(b)\|,
$$

and

$$
\begin{aligned}
&-\frac{\eta_b}{n}\sum_{j=1}^n \sum_{t=(b-2)L+1}^{(b-1)L} \langle \nabla f_{t,j}(\mathbf{x}_j(b-1)), \overline{\mathbf{x}}(b) - \mathbf{x}\rangle \\
&= -\frac{\eta_b}{n}\sum_{j=1}^n \sum_{t=(b-2)L+1}^{(b-1)L} \langle \nabla f_{t,j}(\mathbf{x}_j(b-1)), \overline{\mathbf{x}}(b) - \overline{\mathbf{x}}(b-1) + \overline{\mathbf{x}}(b-1) - \mathbf{x}\rangle \\
&= -\frac{\eta_b}{n}\sum_{j=1}^n \sum_{t=(b-2)L+1}^{(b-1)L} \langle \nabla f_{t,j}(\mathbf{x}_j(b-1)), \overline{\mathbf{x}}(b) - \overline{\mathbf{x}}(b-1)\rangle - \frac{\eta_b}{n}\sum_{j=1}^n \sum_{t=(b-2)L+1}^{(b-1)L} \langle \nabla f_{t,j}(\mathbf{x}_j(b-1)), \overline{\mathbf{x}}(b-1) - \mathbf{x}\rangle.
\end{aligned}
$$

Next, we give the bound of these two terms. For the first term, we have

$$
-\frac{\eta_b}{n}\sum_{j=1}^{n}\sum_{t=(b-2)L+1}^{(b-1)L}\langle\nabla f_{t,j}(\mathbf{x}_j(b-1)),\overline{\mathbf{x}}(b)-\overline{\mathbf{x}}(b-1)\rangle
$$

$$
=\langle\frac{\eta_b}{n}\sum_{j=1}^{n}\sum_{t=(b-2)L+1}^{(b-1)L}\nabla f_{t,j}(\mathbf{x}_j(b-1)),\frac{\eta_{b-1}}{n}\sum_{j=1}^{n}\sum_{t=(b-2)L+1}^{(b-1)L}\nabla f_{t,j}(\mathbf{x}_j(b-2))\rangle
$$

$$
-\langle\frac{\eta_b}{n}\sum_{j=1}^{n}\sum_{t=(b-2)L+1}^{(b-1)L}\nabla f_{t,j}(\mathbf{x}_j(b-1)),\frac{1}{n}\sum_{i=1}^{n}\mathbf{r}_i(b)\rangle
$$

$$
\le\left\|\frac{\eta_b}{n}\sum_{j=1}^{n}\sum_{t=(b-2)L+1}^{(b-1)L}\nabla f_{t,j}(\mathbf{x}_j(b-1))\right\|\left\|\frac{\eta_{b-1}}{n}\sum_{j=1}^{n}\sum_{t=(b-2)L+1}^{(b-1)L}\nabla f_{t,j}(\mathbf{x}_j(b-2))\right\|
$$

$$
+\frac{1}{2}\left\|\frac{\eta_b}{n}\sum_{j=1}^{n}\sum_{t=(b-2)L+1}^{(b-1)L}\nabla f_{t,j}(\mathbf{x}_j(b-1))\right\|^2+\frac{1}{2}\left\|\frac{1}{n}\sum_{i=1}^{n}\mathbf{r}_i(b)\right\|^2
$$

$$
\le\eta_b\eta_{b-1}G^2L^2+\frac{1}{2}\eta_b^2G^2L^2+\frac{1}{2n}\|R(b)\|_F^2\,,
$$

(26)

where the first equality is due to $\overline{\mathbf{x}}(b)=\overline{\mathbf{x}}(b-1)-\frac{\eta_{b-1}}{n}\sum_{j=1}^{n}\sum_{t=(b-2)L+1}^{(b-1)L}\nabla f_{t,j}(\mathbf{x}_j(b-2))+\frac{1}{n}\sum_{i=1}^{n}\mathbf{r}_i(b)$ and the first inequality is due to $\langle a,b\rangle\le\|a\|\|b\|$ and $-\langle a,b\rangle\le\frac{\|a\|^2+\|b\|^2}{2}$.

For the second term, we have

$$
-\frac{\eta_b}{n}\sum_{j=1}^{n}\sum_{t=(b-2)L+1}^{(b-1)L}\langle\nabla f_{t,j}(\mathbf{x}_j(b-1)),\overline{\mathbf{x}}(b-1)-\mathbf{x}\rangle
$$

$$
=\frac{\eta_b}{n}\sum_{j=1}^{n}\sum_{t=(b-2)L+1}^{(b-1)L}\langle\nabla f_{t,j}(\mathbf{x}_j(b-1)),\mathbf{x}-\overline{\mathbf{x}}(b-1)\rangle
$$

$$
=\frac{\eta_b}{n}\sum_{j=1}^{n}\sum_{t=(b-2)L+1}^{(b-1)L}\langle\nabla f_{t,j}(\mathbf{x}_j(b-1)),\mathbf{x}-\mathbf{x}_j(b-1)\rangle
$$

$$
+\frac{\eta_b}{n}\sum_{j=1}^{n}\sum_{t=(b-2)L+1}^{(b-1)L}\langle\nabla f_{t,j}(\mathbf{x}_j(b-1)),\mathbf{x}_j(b-1)-\overline{\mathbf{x}}(b-1)\rangle
$$

$$
\le\frac{\eta_b}{n}\sum_{j=1}^{n}\sum_{t=(b-2)L+1}^{(b-1)L}f_{t,j}(\mathbf{x})-f_{t,j}(\mathbf{x}_j(b-1))+\frac{\eta_b}{n}\sum_{j=1}^{n}GL\|\mathbf{x}_j(b-1)-\overline{\mathbf{x}}(b-1)\|
$$

$$
=\frac{\eta_b}{n}\sum_{j=1}^{n}\sum_{t=(b-2)L+1}^{(b-1)L}f_{t,j}(\mathbf{x})-f_{t,j}(\mathbf{x}_i(b-1))+f_{t,j}(\mathbf{x}_i(b-1))-f_{t,j}(\mathbf{x}_j(b-1))
$$

$$
+\frac{\eta_b}{n}\sum_{j=1}^{n}GL\|\mathbf{x}_j(b-1)-\overline{\mathbf{x}}(b-1)\|
$$

$$
\le\frac{\eta_b}{n}\sum_{j=1}^{n}\sum_{t=(b-2)L+1}^{(b-1)L}f_{t,j}(\mathbf{x})-f_{t,j}(\mathbf{x}_i(b-1))+\frac{\eta_b}{n}GL\sum_{j=1}^{n}\|\mathbf{x}_i(b-1)-\mathbf{x}_j(b-1)\|
$$

$$
+\frac{\eta_b}{n}GL\sum_{j=1}^{n}\|\mathbf{x}_j(b-1)-\overline{\mathbf{x}}(b-1)\|\,,
$$

where the first and the second inequalities are due to the convexity.

By using the fact that

$$\sum_{j=1}^{n} \|\mathbf{x}_j(b-1) - \overline{\mathbf{x}}(b-1)\| \le \sqrt{n} \left\| X(b-1) - \overline{X}(b-1) \right\|_F,$$

$$\sum_{j=1}^{n} \|\mathbf{x}_i(b-1) - \mathbf{x}_j(b-1)\|$$
$$= \sum_{j=1}^{n} \|\overline{\mathbf{x}}(b-1) - \mathbf{x}_j(b-1)\| + n \|\mathbf{x}_i(b-1) - \overline{\mathbf{x}}(b-1)\|$$
$$\le \sqrt{n} \left\| X(b-1) - \overline{X}(b-1) \right\|_F + n \left\| X(b-1) - \overline{X}(b-1) \right\|_F,$$

and thus we have

$$-\frac{\eta_b}{n} \sum_{j=1}^{n} \sum_{t=(b-2)L+1}^{(b-1)L} \langle \nabla f_{t,j}(\mathbf{x}_i(b-1)), \overline{\mathbf{x}}(b) - \mathbf{x} \rangle$$
$$\le \frac{\eta_b}{n} \sum_{j=1}^{n} \sum_{t=(b-2)L+1}^{(b-1)L} f_{t,j}(\mathbf{x}) - f_{t,j}(\mathbf{x}_i(b-1)) + \frac{\eta_b}{n} 3nGL \left\| X(b-1) - \overline{X}(b-1) \right\|_F. \tag{27}$$

By combining (24), (25), (26) and (27), we can derive

$$\|\overline{\mathbf{x}}(b+1) - \mathbf{x}\|^2 = \|\overline{\mathbf{x}}(b) - \mathbf{x}\|^2 + 3L^2 \eta_b^2 G^2 + \frac{3}{n} \|R(b+1)\|_F^2 + \frac{1}{n} \left\| X(b) - \tilde{X}(b+1) \right\|_F + 2\eta_b \eta_{b-1} G^2 L^2$$
$$+ \frac{1}{n} \|R(b)\|_F^2 + \frac{2\eta_b}{n} \sum_{j=1}^{n} \sum_{t=(b-2)L+1}^{(b-1)L} f_{t,j}(\mathbf{x}) - f_{t,j}(\mathbf{x}_i(b-1)) + \frac{6\eta_b}{n} nGL \left\| X(b-1) - \overline{X}(b-1) \right\|_F,$$

which implies

$$\sum_{t=(b-2)L+1}^{(b-1)L} \sum_{j=1}^{n} f_{t,j}(\mathbf{x}_i(b-1)) - f_{t,j}(\mathbf{x})$$
$$\le \frac{n}{2\eta_b} (\|\overline{\mathbf{x}}(b) - \mathbf{x}\|^2 - \|\overline{\mathbf{x}}(b+1) - \mathbf{x}\|^2) + \frac{3}{2\eta_b} \|R(b+1)\|_F^2 + \frac{1}{2\eta_b} \|R(b)\|_F^2 + \frac{3}{2} L^2 n \eta_b G^2 + L^2 n \eta_{b-1} G^2$$
$$+ \frac{1}{2\eta_b} \left\| X(b) - \tilde{X}(b+1) \right\|_F^2 + 3nGL \left\| X(b-1) - \overline{X}(b-1) \right\|_F.$$

By summing up over all blocks, we can derive

$$\mathbb{E}_{\mathcal{C}}[R(T,i)] = \sum_{b=1}^{T/L} \sum_{t=(b-1)L+1}^{bL} \sum_{j=1}^{n} f_{t,j}(\mathbf{x}_i(b)) - \sum_{t=1}^{T} \sum_{j=1}^{n} f_{t,j}(\mathbf{x})$$
$$\le \frac{nD^2}{2\eta_{T/L}} + 3L^2 G^2 n \sum_{b=1}^{T/L} \eta_b + \sum_{b=1}^{T/L} \frac{3}{2\eta_b} \mathbb{E}_{\mathcal{C}} \left[ \|R(b+1)\|_F^2 \right] + \frac{1}{2\eta_b} \mathbb{E}_{\mathcal{C}} \left[ \left\| X(b) - \tilde{X}(b+1) \right\|_F^2 \right] \tag{28}$$
$$+ 3nGL \sum_{b=1}^{T/L} \mathbb{E}_{\mathcal{C}} \left[ \left\| X(b) - \overline{X}(b) \right\|_F \right] + \sum_{b=1}^{T/L} \frac{1}{2\eta_b} \mathbb{E}_{\mathcal{C}} \left[ \|R(b)\|_F^2 \right].$$

## E.2. Proof of Lemma D.2

Before we give the proof of Lemma D.2, we first introduce a lemma to give the guarantee of our online gossip technique.

**Lemma E.1.** *Given an $\omega$-contractive compressor $\mathcal{C}(\cdot)$ and setting the communication rounds $L_1 = \lceil \frac{2 \ln(14n)}{\gamma \rho} \rceil$ and step size $\gamma = \frac{\omega \rho}{2 \rho \beta^2 + 4\beta^2 + (2-\omega)(\beta^2 + 2\beta)\rho + \rho^2}$, we have*

$$e_{L_1+1} \le \frac{1}{14n} e_1.$$

For projection error $\mathbf{r}_i(b+1)$, since $\mathcal{X}$ is convex, $\overline{\mathbf{x}}(b) = \frac{1}{n} \sum_{i=1}^{n} \mathbf{x}_i(b) \in \mathcal{X}$ and $(1-\gamma)\mathbf{x}_i(b) + \gamma \sum_{j \in \mathcal{N}_i} P_{ij} \mathbf{x}_j(b) \in \mathcal{X}$, for $\gamma \in (0, 1]$, we have

$$
\begin{aligned}
\|\mathbf{r}_i(b+1)\|^2 &= \|\Pi_{\mathcal{X}}[\tilde{\mathbf{x}}_i(b+1)] - \tilde{\mathbf{x}}_i(b+1)\|^2 \\
&\le \left\| \overline{\mathbf{x}}(b) - \mathbf{y}_i^{(L_1+1)}(b) \right\|^2 \\
&= \left\| \overline{\mathbf{x}}(b) - \overline{\mathbf{y}}^{(L_1+1)}(b) + \overline{\mathbf{y}}^{(L_1+1)}(b) - \mathbf{y}_i^{(L_1+1)}(b) \right\|^2 \\
&= \left\| \overline{\mathbf{x}}(b) - \overline{\mathbf{y}}^{(1)}(b) + \overline{\mathbf{y}}^{(L_1+1)}(b) - \mathbf{y}_i^{(L_1+1)}(b) \right\|^2 \\
&= \left\| \overline{\mathbf{x}}(b) - \left(\overline{\mathbf{x}}(b) - \frac{\eta_b}{n} \sum_{i=1}^{n} \mathbf{z}_i(b-1)\right) + \overline{\mathbf{y}}^{(L_1+1)}(b) - \mathbf{y}_i^{(L_1+1)}(b) \right\|^2 \\
&\le 2 \left\| \frac{\eta_b}{n} \sum_{i=1}^{n} \mathbf{z}_i(b-1) \right\|^2 + 2 \left\| \overline{\mathbf{y}}^{(L_1+1)}(b) - \mathbf{y}_i^{(L_1+1)}(b) \right\|^2 \\
&\le 2\eta_b^2 L^2 G^2 + 2 \left\| \overline{\mathbf{y}}^{(L_1+1)}(b) - \mathbf{y}_i^{(L_1+1)}(b) \right\|^2
\end{aligned}
$$

where the third equality is due to $\overline{\mathbf{y}}^{(L_1+1)}(b) = \overline{\mathbf{y}}^{(1)}(b)$.

By using Lemma E.1, we have

$$
\begin{aligned}
\mathbb{E}_{\mathcal{C}} \left[ \|R(b+1)\|_F^2 \right] &= \mathbb{E}_{\mathcal{C}} \left[ \sum_{i=1}^{n} \|\mathbf{r}_i(b)\|^2 \right] \\
&\le 2 \sum_{i=1}^{n} \left\| \overline{\mathbf{y}}^{(L_1+1)}(b) - \mathbf{y}_i^{(L_1+1)}(b) \right\|^2 + 2\eta_b^2 n L^2 G^2 \\
&\le \frac{1}{7n} \mathbb{E}_{\mathcal{C}} \left[ \sum_{i=1}^{n} \left\| \overline{\mathbf{y}}^{(1)}(b) - \mathbf{y}_i^{(1)}(b) \right\|^2 + \left\| \hat{\mathbf{y}}^{(1)}(b) - \mathbf{y}_i^{(1)}(b) \right\|^2 \right] + 2n\eta_b^2 L^2 G^2 \\
&= \frac{1}{7n} \mathbb{E}_{\mathcal{C}} \left[ \sum_{i=1}^{n} \|\overline{\mathbf{x}}(b) - \eta_b \overline{\mathbf{z}}(b-1) - \mathbf{x}_i(b) + \eta_b \mathbf{z}_i(b-1)\|^2 + \|\hat{\mathbf{x}}(b) - \mathbf{x}_i(b) + \eta_b \mathbf{z}_i(b-1)\|^2 \right] \\
&\quad + 2n\eta_b^2 L^2 G^2 \\
&\le \frac{2}{7n} \mathbb{E}_{\mathcal{C}} \left[ \sum_{i=1}^{n} \|\overline{\mathbf{x}}(b) - \mathbf{x}_i(b)\|^2 + \|\hat{\mathbf{x}}(b) - \mathbf{x}_i(b)\|^2 \right] + \frac{10}{7} \eta_b^2 L^2 G^2 + 2n\eta_b^2 L^2 G^2.
\end{aligned}
$$

For the second term, we have

$$
\mathbb{E}_{\mathcal{C}} \left[ \|X(b+1) - \overline{X}(b+1)\|_F^2 \right] = \mathbb{E}_{\mathcal{C}} \left[ \sum_{i=1}^{n} \left\| \mathbf{x}_i(b+1) - \frac{1}{n} \sum_{j=1}^{n} \mathbf{x}_j(b+1) \right\|^2 \right].
$$

Different from the previous work that introduces the additional projection error term, we will prove the equality

$$\sum_{i=1}^{n}\left\|\mathbf{x}_i(b+1)-\frac{1}{n}\sum_{j=1}^{n}\mathbf{x}_j(b+1)\right\|^2 = \frac{1}{2n}\sum_{i=1}^{n}\sum_{j=1}^{n}\|\mathbf{x}_i(b+1)-\mathbf{x}_j(b+1)\|^2, \tag{29}$$

which avoids the incurrence of an additional projection error term.

As for the left term, we have

$$\sum_{i=1}^{n}\left\|\mathbf{x}_i(b+1)-\frac{1}{n}\sum_{j=1}^{n}\mathbf{x}_j(b+1)\right\|^2$$

$$=\sum_{i=1}^{n}\|\mathbf{x}_i(b+1)\|^2 + \left\|\frac{1}{n}\sum_{j=1}^{n}\mathbf{x}_j(b+1)\right\|^2 - 2\langle\mathbf{x}_i(b+1),\frac{1}{n}\sum_{j=1}^{n}\mathbf{x}_j(b+1)\rangle$$

$$=\sum_{i=1}^{n}\|\mathbf{x}_i(b+1)\|^2 + \frac{1}{n}\left\|\sum_{j=1}^{n}\mathbf{x}_j(b+1)\right\|^2 - \frac{2}{n}\langle\sum_{j=1}^{n}\mathbf{x}_j(b+1),\sum_{j=1}^{n}\mathbf{x}_j(b+1)\rangle$$

$$=\sum_{i=1}^{n}\|\mathbf{x}_i(b+1)\|^2 - \frac{1}{n}\left\|\sum_{j=1}^{n}\mathbf{x}_j(b+1)\right\|^2.$$

For the right term, we have

$$\frac{1}{2n}\sum_{i=1}^{n}\sum_{j=1}^{n}\|\mathbf{x}_i(b+1)-\mathbf{x}_j(b+1)\|^2 = \frac{1}{2n}(2n\sum_{i=1}^{n}\|\mathbf{x}_i(b+1)\|^2 - 2\sum_{i=1}^{n}\sum_{j=1}^{n}\langle\mathbf{x}_i(b+1),\mathbf{x}_j(b+1)\rangle)$$

$$=\sum_{i=1}^{n}\|\mathbf{x}_i(b+1)\|^2 - \frac{1}{n}\langle\sum_{i=1}^{n}\mathbf{x}_i(b+1),\sum_{j=1}^{n}\mathbf{x}_j(b+1)\rangle$$

$$=\sum_{i=1}^{n}\|\mathbf{x}_i(b+1)\|^2 - \frac{1}{n}\left\|\sum_{i=1}^{n}\mathbf{x}_i(b+1)\right\|^2.$$

Therefore we can derive equality (29). By using equality (29), we have

$$\mathbb{E}_{\mathcal{C}}\left[\left\|X(b+1)-\overline{X}(b+1)\right\|_F^2\right]$$

$$=\mathbb{E}_{\mathcal{C}}\left[\sum_{i=1}^{n}\left\|\mathbf{x}_i(b+1)-\frac{1}{n}\sum_{j=1}^{n}\mathbf{x}_j(b+1)\right\|^2\right]$$

$$=\frac{1}{2n}\sum_{i=1}^{n}\sum_{j=1}^{n}\mathbb{E}_{\mathcal{C}}\left[\|\mathbf{x}_i(b+1)-\mathbf{x}_j(b+1)\|^2\right] \tag{30}$$

$$\leq\frac{1}{2n}\sum_{i=1}^{n}\sum_{j=1}^{n}\mathbb{E}_{\mathcal{C}}\left[\|\tilde{\mathbf{x}}_i(b+1)-\tilde{\mathbf{x}}_j(b+1)\|^2\right]$$

$$=\sum_{i=1}^{n}\mathbb{E}_{\mathcal{C}}\left[\left\|\tilde{\mathbf{x}}_i(b+1)-\frac{1}{n}\sum_{j=1}^{n}\tilde{\mathbf{x}}_j(b+1)\right\|^2\right].$$

Then we can further derive the upper bound

$$
\sum_{i=1}^{n} \mathbb{E}_{\mathcal{C}} \left[ \left\| \tilde{\mathbf{x}}_i(b+1) - \frac{1}{n} \sum_{j=1}^{n} \tilde{\mathbf{x}}_j(b+1) \right\|^2 \right]
$$

$$
= \sum_{i=1}^{n} \mathbb{E}_{\mathcal{C}} \left[ \left\| \mathbf{y}_i^{(L_1+1)}(b) - \overline{\mathbf{y}}_i^{(L_1+1)}(b) \right\|^2 \right]
$$

$$
\leq \frac{1}{14n} \mathbb{E}_{\mathcal{C}} \left[ \sum_{i=1}^{n} \left\| \mathbf{y}_i^{(1)}(b) - \overline{\mathbf{y}}_i^{(1)}(b) \right\|^2 \right] + \frac{1}{14n} \mathbb{E}_{\mathcal{C}} \left[ \sum_{i=1}^{n} \left\| \mathbf{y}_i^{(1)}(b) - \hat{\mathbf{y}}_i^{(1)}(b) \right\|^2 \right]
$$

$$
= \frac{1}{14n} \mathbb{E}_{\mathcal{C}} \left[ \sum_{i=1}^{n} \left\| \mathbf{x}_i(b) - \eta_b \mathbf{z}_i(b-1) - \overline{\mathbf{x}}(b) + \eta_b \overline{\mathbf{z}}(b-1) \right\|^2 \right] + \frac{1}{14n} \mathbb{E}_{\mathcal{C}} \left[ \sum_{i=1}^{n} \left\| \mathbf{x}_i(b) - \eta_b \mathbf{z}_i(b-1) - \hat{\mathbf{x}}_i(b) \right\|^2 \right]
$$

$$
\leq \frac{1}{7n} \mathbb{E}_{\mathcal{C}} \left[ \sum_{i=1}^{n} \| \mathbf{x}_i(b) - \overline{\mathbf{x}}(b) \|^2 + \sum_{i=1}^{n} \| \mathbf{x}_i(b) - \hat{\mathbf{x}}_i(b) \|^2 \right] + \frac{5}{7} L^2 G^2 \eta_b^2
$$

$$
\leq \frac{1}{7n} \left( \left\| X(b) - \overline{X}(b) \right\|_F^2 + \left\| X(b) - \hat{X}(b) \right\|_F^2 \right) + \frac{5}{7} L^2 G^2 \eta_b^2,
$$

where the second inequality is due to $\| a + b \|^2 \leq 2 \| a \|^2 + 2 \| b \|^2$.

Next, we bound the term $\mathbb{E}_{\mathcal{C}} \left[ \left\| X(b+1) - \hat{X}(b+1) \right\|_F^2 \right]$. As for the repeated compressor, we have $\mathbb{E}_{\mathcal{C}} \left[ \| \mathcal{C}_{L_2}(\mathbf{x}) - \mathbf{x} \|^2 \right] \leq (1 - \omega)^{L_2} \| \mathbf{x} \|^2$. By setting $L_2 = \lceil \frac{\ln(8n)}{\omega} \rceil$, we have $(1 - \omega)^{L_2} \leq \frac{1}{8n}$, which means $\mathbb{E}_{\mathcal{C}} \left[ \| \mathcal{C}_{L_2}(\mathbf{x}) - \mathbf{x} \|^2 \right] \leq \frac{1}{8n} \| \mathbf{x} \|^2$.

$$
\mathbb{E}_{\mathcal{C}} \left[ \left\| X(b+1) - \hat{X}(b+1) \right\|_F^2 \right]
$$

$$
= \sum_{i=1}^{n} \mathbb{E}_{\mathcal{C}} \left[ \| \mathbf{x}_i(b+1) - \hat{\mathbf{x}}_i(b+1) \|^2 \right]
$$

$$
= \sum_{i=1}^{n} \mathbb{E}_{\mathcal{C}} \left[ \left\| \mathbf{y}_i^{(L_1+1)}(b) + \mathbf{r}_i(b+1) - \hat{\mathbf{y}}_i^{(L_1+1)}(b) - \mathbf{r}_i^{\mathcal{C}}(b+1) \right\|^2 \right]
$$

$$
= 2 \sum_{i=1}^{n} \mathbb{E}_{\mathcal{C}} \left[ \left\| \mathbf{y}_i^{(L_1+1)}(b) - \hat{\mathbf{y}}_i^{(L_1+1)}(b) \right\|^2 \right] + 2 \mathbb{E}_{\mathcal{C}} \left[ \sum_{i=1}^{n} \left\| \mathbf{r}_i(b+1) - \mathbf{r}_i^{\mathcal{C}}(b+1) \right\|^2 \right]
$$

$$
\leq \frac{1}{7n} \mathbb{E}_{\mathcal{C}} \left[ \sum_{i=1}^{n} \left\| \mathbf{y}_i^{(1)}(b) - \hat{\mathbf{y}}_i^{(1)}(b) \right\|^2 + \left\| \mathbf{y}_i^{(1)}(b) - \overline{\mathbf{y}}_i^{(1)}(b) \right\|^2 \right] + \frac{1}{4n} \mathbb{E}_{\mathcal{C}} \left[ \sum_{i=1}^{n} \| \mathbf{r}_i(b+1) \|^2 \right]
$$

$$
= \frac{1}{7n} \mathbb{E}_{\mathcal{C}} \left[ \sum_{i=1}^{n} \| \mathbf{x}_i(b) - \eta_b \mathbf{z}_i(b-1) - \hat{\mathbf{x}}_i(b) \|^2 + \| \mathbf{x}_i(b) - \eta_b \mathbf{z}_i(b-1) - \overline{\mathbf{x}}(b) + \eta_b \overline{\mathbf{z}}(b-1) \|^2 \right]
$$

$$
+ \frac{1}{4n} \mathbb{E}_{\mathcal{C}} \left[ \sum_{i=1}^{n} \| \mathbf{r}_i(b+1) \|^2 \right]
$$

$$
\leq \frac{2}{7n} \mathbb{E}_{\mathcal{C}} \left[ \sum_{i=1}^{n} \| \mathbf{x}_i(b) - \hat{\mathbf{x}}_i(b) \|^2 + \| \mathbf{x}_i(b) - \overline{\mathbf{x}}(b) \|^2 \right] + \frac{1}{4n} \mathbb{E}_{\mathcal{C}} \left[ \| R(b+1) \|_F^2 \right] + \frac{5}{7} L^2 G^2 \eta_b^2
$$

$$
\leq \frac{5}{14n} \mathbb{E}_{\mathcal{C}} \left[ \sum_{i=1}^{n} \| \mathbf{x}_i(b) - \hat{\mathbf{x}}_i(b) \|^2 + \| \mathbf{x}_i(b) - \overline{\mathbf{x}}(b) \|^2 \right] + 2 L^2 G^2 \eta_b^2.
$$

$$\sum_{i=1}^{n} \mathbb{E}_{\mathcal{C}}\left[\|\mathbf{x}_i(b+1) - \hat{\mathbf{x}}_i(b+1)\|^2\right]$$

$$= \sum_{i=1}^{n} \mathbb{E}_{\mathcal{C}}\left[\left\|\mathbf{y}_i^{(L_1+1)}(b) + \mathbf{r}_i(b+1) - \hat{\mathbf{y}}_i^{(L_1+1)}(b)\right\|^2\right]$$

$$= 2\sum_{i=1}^{n} \mathbb{E}_{\mathcal{C}}\left[\left\|\mathbf{y}_i^{(L_1+1)}(b) - \hat{\mathbf{y}}_i^{(L_1+1)}(b)\right\|^2\right] + 2\mathbb{E}_{\mathcal{C}}\left[\sum_{i=1}^{n}\|\mathbf{r}_i(b+1)\|^2\right]$$

$$\leq \frac{1}{7n}\mathbb{E}_{\mathcal{C}}\left[\sum_{i=1}^{n}\left\|\mathbf{y}_i^{(1)}(b) - \hat{\mathbf{y}}_i^{(1)}(b)\right\|^2 + \left\|\mathbf{y}_i^{(1)}(b) - \overline{\mathbf{y}}_i^{(1)}(b)\right\|^2\right] + 2\mathbb{E}_{\mathcal{C}}\left[\sum_{i=1}^{n}\|\mathbf{r}_i(b+1)\|^2\right]$$

$$= \frac{1}{7n}\mathbb{E}_{\mathcal{C}}\left[\sum_{i=1}^{n}\|\mathbf{x}_i(b) - \eta_b\mathbf{z}_i(b-1) - \hat{\mathbf{x}}_i(b)\|^2 + \|\mathbf{x}_i(b) - \eta_b\mathbf{z}_i(b-1) - \overline{\mathbf{x}}(b) + \eta_b\overline{\mathbf{z}}(b-1)\|^2\right]$$

$$+ 2\mathbb{E}_{\mathcal{C}}\left[\sum_{i=1}^{n}\|\mathbf{r}_i(b+1)\|^2\right]$$

$$\leq \frac{2}{7n}\mathbb{E}_{\mathcal{C}}\left[\sum_{i=1}^{n}\|\mathbf{x}_i(b) - \hat{\mathbf{x}}_i(b)\|^2 + \|\mathbf{x}_i(b) - \overline{\mathbf{x}}(b)\|^2\right] + \mathbb{E}_{\mathcal{C}}\left[\sum_{i=1}^{n}\|\mathbf{r}(b+1)\|_F^2\right] + \frac{5}{7}L^2 G^2 \eta_b^2$$

$$\leq \frac{4}{7n}\mathbb{E}_{\mathcal{C}}\left[\sum_{i=1}^{n}\|\mathbf{x}_i(b) - \hat{\mathbf{x}}_i(b)\|^2 + \|\mathbf{x}_i(b) - \overline{\mathbf{x}}(b)\|^2\right] + \frac{24}{7}L^2 G^2 \eta_b^2 + 2L^2 G^2 \eta_b^2.$$

To demonstrate the importance of our projection error compensation scheme, we provide the results without this scheme. Concretely, without the projection error compensation scheme, we can derive

$$\sum_{i=1}^{n}\|\mathbf{x}_i(b+1) - \hat{\mathbf{x}}_i(b+1)\|^2 = \sum_{i=1}^{n}\left\|\mathbf{y}_i^{(L_1+1)}(b) + \mathbf{r}_i(b+1) - \hat{\mathbf{y}}_i^{(L_1+1)}(b)\right\|^2$$

$$\leq 2\sum_{i=1}^{n}\left\|\mathbf{y}_i^{(L_1+1)}(b) - \hat{\mathbf{y}}_i^{(L_1+1)}(b)\right\|^2 + 2\sum_{i=1}^{n}\|\mathbf{r}_i(b+1)\|^2$$

$$\leq \frac{2}{7n}\sum_{i=1}^{n}\left(\left\|\mathbf{y}_i^{(1)}(b) - \hat{\mathbf{y}}_i^{(1)}(b)\right\|^2 + \left\|\mathbf{y}_i^{(1)}(b) - \overline{\mathbf{y}}_i^{(1)}(b)\right\|^2\right) + 2\sum_{i=1}^{n}\|\mathbf{r}_i(b+1)\|^2$$

$$\leq \frac{6}{7n}\sum_{i=1}^{n}\left(\|\mathbf{x}_i(b) - \hat{\mathbf{x}}_i(b)\|^2 + \|\mathbf{x}_i(b) - \overline{\mathbf{x}}(b)\|^2\right) + O(n\eta_b^2 L^2 G^2).$$

As can be seen, it suffers an additional $O(n)$ dependence.

### E.3. Proof of Lemma D.3

The proof is similar to that of Lemma D.1, and the key difference is that we need to utilize the strong convexity. According to the proof of Lemma D.1, we first have the following

$$\|\overline{\mathbf{x}}(b+1) - \mathbf{x}\|^2 = \left\|\frac{1}{n}\sum_{i=1}^{n}\overline{\mathbf{x}}(b) - \mathbf{x}\right\|^2 + \frac{1}{n^2}\left\|\sum_{i=1}^{n}\mathbf{r}_i(b+1) - \eta_b\sum_{j=1}^{n}\mathbf{z}_j(b-1)\right\|^2$$

$$+ 2\left\langle\frac{1}{n}\sum_{i=1}^{n}\mathbf{r}_i(b+1), \overline{\mathbf{x}}(b) - \mathbf{x}\right\rangle - \frac{2\eta_b}{n}\sum_{j=1}^{n}\left\langle\sum_{t=(b-2)L+1}^{(b-1)L}\nabla f_{t,j}(\mathbf{x}_j(b-1)), \overline{\mathbf{x}}(b) - \mathbf{x}\right\rangle.$$

For the last term, we have

$$
-\frac{\eta_b}{n}\sum_{j=1}^{n}\sum_{t=(b-2)L+1}^{(b-1)L}\langle\nabla f_{t,j}(\mathbf{x}_j(b-1)),\overline{\mathbf{x}}(b)-\mathbf{x}\rangle
$$

$$
=-\frac{\eta_b}{n}\sum_{j=1}^{n}\sum_{t=(b-2)L+1}^{(b-1)L}\langle\nabla f_{t,j}(\mathbf{x}_j(b-1)),\overline{\mathbf{x}}(b)-\overline{\mathbf{x}}(b-1)+\overline{\mathbf{x}}(b-1)-\mathbf{x}\rangle
$$

$$
=-\frac{\eta_b}{n}\sum_{j=1}^{n}\sum_{t=(b-2)L+1}^{(b-1)L}\langle\nabla f_{t,j}(\mathbf{x}_j(b-1)),\overline{\mathbf{x}}(b)-\overline{\mathbf{x}}(b-1)\rangle-\frac{\eta_b}{n}\sum_{j=1}^{n}\sum_{t=(b-2)L+1}^{(b-1)L}\langle\nabla f_{t,j}(\mathbf{x}_j(b-1)),\overline{\mathbf{x}}(b-1)-\mathbf{x}\rangle.
$$

For the first term, we can directly use (26). For the second term, we have

$$
-\frac{\eta_b}{n}\sum_{j=1}^{n}\sum_{t=(b-2)L+1}^{(b-1)L}\langle\nabla f_{t,j}(\mathbf{x}_j(b-1)),\overline{\mathbf{x}}(b-1)-\mathbf{x}\rangle
$$

$$
=\frac{\eta_b}{n}\sum_{j=1}^{n}\sum_{t=(b-2)L+1}^{(b-1)L}\langle\nabla f_{t,j}(\mathbf{x}_j(b-1)),\mathbf{x}-\overline{\mathbf{x}}(b-1)\rangle
$$

$$
=\frac{\eta_b}{n}\sum_{j=1}^{n}\sum_{t=(b-2)L+1}^{(b-1)L}\langle\nabla f_{t,j}(\mathbf{x}_j(b-1)),\mathbf{x}-\mathbf{x}_j(b-1)\rangle+\frac{\eta_b}{n}\sum_{j=1}^{n}\sum_{t=(b-2)L+1}^{(b-1)L}\langle\nabla f_{t,j}(\mathbf{x}_j(b-1)),\mathbf{x}_j(b-1)-\overline{\mathbf{x}}(b-1)\rangle
$$

$$
\leq\frac{\eta_b}{n}\sum_{j=1}^{n}\sum_{t=(b-2)L+1}^{(b-1)L}f_{t,j}(\mathbf{x})-f_{t,j}(\mathbf{x}_j(b-1))-\frac{\mu}{2}\|\mathbf{x}-\mathbf{x}_j(b-1)\|^2+\frac{\eta_b}{n}\sum_{j=1}^{n}GL\|\mathbf{x}_j(b-1)-\overline{\mathbf{x}}(b-1)\|
$$

$$
=\frac{\eta_b}{n}\sum_{j=1}^{n}\sum_{t=(b-2)L+1}^{(b-1)L}f_{t,j}(\mathbf{x})-f_{t,j}(\mathbf{x}_i(b-1))+f_{t,j}(\mathbf{x}_i(b-1))-f_{t,j}(\mathbf{x}_j(b-1))
$$

$$
+\frac{\eta_b}{n}\sum_{j=1}^{n}GL\|\mathbf{x}_j(b-1)-\overline{\mathbf{x}}(b-1)\|-\frac{\mu L}{2}\|\mathbf{x}-\mathbf{x}_j(b-1)\|^2
$$

$$
\leq\frac{\eta_b}{n}\sum_{j=1}^{n}\sum_{t=(b-2)L+1}^{(b-1)L}f_{t,j}(\mathbf{x})-f_{t,j}(\mathbf{x}_i(b-1))+\frac{\eta_b}{n}GL\sum_{j=1}^{n}\|\mathbf{x}_i(b-1)-\mathbf{x}_j(b-1)\|
$$

$$
+\frac{\eta_b}{n}\left(GL\sum_{j=1}^{n}\|\mathbf{x}_j(b-1)-\overline{\mathbf{x}}(b-1)\|-\frac{\mu L}{2}\|\mathbf{x}-\mathbf{x}_j(b-1)\|^2\right),
$$

where the first inequality is due to the strong convexity. By using the fact that

$$
\sum_{j=1}^{n}\|\mathbf{x}-\mathbf{x}_j(b-1)\|^2\geq\frac{1}{n}\left\|\sum_{j=1}^{n}\mathbf{x}-\mathbf{x}_j(b-1)\right\|^2\geq\frac{1}{n}\|n\mathbf{x}-n\overline{\mathbf{x}}(b-1)\|^2\geq n\|\mathbf{x}-\overline{\mathbf{x}}(b-1)\|^2,
$$

and we have

$$-\frac{\eta_b}{n}\sum_{j=1}^{n}\sum_{t=(b-2)L+1}^{(b-1)L}\langle\nabla f_{t,j}(\mathbf{x}_j(b-1)),\overline{\mathbf{x}}(b-1)-\mathbf{x}\rangle$$

$$\leq\frac{\eta_b}{n}\sum_{j=1}^{n}\sum_{t=(b-2)L+1}^{(b-1)L}f_{t,j}(\mathbf{x})-f_{t,j}(\mathbf{x}_i(b-1))+\frac{\eta_b}{n}GL\sum_{j=1}^{n}\|\mathbf{x}_i(b-1)-\mathbf{x}_j(b-1)\|$$

$$\frac{\eta_b}{n}GL\sum_{j=1}^{n}\|\mathbf{x}_j(b-1)-\overline{\mathbf{x}}(b-1)\|-\frac{\mu L}{2}\|\mathbf{x}-\mathbf{x}_j(b-1)\|^2 \tag{31}$$

$$\leq\frac{\eta_b}{n}\sum_{j=1}^{n}\sum_{t=(b-2)L+1}^{(b-1)L}f_{t,j}(\mathbf{x})-f_{t,j}(\mathbf{x}_i(b-1))+\frac{\eta_b}{n}3nGL\left\|X(b-1)-\overline{X}(b-1)\right\|_F-\frac{\eta_b\mu L}{2}\|\mathbf{x}-\overline{\mathbf{x}}(b-1)\|^2.$$

By combining (24), (25), (26) and (31), we can derive

$$\|\overline{\mathbf{x}}(b+1)-\mathbf{x}\|^2=\|\overline{\mathbf{x}}(b)-\mathbf{x}\|^2-\eta_b\mu L\|\mathbf{x}-\overline{\mathbf{x}}(b-1)\|^2+3L^2\eta_b^2G^2+2\eta_b\eta_{b-1}L^2G^2$$

$$+\frac{3}{n}\|R(b+1)\|_F^2+\frac{1}{n}\left\|X(b)-\tilde{X}(b+1)\right\|_F+\frac{1}{n}\|R(b)\|_F^2$$

$$+\frac{2\eta_b}{n}\sum_{j=1}^{n}\sum_{t=(b-2)L+1}^{(b-1)L}f_{t,j}(\mathbf{x})-f_{t,j}(\mathbf{x}_i(b-1))+\frac{2\eta_b}{n}3nGL\left\|X(b-1)-\overline{X}(b-1)\right\|_F,$$

which implies

$$\sum_{t=(b-2)L+1}^{(b-1)L}\sum_{j=1}^{n}f_{t,j}(\mathbf{x}_i(b-1))-f_{t,j}(\mathbf{x})$$

$$\leq\frac{n}{2\eta_b}(\|\overline{\mathbf{x}}(b)-\mathbf{x}\|^2-\|\overline{\mathbf{x}}(b+1)-\mathbf{x}\|^2)-\frac{n\mu L}{2\eta_b}\|\mathbf{x}-\overline{\mathbf{x}}(b-1)\|^2+\frac{3}{2\eta_b}\|R(b+1)\|_F^2+\frac{3}{2}L^2n\eta_bG^2+L^2n\eta_{b-1}G^2$$

$$+\frac{1}{2\eta_b}\left\|X(b)-\tilde{X}(b+1)\right\|_F^2+3nGL\left\|X(b-1)-\overline{X}(b-1)\right\|_F+\frac{1}{2\eta_b}\|R(b)\|_F^2.$$

By summing up over all blocks, we can derive

$$\mathbb{E}_{\mathcal{C}}\left[R(T,i)\right]=\sum_{b=1}^{T/L}\sum_{t=(b-1)L+1}^{bL}\sum_{j=1}^{n}f_{t,j}(\mathbf{x}_i(b))-\sum_{t=1}^{T}\sum_{j=1}^{n}f_{t,j}(\mathbf{x})$$

$$\leq\frac{nD^2}{2}\sum_{b=1}^{T/L}(\frac{1}{\eta_b}-\frac{1}{\eta_{b-1}}-\mu L)+3L^2G^2n\sum_{b=1}^{T/L}\eta_b+3nGL\sum_{b=1}^{T/L}\mathbb{E}_{\mathcal{C}}\left[\left\|X(b)-\overline{X}(b)\right\|_F\right] \tag{32}$$

$$+\sum_{b=1}^{T/L}\frac{3}{2\eta_b}\mathbb{E}_{\mathcal{C}}\left[\|R(b+1)\|_F^2\right]+\frac{1}{2\eta_b}\mathbb{E}_{\mathcal{C}}\left[\left\|X(b)-\tilde{X}(b+1)\right\|_F^2\right]+\frac{1}{2\eta_b}\mathbb{E}_{\mathcal{C}}\left[\|R(b)\|_F^2\right].$$

### E.4. Proof of Lemma E.1

The efficient implementation of Choco-gossip is summarized in Algorithm 7, where each learner $i$ only needs to maintain three additional variables.

In the following, we give the proof for Lemma E.1. First, we provide its matrix version of Choco-gossip in Algorithm 8 to simplify our proof. The proof of this lemma is based on the analysis of Koloskova et al. (2019). The key difference is that we choose a different $\gamma$ to obtain a tighter guarantee. We introduce the following lemma

**Lemma E.2.** *(Lemma 16 in Koloskova et al. (2019)) For $P$ satisfying Assumption 3.1 and $t\in\mathbb{N}_+$, we have*

$$\left\|P^t-\frac{1}{n}\mathbf{I}\mathbf{I}^\top\right\|_2\leq(1-\rho)^t.$$

---

**Algorithm 7** Efficient Choco-gossip

---

1: **Input:** communication round $L, \mathbf{x}_i(1) \in \mathbb{R}^d$ for $i \in [n], \hat{\mathbf{x}}_i(1) = \mathbf{0}$ for $i \in [n]$
2: **for** learner $i \in [n]$ **do**
3:     **for** $t = 1$ to $L$ **do**
4:         $\mathbf{x}_i(t+1) = \mathbf{x}_i(k) + \gamma \left( \mathbf{s}_i(t) - \hat{\mathbf{x}}_i(t) \right)$
5:         $\mathbf{q}_i(t) = \mathcal{C}(\mathbf{x}_i(t+1) - \hat{\mathbf{x}}_i(t))$
6:         Send $\mathbf{q}_i(t)$ and receive $\mathbf{q}_j(t)$
7:         $\hat{\mathbf{x}}_i(t+1) = \hat{\mathbf{x}}_i(t) + \mathbf{q}_i(t)$
8:         $\mathbf{s}_i(t+1) = \mathbf{s}_i(t) + \sum_{j \in \mathcal{N}_i} P_{ij} \mathbf{q}_j(t)$
9:     **end for**
10: **end for**

---

**Algorithm 8** Choco-gossip (matrix version)

---

1: **Input:** Communication round $L, \mathbf{x}_i(1) \in \mathbb{R}^d$ for $i \in [n], \hat{\mathbf{x}}_i(1) = \mathbf{0}$ for $i \in [n]$
2: **for** learner $i \in [n]$ **do**
3:     **for** $t = 1$ to $L$ **do**
4:         $X(t+1) = X(t) + \gamma \hat{X}(t)(P - I)$
5:         $Q(t) = \mathcal{C}(X(t+1) - \hat{X}(t))$
6:         $\hat{X}_i(t+1) = \hat{X}(t) + Q(t)$
7:     **end for**
8: **end for**

---

Since the variable $\hat{\mathbf{x}}_i(t)$ is same over all neighbors $j \in \mathcal{N}_i$, we have $\sum_{i=1}^n \sum_{j \in \mathcal{N}_i} P_{ij}(\hat{\mathbf{x}}_j(t) - \hat{\mathbf{x}}_i(t)) = \mathbf{0}$. During iterations of the Algorithm 8, we can derive

$$\overline{\mathbf{x}}(t+1) = \overline{\mathbf{x}}(t) + \gamma \frac{1}{n} \sum_{i=1}^n \sum_{j \in \mathcal{N}_i} P_{ij}(\hat{\mathbf{x}}_j(t) - \hat{\mathbf{x}}_i(t)) = \overline{\mathbf{x}}(t),$$

which means the average decision is same over all rounds. We denote $\overline{X} = \overline{X}(1) = \cdots = \overline{X}(L_1)$ and adopt a lemma from (Koloskova et al., 2019).

**Lemma E.3** (Lemma 17 and Lemma 18 in (Koloskova et al., 2019)). *For P satisfying Assumption 3.1 and during iterations of the Algorithm 8, we have*

$$\left\| X(t+1) - \overline{X} \right\|_F^2 \leq \left(1 + \frac{\gamma \rho}{2}\right)(1 - \gamma \rho)^2 \left\| X(t) - \overline{X} \right\|_F + \left(1 + \frac{2}{\gamma \rho}\right) \gamma^2 \beta^2 \left\| \hat{X}(t) - X(t) \right\|_F^2,$$

$$\mathbb{E}_{\mathcal{C}} \left[ \left\| X(t+1) - \hat{X}(t+1) \right\|_F^2 \right] \leq \left(1 + \frac{\omega}{2}\right)(1 - \omega)(1 + \gamma \beta)^2 \mathbb{E}_{\mathcal{C}} \left[ \left\| X(t) - \hat{X}(t) \right\|_F^2 \right]$$
$$+ \left(1 + \frac{2}{\omega}\right)(1 - \omega) \gamma^2 \beta^2 \mathbb{E}_{\mathcal{C}} \left[ \left\| X(t) - \overline{X} \right\|_F^2 \right].$$

We define

$$e_{t+1} = \mathbb{E}_{\mathcal{C}} \left[ \left\| X(t+1) - \hat{X}(t+1) \right\|_F^2 + \left\| X(t+1) - \overline{X} \right\|_F^2 \right]$$
$$= \mathbb{E}_{\mathcal{C}} \left[ \sum_{i=1}^n \left\| \mathbf{x}_i(t+1) - \hat{\mathbf{x}}_i(t+1) \right\|^2 + \left\| \mathbf{x}_i(t+1) - \overline{\mathbf{x}} \right\|^2 \right].$$

We further have

$$e_{t+1} \leq \max\{(1 + \frac{\gamma \rho}{2})(1 - \gamma \rho)^2 + (1 - \omega)(1 + \frac{2}{\omega})\gamma^2 \beta^2, (1 + \frac{2}{\gamma \rho})\gamma^2 \beta^2 + (1 - \omega)(1 + \frac{\omega}{2})(1 + \gamma \beta)^2\} e_t.$$

We want to select an appropriate $\gamma$, which satisfies

$$e_{t+1} \leq (1 - \frac{\rho\gamma}{2})e_t.$$

We have to ensure

$$(1 + \frac{\gamma\rho}{2})(1 - \gamma\rho)^2 + (1 - \omega)(1 + \frac{2}{\omega})\gamma^2\beta^2 \leq 1 - \frac{\rho}{2}\gamma, \tag{33}$$

$$(1 + \frac{2}{\gamma\rho})\gamma^2\beta^2 + (1 - \omega)(1 + \frac{\omega}{2})(1 + \gamma\beta)^2 \leq 1 - \frac{\rho}{2}\gamma. \tag{34}$$

According to inequality (33), we have

$$\gamma \leq \frac{2\omega\rho}{8\beta^2 + \omega\rho^2}.$$

According to inequality (34), we have

$$\gamma \leq \frac{\omega\rho}{2\rho\beta^2 + 4\beta^2 + (2 - \omega)(\beta^2 + 2\beta)\rho + \rho^2}.$$

Therefore, we choose $\gamma = \frac{\omega\rho}{2\rho\beta^2 + 4\beta^2 + (2-\omega)(\beta^2+2\beta)\rho + \rho^2}$ and have

$$e_{t+1} \leq (1 - \frac{\gamma\rho}{2})e_t \leq (1 - \frac{\gamma\rho}{2})^t e_1.$$

By setting block size $L = \lceil \frac{2\ln(14n)}{\gamma\rho} \rceil$, we have $e_{L+1} \leq \left(1 - \frac{\gamma\rho}{2}\right)^{\lceil \frac{2\ln(14n)}{\gamma\rho} \rceil} \leq \frac{1}{14n}e_1$.

### E.5. Proof of Lemma D.4

We prove this lemma by induction.

(i) When $b = 1$, this inequality holds.

Suppose that the statement holds for $k$, i.e.,

$$e_{k+1} \leq 4q\eta_k^2 L^2.$$

Then for $k + 1$,

$$e_{k+2} \leq \frac{1}{2n}e_{k+1} + q\eta_{k+1}^2 L^2$$
$$\leq \frac{2}{n}qL^2\eta_k^2 L^2 + q\eta_{k+1}^2 L^2.$$

Then we need to prove that

$$\frac{2}{n}q\eta_k^2 L^2 + q\eta_{k+1}^2 L^2 \leq 3qL^2\eta_{k+1}^2,$$

which is equivalent to proving

$$\frac{\eta_{k+1}^2}{\eta_k^2} \leq 1.$$

As $\eta_{k+1} \leq \eta_k$, this inequality holds. We finish the proof.

## F. Refined Analysis of Tu et al. (2022)

In the proof of Tu et al. (2022), they need to bound the compression error and the consensus error terms:

$$e_t = \mathbb{E}_\mathcal{C}[\|X(t+1) - \overline{X}(t+1)\|_F^2] + \mathbb{E}_\mathcal{C}[\|X(t+1) - \hat{X}(t+1)\|_F^2].$$

According to the proof of Tu et al. (2022), they derive

$$
\begin{aligned}
e_{t+1} &\le \|U(\gamma)\| \, e_t + C_3 \gamma^{-1} \rho^{-1} n \eta_t^2 \\
&\le \lambda_{\max}(U(\gamma)) e_t + C_3 \gamma^{-1} \rho^{-1} n \eta_t^2,
\end{aligned}
$$

where $C_3$ is a constant,

$$
U(\gamma) = \begin{bmatrix} 1 - \rho\gamma & u_1\gamma \\ u_2\gamma^2 & 1 - \frac{\omega}{2} - \frac{\omega^2}{2} + u_3\gamma \end{bmatrix},
$$

and $u_1 = 9\left(1 + \frac{2}{\rho}\right)(1-\omega)\beta^2$, $u_2 = 3\left(1 + \frac{2}{\omega}\right)\beta^2$, $u_3 = \left(1 + \frac{\omega}{2}\right)(1-\omega)(\beta^2 + 2\beta) + 6\left(1 + \frac{2}{\omega}\right)(1-\omega)\beta^2$.

However, their use of the inequality is incorrect due to $\lambda_{\max}(U(\gamma)) \le \|U(\gamma)\|$.

We give a correct proof in the following.

It is not hard to derive $e_t = (1 - \rho\gamma + u_2\gamma^2)\mathbb{E}_\mathcal{C}[\|X(t+1) - \overline{X}(t+1)\|_F^2] + (u_1\gamma + u_3\gamma + 1 - \frac{\omega}{2} - \frac{\omega^2}{2})\mathbb{E}_\mathcal{C}[\|X(t+1) - \hat{X}(t+1)\|_F^2] + C_3\gamma^{-1}\rho^{-1}n\eta_t^2$.

We need to choose $\gamma$ that ensures $\max\{(1 - \rho\gamma + u_2\gamma^2), (u_1\gamma + u_3\gamma + 1 - \frac{\omega}{2} - \frac{\omega^2}{2})\} \le 1 - \frac{3}{4}\gamma\rho$, which means

$$
\begin{aligned}
1 - \rho\gamma + u_2\gamma^2 &\le 1 - \frac{3}{4}\gamma\rho, \\
u_1\gamma + u_3\gamma + 1 - \frac{\omega}{2} - \frac{\omega^2}{2} &\le 1 - \frac{3}{4}\gamma\rho.
\end{aligned}
\tag{35}
$$

Based on (35), we can derive

$$
\begin{aligned}
\gamma &\le \frac{\rho}{4u_2}, \\
\gamma &\le \frac{\omega + \omega^2}{2(\rho + u_1 + u_3)}.
\end{aligned}
$$

Therefore, we select $\gamma \le \frac{\rho(\omega^2 + \omega)}{2(\rho + u_1 + 4u_2 + u_3)}$, because $\omega^2 + \omega \le 2$.

We choose $\gamma = \frac{3\rho^3\omega^2(\omega+1)}{2\rho^2\omega + 9\beta^2(\rho+2)(\omega-\omega^2) + 24\beta^2(\omega+2) + \omega(\omega+2)(1-\omega)(\beta^2+2\beta) + 12\beta^2(\omega+2)(1-\omega)} < 1$.

And we have $\gamma^{-1} \le O(\omega^{-2}\rho^{-3})$, which is of the same order as the result in Tu et al. (2022).

## G. Experiments

In this section, we assess the performance of the proposed methods on an online classification task. We compare the proposed algorithms Top-DOGD, Top-DOBD-1, and Top-DOBD-2 with the existing methods DC-DOGD, DC-DOBD, and DC-DOBD-2 (Tu et al., 2022).

**Setup.** We perform online classification to evaluate the performance of our method on the ijcnn1, SUSY and a9a datasets from LIBSVM (Chang & Lin, 2011). In each round $t \in [T]$, a batch of training examples $\{(\mathbf{w}_{t,1}, y_{t,1}), \ldots, (\mathbf{w}_{t,m}, y_{t,m})\}$ arrives, where $(\mathbf{w}_{t,k}, y_{t,k}) \in [-1,1]^d \times \{-1,1\}, k = 1, \ldots, m$. Each online learner $i$ aims to predict a linear model $\mathbf{x}_i(t)$ and suffers a local loss. We consider two types of convex functions. For general convex functions, learner $i$ suffers the logistic loss $f_{t,i}(\mathbf{x}) = \frac{1}{m_i}\sum_{k=1}^{m_i} \ln(1 + \exp(-y_{t,k}\mathbf{x}^\top\mathbf{w}_{t,k}))$, and for strongly convex functions, learner $i$ suffers the regularized hinge loss $f_{t,i}(\mathbf{x}) = \frac{1}{m_i}\sum_{k=1}^{m_i} \max\{0, 1 - y_{t,k}\mathbf{x}^\top\mathbf{w}_{t,k}\} + \frac{\mu}{2}\|\mathbf{x}\|^2$, where $\mu$ is the regularization parameter, $m_i$ is the data available to learner $i$ and $\sum_{i=1}^n m_i = m$. The connected communication network $\mathcal{G}([n], E)$ with $n$ nodes

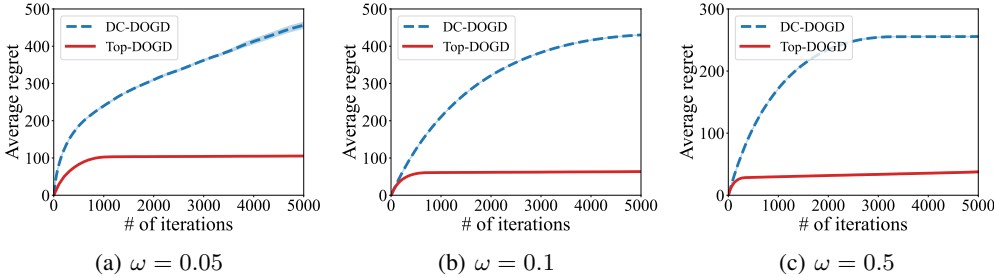

*Figure 1.* Results for convex functions with 16 nodes under the full-information setting.

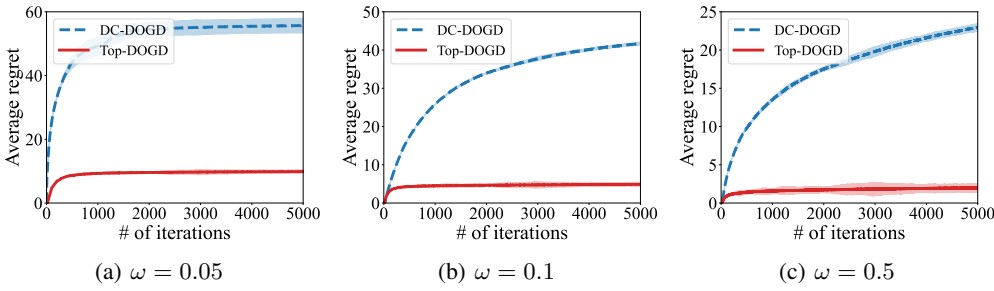

*Figure 2.* Results for strongly convex functions with 16 nodes under the full-information setting.

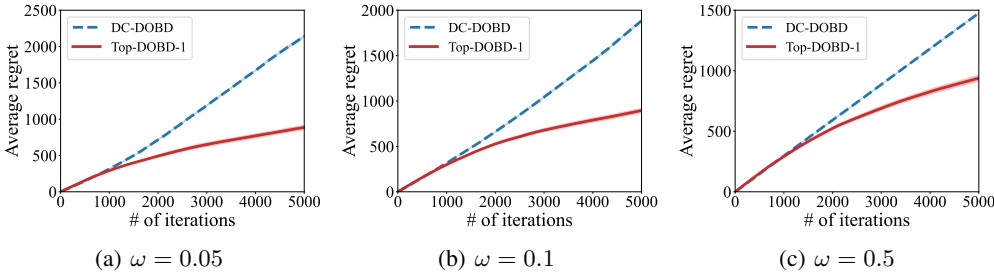

*Figure 3.* Results for convex functions with 16 nodes under the one-point bandit setting.

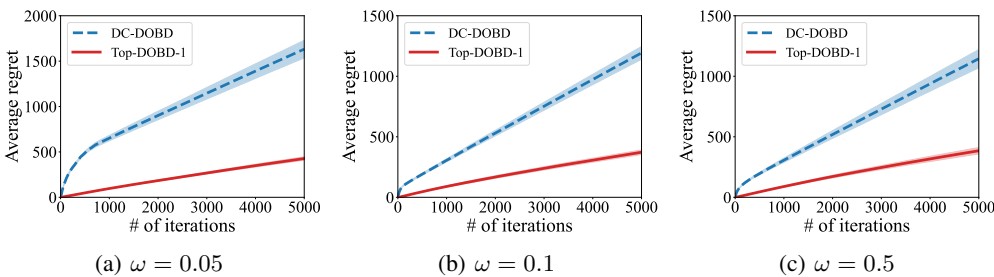

*Figure 4.* Results for strongly convex functions with 16 nodes under the one-point bandit setting.

and $|E|$ edges is generated randomly by the tool NetworkX (Hagberg et al., 2008), and then we use the Metropolis rule (Xiao & Boyd, 2004) to construct the connectivity matrix $P$. We set $T = 5000, m = 48$ and $\mu = 0.1$ for the strongly convex setting, and $n = 16, |E| = 24$ on the ijcnn1 dataset. We conduct the experiments with 3 types of compressors with $\omega = 0.05, 0.1, 0.5$. We also perform experiments on the a9a dataset under the full-information setting, and the SUSY dataset

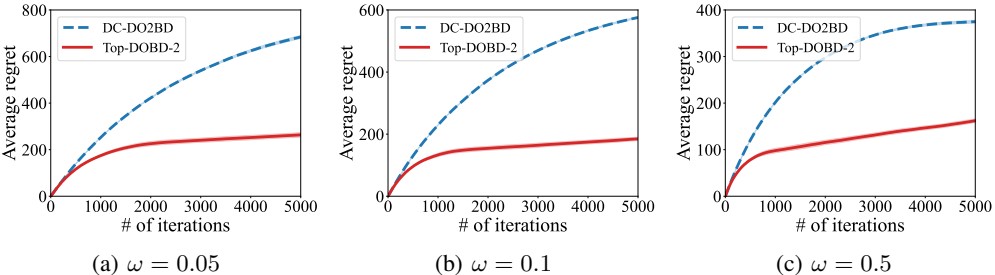

*Figure 5.* Results for convex functions with 16 nodes under the two-point bandit setting.

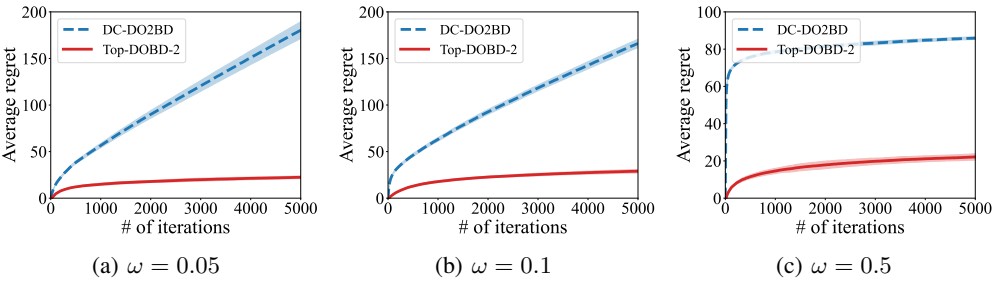

*Figure 6.* Results for strongly convex functions with 16 nodes under the two-point bandit setting.

under both one-point bandit feedback and two-point bandit feedback settings. We set $T = 2000, m = 24$ and $\mu = 0.1$ for the strongly convex setting. For the network topology, we employ a graph with $24$ nodes and $48$ edges.

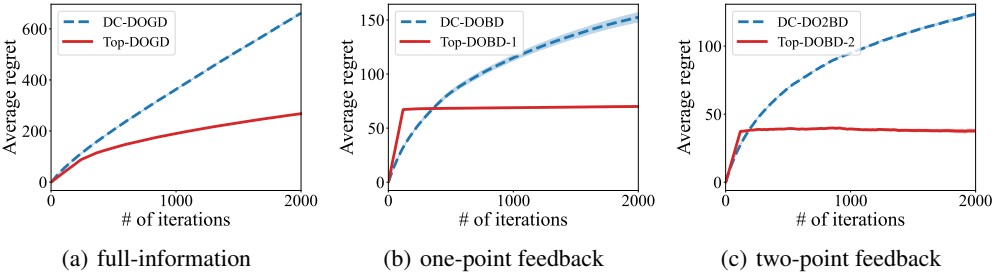

*Figure 7.* Results for convex functions with 24 nodes.

In our experiments, we adopt an equivalent implementation of Top-DOGD that uses the repeated compressor $\mathcal{C}_L(\cdot)$ with $L = \lceil 1/\omega \rceil$. In this case, $\mathcal{C}_L(\cdot)$ can be viewed as a single compressor with a compression ratio $\omega' = 1 - 1/e$. Accordingly, we set $L_1 = \lceil 1/\omega \rceil L_1'$, $L_1' = \lceil \frac{2\ln(14n)}{\gamma\rho} \rceil$, $L_2 = \lceil \frac{1}{\omega} \rceil$. It is not hard to verify that these two versions obtain the same regret bounds.

**Results.** We repeat each experiment 5 times and plot the mean curve with error bars. We record the average regret $\frac{1}{n} \sum_{i=1}^{n} R(T, i)$. As demonstrated in Figures 1, 2, 3, 4, 5, 6, 7 and 8, it is evident that our methods suffer less regret, outperforming the approaches in Tu et al. (2022). Moreover, we also observe that our methods exhibit increasing improvement over the existing methods as the compression ratio $\omega$ decreases, which aligns with the theoretical guarantees. Owing to the advantage in the dependence on $\omega$, our methods are more robust compared to those of Tu et al. (2022).

**Communication efficiency.** We present instantaneous loss versus transmitted bits, comparing our algorithm against the uncompressed optimal method, AD-FTAL (Wan et al., 2025). We conduct the experiments on the ijcnn1 dataset and set $T = 5000, m = 48$, $n = 16$ and $|E| = 24$. The results presented in Figure 9 demonstrate that our algorithm has better

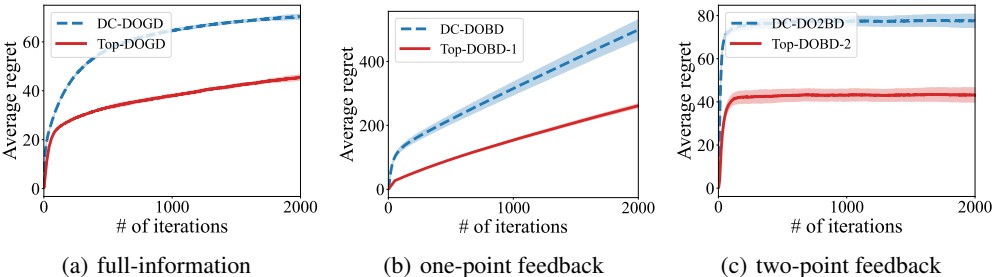

(a) full-information      (b) one-point feedback      (c) two-point feedback

*Figure 8.* Results for strongly convex functions with 24 nodes.

performance under the same communication budget.

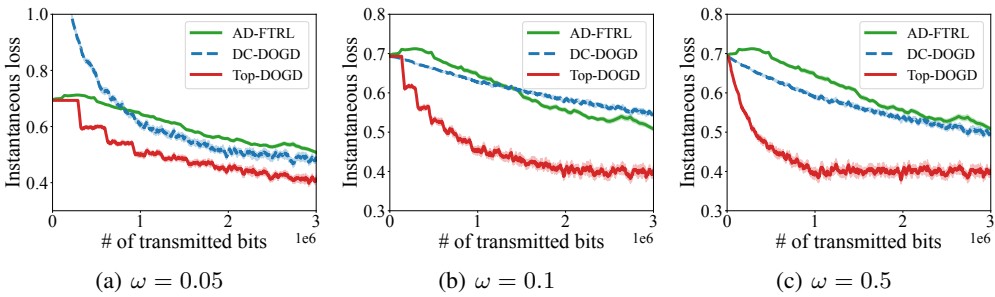

(a) $\omega = 0.05$      (b) $\omega = 0.1$      (c) $\omega = 0.5$

*Figure 9.* Instantaneous loss on the ijcnn1 dataset with 16 nodes under the full-information setting.

