# OpenReview forum: "Decentralized Online Convex Optimization with Efficient Communication: Improved Algorithm and Lower Bounds"
_ICML.cc/2026/Conference — ICML 2026 regular_

### Official Review · Reviewer_4ZcT · 2026-03-05

**Soundness:** 4
**Presentation:** 4
**Significance:** 3
**Originality:** 2
**Overall Recommendation:** 5
**Confidence:** 2

**Summary:**

The paper considers a distributed online convex optimization problem in which a communication constraint is modeled by a compressor with contraction factor $\omega$. This work provides state-of-the-art theoretical guarantees for convex and strongly convex distributed online optimization with nearly matching lower bounds up to the spectral difference of the communication graph. The novelty of the work lies in a more efficient communication schedule and in deriving the lower bound.

**Compliance With Llm Reviewing Policy:**

Affirmed.

**Final Justification:**

I believe it's a significant paper and should be accepted.

**Key Questions For Authors:**

(1) Although I understand it is not a fault of this work, but rather an adopted metric of the surrounding literature, I do not understand why the total number of exchanged bits is not considered as a communication constraint. I get that using the compressor imposes a bound on the size of the message, but it does not say anything about the frequency of the said messages. If I am not mistaken, the number of exchanged bits in this work is still linear in time ($O(T/L)$ to be precise). Is there a strand of work that explores D-OCO with bit-level communication constraints?

**Limitations:**

Yes

**Strengths And Weaknesses:**

Soundness: The paper appears sound on a high level, although I did not check  the details thoroughly

Presentation: The paper is quite technical; however, I did not find it too dense or uninteresting, even though I am not an expert on the subject matter. The introduction of the related works and the positioning of the paper in the literature are good and provide reasonable  insight into the difficulty of the problem at hand. The ideas/ motivations for the algorithm design are also very well explained. Lower bound argument is reasonably fleshed out in the main text and not overstated; it follows the previous communication-agnostic literature with a probabilistic argument to account for the compression constraint.

Significance: The paper is significant in that it provides a new theoretical SOTA on a relevant, well-studied problem. Almost matching lower bounds are also derived in the work.

Originality: Although most ideas and approaches in the work are adaptations from elsewhere in the related literature, the authors does not obfuscate this fact. The originality lies in their combination and appropriate algorithm design to fit the new setting.

---

> ### Author Rebuttal · Authors · 2026-03-28
>
> Many thanks for the constructive reviews!
>
> ---
>
> > __Q1.__ I get that using the compressor imposes a bound on the size of the message, but it does not say anything about the frequency of the sent messages.
>
> __A1.__ Our work does not restrict the frequency of communication: each learner sends a compressed messages to its neighbors at every round. The compressor reduces the per-round bit cost compared to uncompressed communication.
>
> ---
>
> > __Q2.__ Is there a strand of work that explores D-OCO with bit-level communication constraints?
>
> __A2.__ Regarding communication frequency, we note that the works [1, 2] study projection-free decentralized online convex optimization. They reduce the number of communication rounds from $O(T)$ to $O(\sqrt{T})$ while preserving the same $O(T^{3/4})$ regret bound as standard projection-free methods with full communication.
>
> Moreover, the work [3] investigates D-OCO with a constraint of $C$ communication rounds and establishes an $\Omega(n\rho^{-1/4}T/\sqrt{C})$ lower bound for convex loss functions. Focusing on the $T$-dependence, any algorithm achieving an $O(\sqrt{T})$ regret bound must require $O(T)$ communication rounds. Thus, the $O(T)$ communication rounds of our method are optimal with respect to $T$.
>
> To the best of our knowledge, D-OCO under a strict total bit budget has not yet been studied, and we believe this is a meaningful direction for future work.
>
> ---
>
> We hope the above responses address your concerns and are happy to provide further clarification if needed.
>
> ---
>
> __References__
>
> [1] Projection-free Distributed Online Convex Optimization with $O(\sqrt{T})$ Communication Complexity. ICML, 2020.
>
> [2] Projection-free Distributed Online Learning with Sublinear Communication Complexity. JMLR, 2022.
>
> [3] Optimal and Efficient Algorithms for Decentralized Online Convex Optimization. JMLR, 2025.

---

> > ### Author Rebuttal · Reviewer_4ZcT · 2026-04-01
> >
> > I believe it is a significant paper and I vote for its acceptance.

---

> > > ### Author Response · Authors · 2026-04-01
> > >
> > > Dear Reviewer 4ZcT,
> > >
> > > We are glad that our responses have fully addressed your concerns, and we will continue to refine our paper based on your valuable suggestions.
> > >
> > > Best regards,
> > >
> > > Authors

---

### Official Review · Reviewer_2FBb · 2026-03-12

**Soundness:** 3
**Presentation:** 2
**Significance:** 2
**Originality:** 3
**Overall Recommendation:** 4
**Confidence:** 4

**Summary:**

This paper proposes a novel decentralized online convex optimization algorithm incorporating compressed communication mechanisms. Specifically, the algorithm employs a dual-layer block update framework that integrates an online compressed gossip strategy with projection error compensation, aiming to improve existing regret bounds. Notably, this work derives, for the first time, the regret lower bounds for compressed distributed online convex optimization. Furthermore, the authors extend the algorithm to the bandit feedback setting.

**Compliance With Llm Reviewing Policy:**

Affirmed.

**Final Justification:**

Thank the authors for providing additional experiments and relevant literature support. The authors' detailed response has resolved my concerns about this paper, so I will continue to give positive feedback.

**Key Questions For Authors:**

1.	What are the specific values used for the block $ L_1,L_2 $ in the experiments? How sensitive is the algorithm's performance to these hyperparameters?\
2.	Please discuss the similarities and differences between your method and the approach presented by Cao & Basar (Automatica), 2023.

**Limitations:**

The scope of this paper is currently limited to the convex optimization domain. While this has direct application value in areas like distributed sensor networks or certain linear model training, it restricts the algorithm's direct applicability to mainstream non-convex tasks such as deep learning. In large-scale distributed training scenarios, non-convexity, over-parameterization, and the complex geometric structure of loss functions could significantly alter the propagation dynamics of compression errors and noise. Therefore, the effectiveness of the proposed algorithm when handling non-convex objective functions remains to be further verified

**Strengths And Weaknesses:**

Strengths:\
1)Strong Theoretical Innovation: The paper presents a significant theoretical contribution by deriving the regret lower bounds for compressed distributed online convex optimization for the first time, filling a critical gap in the current literature.\
2)Ingenious Algorithm Design: The proposed dual-layer block update framework, which combines an online compressed gossip strategy with projection error compensation, offers a novel and effective approach for addressing distributed optimization problems under communication constraints.\
3)Broad Scenario Extensibility: The authors successfully extend the algorithm to the bandit feedback setting and theoretically demonstrate that its performance surpasses existing baseline methods.\
Weakness:\
1. Insufficient Experimental Evaluation:\
Lack of Diversity: The datasets and network topologies used in the appendix lack diversity. \
Missing Parameter Details: The experiments in the supplementary material do not specify the block $ L_1,L_2 $ size, nor do they include an analysis of the algorithm's sensitivity to these parameters.\
Incomplete Metrics: The current results only plot average regret versus the number of iterations. The paper fails to include plots of average regret versus communication cost (in bits), which are essential to demonstrate the actual communication efficiency of the proposed method.\
2. Inadequate Comparison with Related Work: A paper [1] on decentralized online convex optimization was published in 2023 by Cao & Basar (Automatica). The author fails to discuss or compare the proposed method with this relevant work.\
3. Issues with Pseudocode Rigor:\
In Algorithm 2 (Line 12), the variable $y_j$ is not specified.\
In Algorithm 4 (Line 7), while the variable $ \triangle_{i}^{(2)}(b) $ is transmitted to neighbors, the algorithm flow fails to incorporate the received $ \triangle_{j}^{(2)}(b) $ from neighbors into the local update, leading to logical discontinuity.\
[1] Cao, X., and Basar, T. Decentralized online convex optimization with compressed communications. Automatica. 2023, 156: 111186.

---

> ### Author Rebuttal · Authors · 2026-03-28
>
> We sincerely thank you for your constructive feedback and your positive recognition of our paper.
>
> ---
>
> > __Q1.__ Insufficient Experimental Evaluation.
>
> __A1.__ Thanks for your suggestion. We have conducted additional experiments, which are reported at the following anonymous link: https://anonymous.4open.science/r/ICML_19912-rebuttal/results.pdf .
>
> - __Additional datasets.__ We perform additional experiments on the a9a dataset ($d=123$) under the full-information setting, and the SUSY dataset ($d=18$) under both one-point bandit feedback and two-point bandit feedback settings. We set $T=2000$ and $m=48$. For convex loss functions, we set the compression ratio $\omega=0.05$. For strongly convex loss functions, we set $\omega=0.5$ and $\mu=0.1$. For the network topology, we employ a graph with $n=24$ nodes and $|E|=48$ edges.
>
> - **Hyperparameter selection.** In our experiments, we determine $L_1$ via grid search because the theoretically prescribed constant is derived from worst-case bounds and may be conservative in practice. We set $L_1 = c\lceil 1/\omega \rceil$ and search $c$ over $\\{1,2,\ldots,10\\}$. For the convex setting, we set $L_1 = 100, 50, 20$  corresponding to $\omega = 0.05, 0.1, 0.5$, respectively. For the strongly convex setting, we set $L_1 = 100, 50, 10$, respectively. Regarding $L_2$, we observe that for the Rand-$k$ and Top-$k$ compressors, setting $L_2 = \lceil 1/\omega \rceil$ reduces the compression error to zero. Furthermore, when the projection error is zero, we bypass the projection error compensation strategy.
>
> - **Sensitivity analysis.** We present sensitivity experiments for $L_1$ and $L_2$. Regarding $L_1$, setting it to a larger value improves consensus but reduces the update frequency, leading to worse regret. Conversely, a smaller $L_1$ allows the algorithm to update more frequently but suffers from poor consensus. For $L_2$, we find that decreasing its value does not significantly degrade performance, primarily because the projection error is small compared to other error terms. Configuring a large $L_2$ results in a redundant block size, which worsens the regret.
>
> - **Communication efficiency.** We present instantaneous loss versus transmitted bits, comparing our algorithm against the uncompressed optimal method, AD-FTAL [1]. The results demonstrate that our algorithm has better performance under the same communication budget.
>
> ---
>
> > __Q2.__ Comparison with related work.
>
> __A2.__  [2] is a contemporaneous work with [3], and both adopt essentially the similar algorithmic design. They perform a single round of Choco gossip [4] per update. As a result, the algorithms of [2] and [3] inherit the same structure and enjoy the same order of regret guarantees.
>
> Our algorithm adopts a two-level blocking update framework that incorporates the online compressed gossip and projection error compensation strategies, which effectively control the consensus error, compression error, and projection error at the same time. We will add the comparison with [2] and explicitly clarify these distinctions in the revised manuscript.
>
> ---
>
> > __Q3.__ Issues with Pseudocode Rigor.
>
> __A3.__ Thank you for pointing out these issues. We will correct them and other typos in the revised version.
>
> ---
>
> We hope the above responses address your concerns and are happy to provide further clarification if needed.
>
> ---
>
> __References__
>
> [1] Nearly Optimal Regret for Decentralized Online Convex Optimization. COLT, 2024.
>
> [2] Decentralized Online Convex Optimization with Compressed Communications. Automatica, 2023.
>
> [3] Distributed Online Convex Optimization with Compressed Communication. NeurIPS, 2022.
>
> [4] Decentralized Stochastic Optimization and Gossip Algorithms with Compressed Communication. ICML, 2019.

---

> > ### Author Rebuttal · Reviewer_2FBb · 2026-04-01
> >
> > I thank the author for the detailed response, and all my concerns are addressed.

---

> > > ### Author Response · Authors · 2026-04-01
> > >
> > > Dear Reviewer 2FBb,
> > >
> > > We are pleased that our rebuttal has resolved your concerns. We will refine our paper according to your constructive comments!
> > >
> > > Best regards,
> > >
> > > Authors

---

### Official Review · Reviewer_xADF · 2026-03-13

**Soundness:** 4
**Presentation:** 2
**Significance:** 4
**Originality:** 3
**Overall Recommendation:** 5
**Confidence:** 4

**Summary:**

The authors propose a novel algorithm for decentralized online convex optimization, in which agents use compressed communications. The key idea behind the algorithm is to break the total time horizon into blocks, each block broken itself into sub-blocks: the first sub-block runs a gossip-like algorithm to diffuse information across the network, and the second sub-block compensates for the projection error. Agents play the same action throughout each block, revising it only at the end based on the information meanwhile aggregated during the block. The authors prove that the new algorithm significantly outperforms the state-of-art by reducing the dependence on the compression factor and the spectral gap of the mixing matrix. The work also includes an extension to the bandit feedback scenario.

**Compliance With Llm Reviewing Policy:**

Affirmed.

**Final Justification:**

The authors fully addressed my concerns. Therefore I maintain the positive score of 5.

**Key Questions For Authors:**

Q1: The proposed Top-DOGD algorithm freezes the actions of the agents during each block. This feels inefficient, as actions are not revised as soon as new information arrives at the agent from the diffusion communication steps. Can the authors give an insight on why an algorithm architecture based on revising the plays at each time instant (so, in sync with the pace of arrival of fresh information) would not necessarily be better (say, either from a practical performance viewpoint or from a theoretical proof analysis viewpoint)?

Q2: Lines 250-254, "The projection operation in (3) introduces an additional error, which couples with the compression error and further induces an  O(n) dependence in the approximation error.": It's unclear to me how such coupling arises and also how it leads to the O(n) dependence. Can the authors elaborate on this point?

Q3: The proposed Top-DOGD is inspired from (3) which, for the static optimization setting (all functions f_{t,i} = f_i are fixed), is the blueprint of earlier distributed optimization algorithms. Because those algorithms require a vanishing stepsize \eta_t to converge to the optimal solution, and therefore become very slow, other blueprints for first-order methods were introduced in the distributed optimization community, e.g., one of the first ones being EXTRA, which enabled a fixed step-size and were much faster. Can the authors briefly comment or speculate on the possibility of starting instead from the EXTRA blueprint to obtain an online distributed convex optimization algorithm?

**Limitations:**

Yes.

**Strengths And Weaknesses:**

Strenghts:
- Novel design of a distributed algorithm for online convex optimization;
- New algorithm improves substantially the dependence of the regret on the spectral gap of the underlying coommunication graph and on the compression factor;
- A new lower bound is derived, showing the tightness of the achievable performance of the proposed algorithm.

Weaknesses:
- The proof, spanning several pages, is presented more or less as a mass of inequalities, with little interpretability. Authors can and should do a better job at parsing the proof into hierarchical modules; as the proof stands, the risk of oversight in an analytical step is quite high and makes checking for the correctness of the proof a long and laborious task.

---

> ### Author Rebuttal · Authors · 2026-03-28
>
> Thanks for the insightful feedback and the interest in our work!
>
> ---
>
> > __Q1.__ The proof is presented more or less as a mass of inequalities with little interpretability.
>
> __A1.__ We apologize for the lack of clarity. In the revised version, we will reorganize the proof by dividing it into clearly hierarchical modules to improve readability.
>
> ---
>
> > __Q2.__ Why an algorithm based on revising the plays at each time instant would not necessarily be better?
>
> __A2.__ The motivation behind our algorithm design is to leverage multiple gossip steps to reduce the approximation error. As detailed in Section 3.2, both our online compressed gossip strategy and projection error compensation scheme require multiple communication rounds. However, the D-OCO protocol permits only one communication per round. To overcome this, we adopt the blocking update mechanism, which amortizes the multiple communication rounds across the rounds within each block, thereby maintaining a single communication per round.
>
> While updating the decision at every round allows learners to revise their decisions in sync with the arrival of fresh  information, each learner can only perform a single gossip step, which results in poor consensus and thus a worse approximation of the global loss function. Moreover, this approach reduces to DC-DOGD [1], which suffers worse regret bounds due to the large consensus error.
>
> Intuitively, although our algorithm updates decisions less frequently, each update is based on information with better consensus, thereby achieving better regret bounds.
>
> ---
>
> > __Q3.__ How the coupling leads to the $O(n)$ dependence.
>
> __A3.__ We apologize for the unclear explanation and will elaborate on it in the revised version. Specifically, the projection operation introduces a residual $\mathbf{r}_i(b+1)$ for each learner $i$, which appears in the bound of the compression error term $\\|\mathbf{x}_i(b+1)-\hat{\mathbf{x}}_i(b+1)\\|$. When summing over all $n$ learners, it leads to the $O(n)$ dependence in the upper bound.
>
> Concretely, without the projection error compensation scheme, we can derive
>
>  $$ \sum_{i=1}^{n} \\|\mathbf{x}_i(b+1) - \hat{\mathbf{x}}_i(b+1)\\|^2 = \sum _{i=1}^{n} \left\\|\mathbf{y}_i^{(L_1+1)}(b) + \mathbf{r}_i(b+1) - \hat{\mathbf{y}}_i^{(L_1+1)}(b)\right\\|^2$$
>
> $$ \le 2 \sum_{i=1}^{n} \left\\|\mathbf{y}_ {i}^{(L_1+1)}(b) - \hat{\mathbf{y}}_ {i}^{(L_1+1)}(b)\right\\|^2 + 2 \sum_{i=1}^{n} \\|\mathbf{r}_{i}(b+1)\\|^2 $$
>
> $$\le \frac{2}{7n} \sum_{i=1}^{n}\left( \left\\|\mathbf{y}_ {i}^{(1)}(b) - \hat{\mathbf{y}}_ {i}^{(1)}(b)\right\\|^2 + \left\\|\mathbf{y}_ {i}^{(1)}(b) - \overline{\mathbf{y}}_ i^{(1)}(b)\right\\|^2 \right)+ 2 \sum_{i=1}^{n}  \left\\|\mathbf{r}_ i(b+1)\right\\|^2 $$
>
> $$\le \frac{6}{7n} \sum_{i=1}^{n}\left(\left\\|\mathbf{x}_i(b) - \hat{\mathbf{x}}_i(b)\right\\|^2+\left\\|\mathbf{x}_i(b) - \overline{\mathbf{x}}(b)\right\\|^2\right)+O(n\eta_b^2L^2G^2),$$
>
> where the last inequality is due to Lemma D.2. By using Lemma D.2, we can further derive $e_{b+1}\leq O(\frac{e_{b}}{n}) + O(n\eta_b^2L^2G^2)\leq O(n\eta_b^2L^2G^2)$, which depends on $n$. In contrast, with our projection error compensation scheme, we can ensure $\sum_{i=1}^{n} \\|\mathbf{x}_ i(b+1) - \hat{\mathbf{x}}_ i(b+1)\\|^2\leq \frac{5}{14n}\sum_{i=1}^{n}  \left(\left\\|\mathbf{x}_ i(b) - \hat{\mathbf{x}}_ i(b)\right\\|^2 +  \left\\|\mathbf{x}_ i(b) - \overline{\mathbf{x}}(b)\right\\|^2\right) + O(\eta_b^2L^2G^2)$, thereby ensuring $e_{b+1}\leq O(\frac{e_{b}}{n}) + O(\eta_b^2L^2G^2)\leq O(\eta_b^2L^2G^2)$.
>
> ---
>
> > __Q4.__ The possibility of starting instead from the EXTRA to obtain a D-OCO algorithm?
>
> __A4__. Extending EXTRA [2] to D-OCO with compressed communication presents non-trivial theoretical and algorithmic challenges. We respectfully elaborate on them below:
>
> - __Time-varying loss functions.__ EXTRA is designed for __offline__ optimization, where the loss function is fixed over rounds. By incorporating a correction term, it allows for a constant step size, further leading to better performance in practice. The correction term relies on the gradient difference $\\|\nabla f_i(x_i(t)) - \nabla f_i(x_i(t-1))\\|$, where both gradients come from the same loss. However, in D-OCO, the functions are time-varying, which breaks the correction term.
> - __Compressed communication.__ EXTRA requires each learner to transmit exact information to its neighbors. In the compressed communication setting, directly integrating a compressor into EXTRA may fail to achieve average consensus. To resolve this, it would need to leverage difference compression [3] to design a new gossip strategy tailored to EXTRA.
>
> ---
>
> __References__
>
> [1] Distributed Online Convex Optimization with Compressed Communication. NeurIPS, 2022.
>
> [2] EXTRA: An Exact First-Order Algorithm for Decentralized Consensus Optimization. SIAM Journal on Optimization, 2015.
>
> [3] Decentralized Stochastic Optimization and Gossip Algorithms with Compressed Communication. ICML, 2019.

---

> > ### Author Rebuttal · Reviewer_xADF · 2026-04-02
> >
> > The authors have fully addressed the points I raised. Congratulations for the work.

---

> > > ### Author Response · Authors · 2026-04-02
> > >
> > > Dear Reviewer xADF,
> > >
> > > We are glad that our responses have fully addressed your concerns. We will improve  our paper according to your constructive comments!
> > >
> > > Best regards,
> > >
> > > Authors

---

### Official Review · Reviewer_xnQX · 2026-03-13

**Soundness:** 4
**Presentation:** 4
**Significance:** 4
**Originality:** 4
**Overall Recommendation:** 5
**Confidence:** 4

**Summary:**

This work studies decentralized online convex optimization with compressed communication. It proposes a new algorithm, Top-DOGD, based on a two-level blocking framework that combines an online compressed gossip strategy with a projection error compensation scheme. The main result is improved regret bounds over prior work for both convex and strongly convex losses, with substantially better dependence on the compression quality, network spectral gap, and number of learners. The paper also proves the first lower bounds for this setting, showing the upper bounds are nearly optimal up to some factors. In addition, it extends the approach to one-point and two-point bandit feedback settings and reports empirical effectiveness on online classification.

**Compliance With Llm Reviewing Policy:**

Affirmed.

**Final Justification:**

I think the authors have addressed my concerns and question. I believe my original score (5) is appropriate. I suggest an acceptance decision.

**Key Questions For Authors:**

Can you clarify whether the remaining gap in the \rho-dependence between the upper and lower bounds is fundamental or mainly a proof artifact?

**Limitations:**

I think this work does not exhibit any critical limitations from my perspectives.

**Strengths And Weaknesses:**

Strengths:
1. This work substantially improves the regret bounds for decentralized OCO with compressed communication, especially in the dependence on the compression factor, the spectral gap, and the number of learners. These are solid technical results in communication-efficient decentralized OCO.
2. The proposed Top-DOGD is not just a simple tweak of prior techniques. It combines an online compressed gossip strategy with a projection error compensation scheme inside a two-level blocking framework. The authors explains why this design is needed, since the method must simultaneously control consensus, compression, and projection errors while still respecting the one-communication-per-round setting.
3. Another notable strength is that the author establishes the first lower bounds for this compressed-communication setting. These lower bounds show the results are near-optimal up to the spectral-gap and polylogarithmic factors, which makes this work more complete and convincing.
4. They also extends the full-information framework to both one-point and two-point bandit feedback, again improving the dependence on the critical factors, e.g., compression factor, the spectral gap, and the number of learners.

Weaknesses:
1. I think this paper is pretty complete from every perspectives. One minor weak point might be regarding the optimality story. The proved lower bound of the problem match the upper bounds only up to the spectral-gap dependence and polylogarithmic factors in the number of the local learners. But it seems not closed regarding the dependence on the spectral-gap (\rho^{-1} vs. rho^{-1/4} in convex setting and \rho^{-2} vs. rho^{-1/2} in strongly convex setting).

---

> ### Author Rebuttal · Authors · 2026-03-28
>
> Many thanks for the constructive reviews and your positive recognition of our paper.
>
> ---
>
> >  __Q1.__ Whether the remaining gap in the $\rho$-dependence between the upper and lower bounds is fundamental or mainly a proof artifact?
>
> __A1.__ The gap in $\rho$-dependence originates from the Choco-gossip [1] employed in our algorithm, rather than a proof artifact.
>
> - Regarding the lower bound for convex loss functions, the $\rho^{-1/4}$ dependence is consistent with the tight lower bound established by [2] in the standard uncompressed D-OCO setting. Moreover, the $\omega^{-1/2}$ dependence also matches the corresponding upper bound (Theorem 4.1). These together confirm the tightness of our lower bound.
>
> - The gap lies in the upper bound. Specifically, the $\rho$-dependence in our upper bound stems from the approximation error of Choco-gossip, which ensures that the sum of the consensus error and compression error $e_t$ satisfies $e_ {t+L_1} \leq (1-\frac{\rho^2\omega}{82})^{L_1} e_t.$ To control the errors, we need to set the number of gossip rounds $L_1=\Theta(\omega^{-1}\rho^{-2}\ln n)$. However, it results in multiple communication rounds per update, which is not allowed in D-OCO. To ensure the number of communication rounds remains one, we utilize the blocking update mechanism, which further leads to the $\rho^{-1}$ dependence in the regret bound for convex loss functions.
>
> - To fill this theoretical gap, one would need to design an accelerated gossip strategy under compressed communication to achieve faster consensus, thereby allowing for a smaller block size and reducing the dependence on $\rho$. We leave this for future work.
>
> ---
>
> We hope the above responses address your concerns and are happy to provide further clarification if needed.
>
> ---
>
> __References__
>
> [1] Decentralized Stochastic Optimization and Gossip Algorithms with Compressed Communication. ICML, 2019.
>
> [2] Nearly Optimal Regret for Decentralized Online Convex Optimization. COLT, 2024.

---

> > ### Author Rebuttal · Reviewer_xnQX · 2026-04-03
> >
> > I appreciate the detailed explanation regarding the bound gap. Thank you.
> >
> > I would suggest an acceptance decision.

---

> > > ### Author Response · Authors · 2026-04-03
> > >
> > > Dear Reviewer xnQX,
> > >
> > > We are grateful that our rebuttal has addressed your question, and we sincerely appreciate your positive assessment of our work.
> > >
> > > Best regards,
> > >
> > > Authors

---

### Decision · Program_Chairs · 2026-04-30

**Decision:**

Accept (regular)

**Comment:**

The reviewers agree that this is a strong theoretical contribution that improves regret bounds for decentralized OCO with compressed communication. It also proves lower bounds for this setting, showing near-optimality. The two-level blocking framework combining online compressed gossip with projection error compensation is perceived as novel and well-motivated, and the authors are able to successfully extend the framework to bandit feedback settings.

On the negative side are concerns of dense presentation, and limited pratical discussions in the main text. There is also a remaining rho gap that would be interesting to understand better. Some reviewers also found the assumptions to be relatively strong.

I agree that this is a strong and mature theoretical paper with substantial performance improvements and novel lower bounds. Personally, I would have like to also understnad how large the hidden constants are, especially under high compression. This could matter in pratice, since constant overhead could offset asymptotic gains in certain regimes.

Still, the contribution is significant, complete, and well-defended. I therefore recommend Accept.